# Iterative State Estimation in Non-linear Dynamical Systems Using Approximate Expectation Propagation

**Sanket Kamthe**[*]  *sanketkamthe@gmail.com*
*Department of Computing, Imperial College London*
*Chief Technology Office, JP Morgan Chase*

**So Takao**[*]  *so.takao@ucl.ac.uk*
*UCL Centre for Artificial Intelligence, University College London*

**Shakir Mohamed**[†]
*Department of Computer Science, University College London*

**Marc Peter Deisenroth**
*UCL Centre for Artificial Intelligence, University College London*

**Reviewed on OpenReview:** *https://openreview.net/forum?id=xyt4wfdo4J*

## Abstract

Bayesian inference in non-linear dynamical systems seeks to find good posterior approximations of a latent state given a sequence of observations. Gaussian filters and smoothers, including the (extended/unscented) Kalman filter/smoother, which are commonly used in engineering applications, yield Gaussian posteriors on the latent state. While they are computationally efficient, they are often criticised for their crude approximation of the posterior state distribution. In this paper, we address this criticism by proposing a message passing scheme for iterative state estimation in non-linear dynamical systems, which yields more informative (Gaussian) posteriors on the latent states. Our message passing scheme is based on expectation propagation (EP). We prove that classical Rauch–Tung–Striebel (RTS) smoothers, such as the extended Kalman smoother (EKS) or the unscented Kalman smoother (UKS), are special cases of our message passing scheme. Running the message passing scheme more than once can lead to significant improvements of the classical RTS smoothers, so that more informative state estimates can be obtained. We address potential convergence issues of EP by generalising our state estimation framework to damped updates and the consideration of general $\alpha$-divergences.

## 1 Introduction

Dynamical systems (state space models) are powerful tools for describing (latent) time-series with noisy observations. Dynamical systems are commonly used in signal processing, control theory, and climate science. Given a sequence of noisy observations, estimating the (posterior) distribution of the latent states in a dynamical system plays an important role, e.g., in optimal control theory (Åström, 2006; Bertsekas, 2005), data assimilation (Lakshmivarahan & Stensrud, 2009), or robotics (Thrun et al., 2005; Barfoot, 2017).

For linear systems with Gaussian noise, an optimal estimate of posterior distributions of the latent states conditioned on the observations up to the current time step is given in closed form by the Kalman filter (Kalman, 1960). The estimate conditioned on all (including future) measurements of a finite time series is given by the Rauch–Tung–Striebel (RTS) (Rauch et al., 1965) smoother. The efficiency of the Kalman

---

[*]Equal contribution. [†]Also with DeepMind.

filter and the RTS smoother stems from the closure property of Gaussian densities under linear/affine transformations, conditioning, and marginalisation. These closure properties allow us to derive recursive filtering and smoothing equations in closed form (Anderson & Moore, 2005).

In non-linear systems, these computationally convenient properties no longer hold; in particular, a non-linear transformation of a Gaussian density yields a non-Gaussian density. Typically, we tackle this challenge either by approximating a non-Gaussian density by a Gaussian density (Maybeck, 1979; Julier et al., 2000; Arasaratnam & Haykin, 2009) or by using sampling-based methods, e.g., the particle filter (Doucet et al., 2000), to represent non-Gaussian densities by means of particles. In this article, we focus on the former, i.e., finding better Gaussian approximations for non-Gaussian densities.

By making a Gaussian approximation to the distribution of the latent states, we are in fact either implicitly or explicitly linearising the non-linear transformation. The quality of these approximations depends on the mechanism used for the linearisation. A crude approximation may introduce large errors, which we can alleviate by iteratively correcting previous estimates (Lefebvre et al., 2004). For example, in the iterative extended Kalman filter (IEKF) (Corporation, 1974), measurement updates are carried out iteratively until convergence, where the point around which one linearises the measurement function is refined iteratively. This is shown to be equivalent to a Gauß–Newton method for MAP estimation (Bell & Cathey, 1993). An iterative version of the extended Kalman smoother has also been proposed (Bell, 1994), which, again is equivalent to a Gauß–Newton method for MAP estimation. Recent progress (Raitoharju et al., 2018; Gultekin & Paisley, 2017) shows that the Bayesian framework and iterative updates can give significant improvement over standard filtering and smoothing.

All Gaussian filtering systems assume that we can project the true posterior onto a Gaussian (Thrun et al., 2005) with a reasonable approximation error. More generally, this can be interpreted as an approximate assumed density filter (ADF) with a Gaussian assumption on densities. In the ADF, the approximate posterior is obtained by minimising the Kullback–Leibler (KL) divergence between the true posterior and the assumed density. For a Gaussian assumed density, this amounts to computing the first two moments of the true posterior and approximating it with a Gaussian that possesses the same first two moments. In the Assumed Density Smoother, as well in the iterative Kalman smoothers, there is no mechanism to revisit previously filtered estimates and correct them based on all the observations.

This shortcoming was addressed by expectation propagation (EP) proposed by Minka (2001a;b); Opper & Winther (2001). EP extends the ADF by adding a mechanism to iteratively update previous computations without double-counting observations. For linear Gaussian systems, one full iteration of EP is identical to the Kalman filter-smoother (Minka, 2001a; Heskes & Zoeter, 2002; Qi & Minka, 2003). Hence, EP is optimal for linear systems.

One can extend EP to non-linear systems in a manner that is similar to the generalised Gaussian filters. This extension is motivated by results in (Dehaene & Barthelmé, 2016), who show that EP has an order of magnitude lower asymptotic error bound compared to a canonical Gaussian approximation for non-linear systems. Furthermore, applying EP to non-linear systems allows us to perform multiple sweeps of RTS smoothing, iterating over measurement and transition non-linearities conditioned on all available observations. We note that the IEKF systems can only iterate over measurement non-linearities and the IEKS (Bell, 1994), being a MAP estimator, does not properly account for posterior uncertainties. EP can be implemented as a message passing algorithm. Hence, it can be used as a distributed (and asynchronous) state estimator and is not limited to the sequential nature of iterative Kalman filters.

One challenge with implementing EP is that its efficacy relies on the ease of computing moments with respect to the true posterior; for non-linear systems, the exact posterior is often intractable. In this work, we propose to approximate the true posterior by linearisation methods, such as the unscented transform (Julier et al., 2000). Hence, we only implement *approximate* EP when we apply EP to non-linear systems. We prove that this approximate EP exactly recovers the respective RTS smoothers, e.g., with the unscented transform approximation we obtain the unscented Kalman smoother (Särkkä, 2008), and if we explicitly linearise functions, we obtain the extended Kalman smoother. This allows us to extend EP to all non-linear Gaussian filters/smoothers, and generalise them to a setting where posterior state distributions can be iteratively refined to yield better Gaussian approximations.

**Contribution** [1]

1. We prove that any Gaussian filter for non-linear dynamical systems is equivalent to a single iteration of *approximate* EP. This equivalence will allow us to implement (approximate) EP using Kalman-filter-style updates and without explicitly computing gradients of the log-partition function. Additionally, we can improve non-linear Gaussian smoothers by iteratively refining the marginal posterior distributions using approximate EP.

2. Because of approximate EP's equivalence with Gaussian smoothing, we can further improve estimation of the marginal posteriors by pre-existing techniques in the EP literature: (1) Damping, which reduces oscillatory behaviour; (2) Power EP, which allows us to interpolate between EP (mode averaging) and variational Bayes (mode seeking) for iterative smoothing in non-linear state-space models.

We use a classic non-linear system benchmark, the uniform nonlinear growth model (UNGM) (Doucet et al., 2000) with a fixed seed as a running example to illustrate properties of the iterative smoothing scheme. The article is organised as follows. In Section 2, we establish the notation and review classic Kalman filtering and smoothing, which we then apply to the UNGM benchmark. In Section 3, we introduce EP as a way for iteratively computing posterior distributions on latent states. Furthermore, we prove that the first iteration of derivative-based (approximate) EP is equivalent to smoothing with non-linear Kalman filters, which does not require any derivatives. We demonstrate that EP further improves the standard smoothing results of the UNGM example. Section 4.2 then addresses potential limit-cycle behaviours of EP by using damped updates (Heskes & Zoeter, 2003), and then further improves the results by using $\alpha$-divergence EP (power EP) (Minka, 2004). In Section 5, we analyse the efficacy of our method and its sensitivity with respect to $\alpha$, the damping parameter, and the linearisation technique.

## 2    Gaussian Filtering and Smoothing

Consider the state-space models of the form

$$\boldsymbol{x}_t = \boldsymbol{f}(\boldsymbol{x}_{t-1}) + \boldsymbol{w}_t, \tag{1}$$

$$\boldsymbol{y}_t = \boldsymbol{h}(\boldsymbol{x}_t) + \boldsymbol{v}_t, \tag{2}$$

where $\boldsymbol{x}_t \in \mathbb{R}^D$ is an unobserved/latent state and $\boldsymbol{y}_t \in \mathbb{R}^E$ is a measurement at time step $t = 1, \ldots, T$. Moreover, $\boldsymbol{f}$ and $\boldsymbol{h}$ are the (non-linear) transition and measurement functions, respectively, and the noise processes $\boldsymbol{w}_t$ and $\boldsymbol{v}_t$ are i.i.d. zero-mean Gaussian with diagonal covariances $\boldsymbol{Q}$ and $\boldsymbol{R}$, respectively. The initial state distribution is given by $p(\boldsymbol{x}_0) = \mathcal{N}(\boldsymbol{x}_0 \,|\, \boldsymbol{\mu}_0, \boldsymbol{\Sigma}_0,)$. The purpose of this section is to provide background details on Gaussian filtering and smoothing algorithms on the state-space model (1)– (2).

### 2.1    Optimal Filtering and Smoothing

Bayesian filtering corresponds to finding a posterior distribution $p(\boldsymbol{x}_t|\boldsymbol{y}_{1:t})$ on the unobserved latent state $\boldsymbol{x}_t$, given observations $\boldsymbol{y}_{1:t} := (\boldsymbol{y}_1, \ldots, \boldsymbol{y}_t)$ up until the most recent time step $t$. It proceeds by sequentially computing a time update and a measurement update (Anderson & Moore, 2005), as we explain below.

Given a filtering distribution $p(\boldsymbol{x}_{t-1}|\boldsymbol{y}_{1:t-1})$, the *time update*

$$p(\boldsymbol{x}_t|\boldsymbol{y}_{1:t-1}) = \int p(\boldsymbol{x}_t|\boldsymbol{x}_{t-1})p(\boldsymbol{x}_{t-1}|\boldsymbol{y}_{1:t-1}) \, \mathrm{d}\boldsymbol{x}_{t-1} \tag{3}$$

computes a predictive distribution of the state $\boldsymbol{x}_t$ given observations up to time step $t - 1$. Here, $p(\boldsymbol{x}_t|\boldsymbol{x}_{t-1}) = \mathcal{N}(\boldsymbol{x}_t \,|\, \boldsymbol{f}(\boldsymbol{x}_{t-1}), \boldsymbol{Q})$ is the transition probability, which follows from the dynamical system (1). The *measurement update* then incorporates the observation $\boldsymbol{y}_t$ at time $t$ by computing

$$p(\boldsymbol{x}_t|\boldsymbol{y}_{1:t}) = \frac{p(\boldsymbol{y}_t|\boldsymbol{x}_t)p(\boldsymbol{x}_t|\boldsymbol{y}_{1:t-1})}{\int p(\boldsymbol{y}_t|\boldsymbol{x}_t)p(\boldsymbol{x}_t|\boldsymbol{y}_{1:t-1})\mathrm{d}\boldsymbol{x}_t} \propto p(\boldsymbol{y}_t|\boldsymbol{x}_t)p(\boldsymbol{x}_t|\boldsymbol{y}_{1:t-1}), \tag{4}$$

---

[1]Code available at `https://github.com/sanket-kamthe/EPyStateEstimator`

which yields the filtering distribution. Here, $p(\boldsymbol{y}_t|\boldsymbol{x}_t) = \mathcal{N}(\boldsymbol{y}_t \,|\, \boldsymbol{h}(\boldsymbol{x}_t), \boldsymbol{R})$ is the observation likelihood, which follows from (2).

On the other hand, in Bayesian smoothing, the posterior distribution $p(\boldsymbol{x}_t|\boldsymbol{y}_{1:T})$ of the latent state $\boldsymbol{x}_t$ given all observations in the time series is computed. Here, we consider the Rauch–Tung–Striebel (RTS) smoother, which computes this posterior iteratively by applying the *smoothing update*

$$p(\boldsymbol{x}_t|\boldsymbol{y}_{1:T}) = \int p(\boldsymbol{x}_t|\boldsymbol{x}_{t+1}, \boldsymbol{y}_{1:T}) p(\boldsymbol{x}_{t+1}|\boldsymbol{y}_{1:T}) \mathrm{d}\boldsymbol{x}_{t+1} = \int p(\boldsymbol{x}_t|\boldsymbol{x}_{t+1}, \boldsymbol{y}_{1:t}) p(\boldsymbol{x}_{t+1}|\boldsymbol{y}_{1:T}) \mathrm{d}\boldsymbol{x}_{t+1}, \qquad (5)$$

where we used the Markov property $p(\boldsymbol{x}_t|\boldsymbol{x}_{t+1}, \boldsymbol{y}_{1:t}) = p(\boldsymbol{x}_t|\boldsymbol{x}_{t+1}, \boldsymbol{y}_{1:T})$.

Equations (3)–(5) are generally intractable for non-linear systems. However, if we focus on Gaussian approximations to the filtering and smoothing distributions, we can derive equations for the time update, measurement update and the smoothing update, provided that we can approximate the densities $p(\boldsymbol{x}_{t-1}, \boldsymbol{x}_t|\boldsymbol{y}_{1:t-1})$, $p(\boldsymbol{x}_t, \boldsymbol{y}_t|\boldsymbol{y}_{1:t-1})$ and $p(\boldsymbol{x}_t, \boldsymbol{x}_{t+1}|\boldsymbol{y}_{1:t})$ respectively by joint Gaussian distributions (Deisenroth, 2010; Särkkä et al., 2015).

To see this, let us take for example an approximation of $p(\boldsymbol{x}_t, \boldsymbol{y}_t|\boldsymbol{y}_{1:t-1})$ as a joint Gaussian

$$p(\boldsymbol{x}_t, \boldsymbol{y}_t|\boldsymbol{y}_{1:t-1}) = \mathcal{N}\left( \begin{pmatrix} \boldsymbol{x}_t \\ \boldsymbol{y}_t \end{pmatrix} \,\middle|\, \begin{pmatrix} \boldsymbol{\mu}_{t|t-1}^x \\ \boldsymbol{\mu}_{t|t-1}^y \end{pmatrix}, \begin{pmatrix} \boldsymbol{\Sigma}_{t|t-1}^x & \boldsymbol{\Sigma}_{t|t-1}^{xy} \\ \boldsymbol{\Sigma}_{t|t-1}^{yx} & \boldsymbol{\Sigma}_{t|t-1}^y \end{pmatrix} \right). \qquad (6)$$

The subscripts $t|t-1$ indicate that the statistics of random variables (e.g., the mean) at time step $t$ are conditioned on information/observations up to time step $t-1$ and the superscripts indicate whether the quantity is observed in latent or observed space (for example $\boldsymbol{\Sigma}_{t|1:t-1}^{xy} := \mathrm{cov}[\boldsymbol{x}_t, \boldsymbol{y}_t|\boldsymbol{y}_1, \ldots, \boldsymbol{y}_{t-1}]$).

By exploiting the joint Gaussianity in (6), the measurement update equation can be computed by Gaussian conditioning, leading us to the approximate filter distribution $p(\boldsymbol{x}_t|\boldsymbol{y}_{1:t}) \approx \mathcal{N}(\boldsymbol{x}_t|\boldsymbol{\mu}_{t|t}^x, \boldsymbol{\Sigma}_{t|t}^x)$, where

$$\boldsymbol{\mu}_{t|t}^x = \boldsymbol{\mu}_{t|t-1}^x + \boldsymbol{K}_t(\boldsymbol{y}_t - \boldsymbol{\mu}_{t|t-1}^y), \qquad (7)$$

$$\boldsymbol{\Sigma}_{t|t}^x = \boldsymbol{\Sigma}_{t|t-1}^x - \boldsymbol{K}_t \boldsymbol{\Sigma}_{t|t-1}^{yx}. \qquad (8)$$

Here, the matrix $\boldsymbol{K}_t := \boldsymbol{\Sigma}_{t|t-1}^{xy} (\boldsymbol{\Sigma}_{t|t-1}^y)^{-1}$ corresponds to the Kalman gain in linear systems. Now, to arrive at the Gaussian approximation (6), we do moment matching, which requires one to compute the means and covariances of the true distribution $p(\boldsymbol{x}_t, \boldsymbol{y}_t|\boldsymbol{y}_{1:t-1})$. This has the following exact expressions

$$\boldsymbol{\mu}_{t|t-1}^x = \int \boldsymbol{f}(\boldsymbol{x}_{t-1}) p(\boldsymbol{x}_{t-1}|\boldsymbol{y}_{1:t-1}) \mathrm{d}\boldsymbol{x}_{t-1}, \qquad (9)$$

$$\boldsymbol{\Sigma}_{t|t-1}^x = \int \boldsymbol{f}(\boldsymbol{x}_{t-1})(\boldsymbol{f}(\boldsymbol{x}_{t-1}))^\top p(\boldsymbol{x}_{t-1}|\boldsymbol{y}_{1:t-1}) \mathrm{d}\boldsymbol{x}_{t-1} - \boldsymbol{\mu}_{t|t-1}^x (\boldsymbol{\mu}_{t|t-1}^x)^\top + \boldsymbol{Q}, \qquad (10)$$

$$\boldsymbol{\mu}_{t|t-1}^y = \int \boldsymbol{h}(\boldsymbol{x}_t) p(\boldsymbol{x}_t|\boldsymbol{y}_{1:t-1}) \,\mathrm{d}\boldsymbol{x}_t, \qquad (11)$$

$$\boldsymbol{\Sigma}_{t|t-1}^y = \int \boldsymbol{h}(\boldsymbol{x}_t)(\boldsymbol{h}(\boldsymbol{x}_t))^\top p(\boldsymbol{x}_t|\boldsymbol{y}_{1:t-1}) \,\mathrm{d}\boldsymbol{x}_t - \boldsymbol{\mu}_{t|t-1}^y (\boldsymbol{\mu}_{t|t-1}^y)^\top + \boldsymbol{R}, \qquad (12)$$

$$\boldsymbol{\Sigma}_{t|t-1}^{xy} = \int \boldsymbol{x}_t(\boldsymbol{h}(\boldsymbol{x}_t))^\top p(\boldsymbol{x}_t|\boldsymbol{y}_{1:t-1}) \,\mathrm{d}\boldsymbol{x}_t - \boldsymbol{\mu}_{t|t-1}^x (\boldsymbol{\mu}_{t|t-1}^y)^\top. \qquad (13)$$

Generally, these cannot be computed in closed form when $\boldsymbol{f}$ and $\boldsymbol{h}$ are non-linear. Hence, we resort to methods that approximate these integrals, such as linearising the non-linear terms or applying the unscented transform (Deisenroth & Ohlsson, 2011; Särkkä, 2013).

Similar update equations can be obtained for the RTS smoother by approximating the joint distribution $p(\boldsymbol{x}_t, \boldsymbol{x}_{t+1}|\boldsymbol{y}_{1:t})$ by a joint Gaussian (Deisenroth et al., 2012), which leads to closed-form equations for the mean and covariance of the smoothing distribution, given by

$$\boldsymbol{\mu}_{t|T}^x = \boldsymbol{\mu}_{t|t}^x + \boldsymbol{L}_t(\boldsymbol{\mu}_{t+1|T}^x - \boldsymbol{\mu}_{t+1|t}^x), \qquad (14)$$

$$\boldsymbol{\Sigma}_{t|T}^x = \boldsymbol{\Sigma}_{t+1|t+1}^x + \boldsymbol{L}_t \left( \boldsymbol{\Sigma}_{t+1|T}^x - \boldsymbol{\Sigma}_{t+1|t}^x \right) \boldsymbol{L}_t^\top, \qquad (15)$$

respectively. Here, the matrix $\boldsymbol{L}_t = \boldsymbol{\Sigma}^x_{t,t+1|t}(\boldsymbol{\Sigma}^x_{t+1|t})^{-1}$ is called the *smoothing gain*, where $\boldsymbol{\Sigma}^x_{t,t+1|t}$ denotes the cross-covariance between $\boldsymbol{x}_t$ and $\boldsymbol{x}_{t+1}$, which has the true expression

$$\boldsymbol{\Sigma}^x_{t,t+1|t} = \int \boldsymbol{x}_t(\boldsymbol{f}(\boldsymbol{x}_t))^\top p(\boldsymbol{x}_t|\boldsymbol{y}_{1:t})\,\mathrm{d}\boldsymbol{x}_t - \boldsymbol{\mu}^x_{t|t}(\boldsymbol{\mu}^x_{t+1|t})^\top. \tag{16}$$

Again, for a non-linear transition function $\boldsymbol{f}$, the integral is intractable. Hence, we approximate it by means of, for example, linearisation or the unscented transformation.

For the time update, if we similarly approximate the distribution $p(\boldsymbol{x}_{t-1}, \boldsymbol{x}_t|\boldsymbol{y}_{1:t-1})$ by a joint Gaussian, we can obtain an approximation to the predictive distribution (3) by a simple Gaussian marginalisation.

## 2.2  Implicit Linearisation in Non-linear Gaussian Filters and Smoothers

Applying moment matching to approximate the true distributions $p(\boldsymbol{x}_{t-1}, \boldsymbol{x}_t|\boldsymbol{y}_{1:t-1})$, $p(\boldsymbol{x}_t, \boldsymbol{y}_t|\boldsymbol{y}_{1:t-1})$, $p(\boldsymbol{x}_t, \boldsymbol{x}_{t+1}|\boldsymbol{y}_{1:t})$ as joint Gaussians in Section 2.1 implicitly linearises the respective transition and measurement functions, as two random variables are jointly Gaussian if and only if they are related by an affine transformation up to zero-mean Gaussian noise. This is stated more precisely in the following Lemma.

**Lemma 1** (Implicit linearisation). A random variable $(\boldsymbol{x}, \boldsymbol{y})$ is jointly Gaussian if and only if there exist matrices $\boldsymbol{M} \in \mathbb{R}^{E \times D}, \boldsymbol{P} \in \mathbb{R}^{E \times E}$ and a vector $\boldsymbol{v} \in \mathbb{R}^E$ such that $\boldsymbol{y} = \boldsymbol{M}\boldsymbol{x} + \boldsymbol{v} + \boldsymbol{\varepsilon}$, where $\boldsymbol{\varepsilon} \sim \mathcal{N}(\boldsymbol{0}, \boldsymbol{P})$.

*Proof.* If $(\boldsymbol{x}, \boldsymbol{y})$ is jointly Gaussian, we can write

$$\begin{pmatrix} \boldsymbol{x} \\ \boldsymbol{y} \end{pmatrix} \sim \mathcal{N}\left( \begin{pmatrix} \boldsymbol{\mu}^x \\ \boldsymbol{\mu}^y \end{pmatrix}, \begin{pmatrix} \boldsymbol{\Sigma}^x & \boldsymbol{\Sigma}^{xy} \\ \boldsymbol{\Sigma}^{yx} & \boldsymbol{\Sigma}^y \end{pmatrix} \right), \tag{17}$$

where $\boldsymbol{\Sigma}^{yx} = (\boldsymbol{\Sigma}^{xy})^\top$. On the other hand, if $\boldsymbol{y} = \boldsymbol{M}\boldsymbol{x} + \boldsymbol{v} + \boldsymbol{\varepsilon}$ with $\boldsymbol{\varepsilon} \sim \mathcal{N}(\boldsymbol{0}, \boldsymbol{P})$ were to hold, then for $\boldsymbol{x} \sim \mathcal{N}(\boldsymbol{\mu}^x, \boldsymbol{\Sigma}^x)$, we have

$$\begin{pmatrix} \boldsymbol{x} \\ \boldsymbol{y} \end{pmatrix} \sim \mathcal{N}\left( \begin{pmatrix} \boldsymbol{\mu}^x \\ \boldsymbol{M}\boldsymbol{\mu}^x + \boldsymbol{v} \end{pmatrix}, \begin{pmatrix} \boldsymbol{\Sigma}^x & \boldsymbol{\Sigma}^x \boldsymbol{M}^\top \\ \boldsymbol{M}\boldsymbol{\Sigma}^x & \boldsymbol{M}\boldsymbol{\Sigma}^x \boldsymbol{M}^\top + \boldsymbol{P} \end{pmatrix} \right). \tag{18}$$

We see that the RHS of expressions (17) and (18) agree if and only if

$$\boldsymbol{M} = \boldsymbol{\Sigma}^{yx}(\boldsymbol{\Sigma}^x)^{-1}, \tag{19}$$

$$\boldsymbol{P} = \boldsymbol{\Sigma}^y - \boldsymbol{M}\boldsymbol{\Sigma}^x \boldsymbol{M}^\top, \tag{20}$$

$$\boldsymbol{v} = \boldsymbol{\mu}^y - \boldsymbol{M}\boldsymbol{\mu}^x. \tag{21}$$

It is therefore always possible to write $\boldsymbol{y} = \boldsymbol{M}\boldsymbol{x} + \boldsymbol{v} + \boldsymbol{\varepsilon}$ for random variables $\boldsymbol{x}$ and $\boldsymbol{y}$ that are jointly Gaussian and vice versa. Note that the matrix $\boldsymbol{P}$ in (20) is guaranteed to be positive semidefinite as it is exactly the expression for the covariance of the conditional distribution $p(\boldsymbol{y}|\boldsymbol{x})$. $\qquad \square$

We illustrate this idea of linearisation that arises as a result of forming joint Gaussians (and vice versa) in the following two examples. In the first, we look at the case when we use the Taylor approximation to *explicitly* linearise the nonlinear transition function and how that results in a joint Gaussian approximation between two random variables. In the second example, we look at using particle-based methods to approximate a general joint distribution $p(\boldsymbol{x}, \boldsymbol{y})$ as joint Gaussians and how that *implicitly* leads to a linear relationship between $\boldsymbol{x}$ and $\boldsymbol{y}$.

*Example* 1 (Taylor approximation). Let $\boldsymbol{x} \sim \mathcal{N}(\boldsymbol{\mu}^x, \boldsymbol{\Sigma}^x)$ and $\boldsymbol{y}|\boldsymbol{x} \sim \mathcal{N}(\boldsymbol{g}(\boldsymbol{x}), \boldsymbol{P})$, where $\boldsymbol{g}$ is a non-linear differentiable function. Since the joint distribution $p(\boldsymbol{x}, \boldsymbol{y}) = \mathcal{N}(\boldsymbol{x}|\boldsymbol{\mu}^x, \boldsymbol{\Sigma}^x)\mathcal{N}(\boldsymbol{y}|\boldsymbol{g}(\boldsymbol{x}), \boldsymbol{P})$ is non-Gaussian, we approximate it by a Gaussian by means of explicitly linearising $\boldsymbol{g}$ around $\boldsymbol{\mu}^x$.

$$\boldsymbol{g}(\boldsymbol{x}) \approx \boldsymbol{g}(\boldsymbol{\mu}^x) + \boldsymbol{M}(\boldsymbol{x} - \boldsymbol{\mu}^x), \tag{22}$$

where $\boldsymbol{M} := \nabla \boldsymbol{g}(\boldsymbol{\mu}^x)$. This gives us the linear relation $\boldsymbol{y} = \boldsymbol{M}\boldsymbol{x} + \boldsymbol{v} + \boldsymbol{\varepsilon}$, $\boldsymbol{\varepsilon} \sim \mathcal{N}(\boldsymbol{0}, \boldsymbol{P})$, where $\boldsymbol{v} = \boldsymbol{g}(\boldsymbol{\mu}^x) - \boldsymbol{M}\boldsymbol{\mu}^x$ and the following Gaussian approximation to $p(\boldsymbol{x}, \boldsymbol{y})$ holds:

$$\begin{pmatrix} \boldsymbol{x} \\ \boldsymbol{y} \end{pmatrix} \stackrel{\text{approx}}{\sim} \mathcal{N}\left( \begin{pmatrix} \boldsymbol{\mu}^x \\ \boldsymbol{g}(\boldsymbol{\mu}^x) \end{pmatrix}, \begin{pmatrix} \boldsymbol{\Sigma}^x & \boldsymbol{\Sigma}^x \boldsymbol{M}^T \\ \boldsymbol{M}\boldsymbol{\Sigma}^x & \boldsymbol{M}\boldsymbol{\Sigma}^x \boldsymbol{M}^\top + \boldsymbol{P} \end{pmatrix} \right). \tag{23}$$

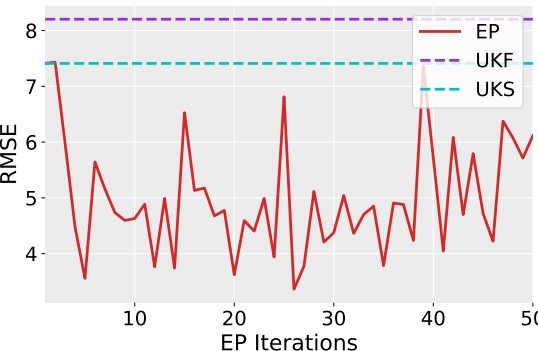 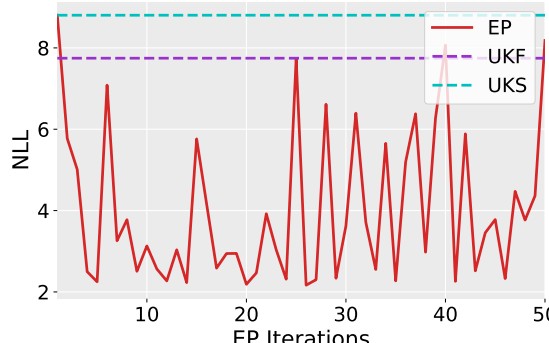

Figure 1: Root mean square error (RMSE) and negative log-likelihood (NLL) performance of estimation algorithms applied to the UNGM example (single seed). The unscented Kalman filter and the unscented RTS smoother form the baselines (dashed lines). EP starts from the RTS smoothing baseline, as its first iteration is identical to the RTS smoother. We can see that the EP estimate improves on the smoothing baselines with an increasing number of EP iterations. The oscillatory behaviour of EP can be addressed by damping, which we discuss in Section 4.1.

*Example* 2 (Monte Carlo and unscented transformation). Alternatively, we can approximate the joint distribution $p(\boldsymbol{x}, \boldsymbol{y})$ in the previous example via a particle-based representation, that is, represent $p(\boldsymbol{x})$ using a collection of particles (which can be determined deterministically as in the unscented transformation or stochastically as in the particle filter) and propagate them forward by $\boldsymbol{g}$ to compute the moments $\boldsymbol{\mu}^y, \boldsymbol{\Sigma}^y, \boldsymbol{\Sigma}^{xy}$ empirically. We can then use the computed moments to deduce the implicit linearisation $\boldsymbol{y} = \boldsymbol{M}\boldsymbol{x} + \boldsymbol{v} + \boldsymbol{\varepsilon}$ with $\boldsymbol{\varepsilon} \sim \mathcal{N}(\boldsymbol{0}, \boldsymbol{P})$, where $\boldsymbol{M}, \boldsymbol{P}, \boldsymbol{v}$ are computed as in (19)–(21). The Monte Carlo and unscented transformation considered in our experiments in Section 5 follow this framework.

Commonly used Bayesian filters and smoothers for non-linear systems are the extended Kalman filter (EKF), the unscented Kalman filter (UKF) (Julier et al., 2000), and the cubature Kalman filter (CKF) (Arasaratnam & Haykin, 2009), all of which compute means and covariances of the unobserved state $\boldsymbol{x}_t$. Since all update equations of these non-linear filters and smoothers can be expressed as (7), (8), and (14), (15) respectively, these methods either explicitly (e.g., EKF) or implicitly (e.g., UKF, CKF) linearise the transition and measurement operators in (1)–(2), which incurs approximation errors.

To account for the approximation errors, we may improve the filtering/smoothing results by iteratively refining the state estimate. For example, in the iterative extended Kalman filter (IEKF) (Senne, 1972), an iterative measurement update is used to improve the standard EKF solution. On the other hand, similar results for smoothing are not well explored. However, we would expect to see similar improvements as obtained in filtering by iteratively linearising the transition function as well. This is challenging, as, unlike the measurement function that only depends on the measurement at time step $t$, the smoothing result depends on all observations from time steps 1 to $T$.

The approach that we propose here is to apply Expectation Propagation (EP), a general approximate inference scheme that iteratively refine approximations of the target distribution (Minka, 2001a). Standard EP is only applicable for linear functions, but it exactly recovers the Kalman and RTS smoother for linear systems (Minka, 2001a). Similar to Kalman-based iterative filters, we expect the EP updates to iteratively improve smoothing results in the non-linear case.

## 2.3 RTS Smoothing Applied to a Non-linear Benchmark

Before describing EP in detail (see Section 3), we motivate the usefulness of its iterative refinement of the posterior state distribution on an instance of the well-known non-linear benchmark:

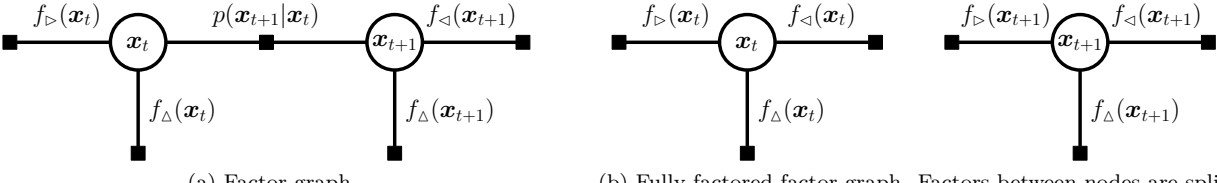

(a) Factor graph        (b) Fully-factored factor graph. Factors between nodes are split.

Figure 2: Factor-graph representations of a dynamical system. (a) Exact factor graph of a dynamical system. Factors (black squares) correspond to conditional probability distributions, e.g., $p(\boldsymbol{x}_{t+1}|\boldsymbol{x}_t)$ for the factor between $\boldsymbol{x}_t$ and $\boldsymbol{x}_{t+1}$. Circles are random variables. Observations $\boldsymbol{y}_t$ are not displayed as they are deterministic. (b) The fully-factored factor graph illustrates how local messages are being sent to update the marginal posterior at each time step. The splitted factors are obtained by applying belief propagation on (a), that is $f_\triangleleft(\boldsymbol{x}_t) = \int p(\boldsymbol{x}_{t+1}|\boldsymbol{x}_t)f_\triangle(\boldsymbol{x}_{t+1})f_\triangleleft(\boldsymbol{x}_{t+1})\mathrm{d}\boldsymbol{x}_{t+1}$ and $f_\triangleright(\boldsymbol{x}_{t+1}) = \int p(\boldsymbol{x}_{t+1}|\boldsymbol{x}_t)f_\triangleright(\boldsymbol{x}_t)f_\triangle(\boldsymbol{x}_t)\mathrm{d}\boldsymbol{x}_t$.

$$x_t = \frac{x_{t-1}}{2} + \frac{25x_{t-1}}{1+x_{t-1}^2} + 8\cos(1.2(t-1)) + w_t, \tag{24}$$

$$y_t = \frac{x_t^2}{20} + v_t, \tag{25}$$

where we use the same set-up as in (Doucet et al., 2000): $w_t \sim \mathcal{N}(0,1)$, $v_t \sim \mathcal{N}(0,10)$ and $p(x_0) = \mathcal{N}(x_0|0,5)$. We consider trajectories of 100 time steps. This benchmark is challenging for state estimation as the transition density around $x \approx 0$ is bimodal and symmetric. The scaled quadratic measurement function makes these modes indistinguishable.

We compare the results of EP with the unscented Kalman filter and smoother (van der Merwe & Wan (2004)).[2] Results are shown in Figure 1, where we plotted the root mean square error (RMSE) and the negative log-likelihood (NLL) of its predictions, all measured in latent space $x$ (see Appendix A.3 for the precise expressions of these metrics). While the RMSE only measures the quality of the posterior mean, the NLL is a measure that allows us to assess the quality of the approximate posterior mean and covariance. Observe that the results from RTS smoothing and the first iteration of EP are identical, a result we prove in Section 3.4. We see that EP iterative smoothing generally improves the results obtained by a single pass. In the next section, we develop the expectation propagation algorithm used to generate the solid red curve in Figure 1.

## 3 Expectation Propagation

After this brief motivation, we are now ready to describe expectation propagation (EP) in more detail.

In the context of dynamical systems, EP is typically derived using a factor-graph formulation (Qi, 2005; Deisenroth & Mohamed, 2012), where the (marginal) distribution over the latent state $p(\boldsymbol{x}_t|\boldsymbol{y}_{1:T})$, i.e., the smoothing distribution, is represented as a product of factors/messages $f_i$, so that $p(\boldsymbol{x}_t|\boldsymbol{y}_{1:T}) = \prod_i f_i(\boldsymbol{x}_t)$; see Figure 2 for an illustration. There are three types of factors $f_i$ in this context: forward, backward, and measurement messages, which we will denote by the symbols $\triangleright, \triangleleft, \triangle$, respectively. The forward factor $f_\triangleright$ is a message/information that is carried from all the latent states and observations prior to the current latent state $\boldsymbol{x}_t$. The backwards factor $f_\triangleleft$ sends information from future states to $\boldsymbol{x}_t$, and the measurement factor $f_\triangle$ incorporates information from the current observation. Thus, each (marginal) state posterior smoothing distribution can be written as a product of factors/messages

$$p(\boldsymbol{x}_t|\boldsymbol{y}_{1:T}) = f_\triangleright(\boldsymbol{x}_t) \quad f_\triangle(\boldsymbol{x}_t) \quad f_\triangleleft(\boldsymbol{x}_t) \tag{26}$$
$$= (\text{forward})\,(\text{measurement})\,(\text{backward}).$$

---

[2]A more thorough comparison across different filters and seeds is discussed in Section 5.

For the non-linear systems we consider here, the exact factors $f_i(\boldsymbol{x}_t)$ above and therefore the posterior distributions $p(\boldsymbol{x}_t|\boldsymbol{y}_{1:T})$ are intractable.

EP (Minka, 2001b;a) is a deterministic Bayesian inference algorithm that can be used to approximate the complex and an often intractable smoothing distribution $p(\boldsymbol{x}_t|\boldsymbol{y}_{1:T}) = \prod_i f_i(\boldsymbol{x}_t)$ by a distribution of the form $q(\boldsymbol{x}_t) = \prod_i q_i(\boldsymbol{x}_t)$, which is a product of simpler and tractable factors $q_i$. The underlying assumption is that if we can find good approximating factors $q_i$, we may get better estimates of the intractable posterior distribution $p(\boldsymbol{x}_t|\boldsymbol{y}_{1:T})$. In practice, we restrict the factors $q_i$ to families of certain distributions, such that they are closed under multiplication and division (Seeger, 2005), e.g., Gaussians. We get approximatations in this family by a procedure known as *projection*.

**Definition 1.** Let $\mathcal{F}$ be a family of probability distributions. The KL projection of $p$ onto $\mathcal{F}$ is a distribution in $\mathcal{F}$ that minimises KL $(p||q)$ over all distributions $q \in \mathcal{F}$. We use the shorthand notation $\text{proj}_{\mathcal{F}}[\cdot]$ to define the operator that projects arbitrary distributions onto the family $\mathcal{F}$:

$$\text{proj}_{\mathcal{F}}[p] := \underset{q \in \mathcal{F}}{\text{argmin}}\ \text{KL}\ (p||q). \tag{27}$$

In EP, we assume that the approximations $q(\boldsymbol{x}_t)$ and the factors $q_i(\boldsymbol{x}_t)$ come from the exponential family. This has the useful property that the operation $\text{proj}_{\mathcal{F}}[p]$ simply amounts to finding $q \in \mathcal{F}$ whose expected sufficient statistics correspond to those of $p$ (Seeger, 2005). For example, if $\mathcal{F}$ is the family of normal distributions, then $\text{proj}_{\mathcal{F}}[p] = \mathcal{N}(\boldsymbol{\mu}, \boldsymbol{\Sigma})$ where $\boldsymbol{\mu}$ and $\boldsymbol{\Sigma}$ are the mean and covariance of $p$.

## 3.1 EP Algorithm

The iterative refinement of each factor $q_i$ is carried out as follows in EP (Minka, 2001a). First, we compute the *cavity distribution*

$$q_{\backslash i}(\boldsymbol{x}_t) \propto q(\boldsymbol{x}_t)/q_i(\boldsymbol{x}_t), \tag{28}$$

which contains all the information about the distribution of $\boldsymbol{x}_t$ except for the $i^{th}$ factor, i.e. the "rest of the distribution" (Minka, 2004).

Second, we form the (unnormalised) *tilted distribution* $f_i(\boldsymbol{x}_t)q_{\backslash i}(\boldsymbol{x}_t)$ by combining the "true" factor $f_i(\boldsymbol{x}_t)$ with the cavity distribution $q_{\backslash i}(\boldsymbol{x}_t)$ and determine an updated factor $q_i(\boldsymbol{x}_t)$, so that $q_i(\boldsymbol{x}_t)q_{\backslash i}(\boldsymbol{x}_t)$ approximates $f_i(\boldsymbol{x}_t)q_{\backslash i}(\boldsymbol{x}_t)$ as close as possible in the KL sense. We achieve this via projection (Definition 1). This gives us the updated posterior $q(\boldsymbol{x}_t)^{new}$ and approximate factor $q_i(\boldsymbol{x}_t)^{\text{new}}$, which reads

$$q(\boldsymbol{x}_t)^{\text{new}} = \text{proj}_{\mathcal{F}}\left[f_i(\boldsymbol{x}_t)q_{\backslash i}(\boldsymbol{x}_t)\right] \tag{29}$$

$$q_i(\boldsymbol{x}_t)^{\text{new}} \propto \frac{q(\boldsymbol{x}_t)^{\text{new}}}{q_{\backslash i}(\boldsymbol{x}_t)}, \tag{30}$$

respectively. Note that the cavity distribution $q_{\backslash i}$ provides the "context" within which to find the updated approximate factor $q_i$; the Gaussian approximation $q_i \approx f_i$ need only hold locally, where the posterior defined by the remaining factors $q_{\backslash i}$ has the most concentration of mass.

For linear systems with Gaussian noise, the true factors $f_i(\boldsymbol{x}_t)$ are also Gaussian, hence the projection step and the implied Gaussian approximation are exact. This means that in this case, the factors $q_{\triangleright}, q_{\triangleleft}, q_{\triangle}$ are unique and optimal, i.e., $q_i = f_i$ (Minka, 2001a). The Kalman smoother obtains these results in a single forward-backward sweep through the factor graph. For non-linear systems, however, computing the exact factors $f_i$ are intractable. Hence, we use approximations to compute these factors (Qi & Minka, 2003), which requires more than a single iteration of EP to converge. If we iterate multiple times, all factors contain information from observations at all times. When correctly set up, we can at each iteration improve the quality of the factors and obtain a closer approximation to the true smoothing distribution $p(\boldsymbol{x}_t|\boldsymbol{y}_{1:T})$.

## 3.2 EP Updates for Generalised Gaussian Filtering/Smoothing

In the following, we provide explicit computational detail of the EP algorithm when the approximating family $\mathcal{F}$ is the family of Gaussians. For simplicity, we drop the subscript $\mathcal{F}$ from the notation $\text{proj}_{\mathcal{F}}[\cdot]$

---

**Algorithm 1** Gaussian EP for Dynamical Systems

---

1: **Init:** Set all factors $q_i$ to $\mathcal{N}(\mathbf{0}, \infty \boldsymbol{I})$; Set $q(\boldsymbol{x}_1) = p(\boldsymbol{x}_1)$ and marginals $q(\boldsymbol{x}_{t>1}) = \mathcal{N}(\mathbf{0}, 10^{10}\boldsymbol{I})$
2: **repeat**
3:     **for** $t = 1$ to $T$ **do**
4:         **for** all factors $q_i(\boldsymbol{x}_t)$, where $i = \triangleright, \triangle, \triangleleft$ **do**
5:             **Compute cavity distribution:**

$$q_{\backslash i}(\boldsymbol{x}_t) \propto q(\boldsymbol{x}_t)/q_i(\boldsymbol{x}_t), \tag{31}$$

6:             **Update posterior by projection:**

$$q^{\text{new}}(\boldsymbol{x}_t) = \text{proj}\left[f_i(\boldsymbol{x}_t)\, q_{\backslash i}(\boldsymbol{x}_t)\right] \tag{32}$$

7:             **Update factor:**

$$q_i^{\text{new}}(\boldsymbol{x}_t) \propto q^{\text{new}}(\boldsymbol{x}_t)/q_{\backslash i}(\boldsymbol{x}_t) \tag{33}$$

8:         **end for**
9:     **end for**
10: **until** Convergence or maximum number of iterations exceeded

---

for the projection in this case. The pseudo-code in Alg. 1 describes the main steps of Gaussian EP. This iteratively updates the marginal $q(\boldsymbol{x}_t)$ and the messages $q_i(\boldsymbol{x}_t), i \in \{\triangleright, \triangle, \triangleleft\}$ that form this marginal.

- **Compute cavity distribution:** First, the *cavity distribution* (28) $q_{\backslash i}(\boldsymbol{x}_t)$ is computed (step 5 in Alg. 1) by removing factor $q_i(\boldsymbol{x}_t)$ from the marginal $q(\boldsymbol{x}_t)$. Since $q(\boldsymbol{x}_t)$ and $q_i(\boldsymbol{x}_t)$ are Gaussian, the cavity distribution is also a (scaled) Gaussian,

$$q_{\backslash i}(\boldsymbol{x}_t) \propto q(\boldsymbol{x}_t)/q_i(\boldsymbol{x}_t) \propto \mathcal{N}(\boldsymbol{x}_t \,|\, \boldsymbol{\mu}_t^{\backslash i}, \boldsymbol{\Sigma}_t^{\backslash i}). \tag{34}$$

The terms $\boldsymbol{\mu}_t^{\backslash i}$, $\boldsymbol{\Sigma}_t^{\backslash i}$ can be computed via the Gaussian division rule (132)–(134) (Appendix A.2),

$$\boldsymbol{\Sigma}_t^{\backslash i} = (\boldsymbol{\Sigma}_t^{-1} - (\boldsymbol{\Sigma}_t^i)^{-1})^{-1}, \tag{35}$$

$$\boldsymbol{\mu}_t^{\backslash i} = \boldsymbol{\Sigma}_t^{\backslash i}(\boldsymbol{\Sigma}_t^{-1}\boldsymbol{\mu}_t - (\boldsymbol{\Sigma}_t^i)^{-1}\boldsymbol{\mu}_t^i). \tag{36}$$

- **Update posterior by projection:** Next, the Gaussian projection of the tilted distribution $f_i(\boldsymbol{x}_t)q_{\backslash i}(\boldsymbol{x}_t)$ is computed (see Def. 1). This requires one to compute the moments of $f_i(\boldsymbol{x}_t)q_{\backslash i}(\boldsymbol{x}_t)$, which can be deduced from the derivatives of the log-partition function (Minka, 2001a;b; 2008)

$$\log Z_i(\boldsymbol{\mu}_t^{\backslash i}, \boldsymbol{\Sigma}_t^{\backslash i}) = \log \int f_i(\boldsymbol{x}_t)q_{\backslash i}(\boldsymbol{x}_t)\mathrm{d}\boldsymbol{x}_t. \tag{37}$$

The moments of the updated (marginal) posterior $q^{\text{new}}(\boldsymbol{x}_t)$ can then be computed as

$$\boldsymbol{\mu}_t^{\text{new}} = \boldsymbol{\mu}_t^{\backslash i} + \boldsymbol{\Sigma}_t^{\backslash i}\boldsymbol{\nabla}_{\mu\backslash i}^{\top}, \tag{38}$$

$$\boldsymbol{\Sigma}_t^{\text{new}} = \boldsymbol{\Sigma}_t^{\backslash i} - \boldsymbol{\Sigma}_t^{\backslash i}(\boldsymbol{\nabla}_{\mu\backslash i}^{\top}\boldsymbol{\nabla}_{\mu\backslash i} - 2\boldsymbol{\nabla}_{\Sigma\backslash i})\boldsymbol{\Sigma}_t^{\backslash i}, \tag{39}$$

$$\text{where } \boldsymbol{\nabla}_{\mu\backslash i} := \mathrm{d}\log Z_i/\mathrm{d}\boldsymbol{\mu}_t^{\backslash i}, \tag{40}$$

$$\text{and } \boldsymbol{\nabla}_{\Sigma\backslash i} := \mathrm{d}\log Z_i/\mathrm{d}\boldsymbol{\Sigma}_t^{\backslash i}. \tag{41}$$

- **Update factor:** Finally, we obtain the updated factor $q_i^{\text{new}}$ (see (30)) again via the Gaussian division rule, since both the updated marginal and the cavity distribution are (unnormalised) Gaussians.

*Remark* 1. In some cases, it is easier to compute the updated factor $q_i^{\text{new}}$ first; the updated posterior is then obtained by multiplying the updated factor with the cavity distribution: $q^{\text{new}} \propto q_i^{\text{new}} q_{\setminus i}$.

*Remark* 2. While the focus of the paper is on smoothing, i.e., the offline inference setting, message passing in the online setting via a moving window (fixed-lag smoothing) could also be implemented. Here, we would use the messages of the last $k$ time steps to find an approximate filtering distribution $p(\boldsymbol{x}_t | \boldsymbol{y}_{t-k:t})$. Iterations within the considered window would still give improved filtering distributions.

### 3.3 Implicit Linearisation Requires Explicit Consideration

We now highlight an important detail in the computation of the derivatives $\boldsymbol{\nabla}_{\mu\setminus i}, \boldsymbol{\nabla}_{\Sigma\setminus i}$, which are required in the EP update formulas (38)–(39). For nonlinear systems, these cannot be computed in closed-form, hence we use approximate derivatives in practice, which is obtained by replacing the partition function $Z_i$ by its approximation, denoted $\tilde{Z}_i$, in the computation of (40)–(41). As we shall see more concretely in the next section, this approximation in general takes a Gaussian form, making it possible to compute derivatives.

As an illustrative example, consider the partition function $Z_\triangle = \int f_\triangle(\boldsymbol{x}_t) q^{\setminus \triangle}(\boldsymbol{x}_t) \mathrm{d}\boldsymbol{x}_t$ corresponding to the measurement message, where $f_\triangle(\boldsymbol{x}_t) = p(\boldsymbol{y}_t | \boldsymbol{x}_t) = \mathcal{N}(\boldsymbol{y}_t | \boldsymbol{h}(\boldsymbol{x}_t), \boldsymbol{R})$. Approximating the tilted distribution $p(\boldsymbol{y}_t | \boldsymbol{x}_t) q^{\setminus \triangle}(\boldsymbol{x}_t)$ by a joint Gaussian in $\boldsymbol{x}_t$ and $\boldsymbol{y}_t$, we get the approximation

$$\tilde{Z}_\triangle = \int \mathcal{N}\left(\begin{pmatrix} \boldsymbol{x}_t \\ \boldsymbol{y}_t \end{pmatrix} \Big| \begin{pmatrix} \boldsymbol{\mu}_t^{x\setminus\triangle} \\ \boldsymbol{\mu}_t^{y\setminus\triangle} \end{pmatrix}, \begin{pmatrix} \boldsymbol{\Sigma}_t^{x\setminus\triangle} & \boldsymbol{\Sigma}_t^{xy\setminus\triangle} \\ \boldsymbol{\Sigma}_t^{yx\setminus\triangle} & \boldsymbol{\Sigma}_t^{y\setminus\triangle} \end{pmatrix}\right) \mathrm{d}\boldsymbol{x}_t = \mathcal{N}(\boldsymbol{y}_t | \boldsymbol{\mu}_t^{y\setminus\triangle}, \boldsymbol{\Sigma}_t^{y\setminus\triangle}), \tag{42}$$

which takes the form of a Gaussian. Here, $\boldsymbol{\mu}_t^{y\setminus\triangle}, \boldsymbol{\Sigma}_t^{y\setminus\triangle}, \boldsymbol{\Sigma}_t^{xy\setminus\triangle}$ correspond to approximations of the integrals $\mathbb{E}_{q^{\setminus\triangle}}[\boldsymbol{h}(\boldsymbol{x}_t)], \mathrm{cov}_{q^{\setminus\triangle}}[\boldsymbol{h}(\boldsymbol{x}_t)]$ and $\mathrm{cov}_{q^{\setminus\triangle}}[\boldsymbol{x}_t, \boldsymbol{h}(\boldsymbol{x}_t)]$ respectively, which can be obtained using the same techniques as we saw in the examples in Section 2.2, by applying the unscented transform, linearisation etc. The terms $\boldsymbol{\mu}_t^{x\setminus\triangle}$ and $\boldsymbol{\Sigma}_t^{x\setminus\triangle}$ are equivalent to $\boldsymbol{\mu}_t^{\setminus\triangle}$ and $\boldsymbol{\Sigma}_t^{\setminus\triangle}$; the superscripts were simply added to distinguish them from moments in the observed space.

Note that $\tilde{Z}_\triangle$ in (42) is a Gaussian in the observed space $\boldsymbol{y}_t$, while the derivatives (40)–(41) are taken with respect to moments in the latent space $\boldsymbol{x}_t$. Using the chain rule, the derivatives thus read

$$\boldsymbol{\nabla}_{\mu\setminus\triangle} := \frac{\mathrm{d}\log\tilde{Z}_i}{\mathrm{d}\boldsymbol{\mu}_t^{x\setminus\triangle}} = \frac{\partial\log\tilde{Z}_\triangle}{\partial\boldsymbol{\mu}_t^{y\setminus\triangle}} \textcolor{red}{\frac{\partial\boldsymbol{\mu}_t^{y\setminus\triangle}}{\partial\boldsymbol{\mu}_t^{x\setminus\triangle}}} + \frac{\partial\log\tilde{Z}_\triangle}{\partial\boldsymbol{\Sigma}_t^{y\setminus\triangle}} \textcolor{red}{\frac{\partial\boldsymbol{\Sigma}_t^{y\setminus\triangle}}{\partial\boldsymbol{\mu}_t^{x\setminus\triangle}}}, \tag{43}$$

$$\boldsymbol{\nabla}_{\Sigma\setminus\triangle} := \frac{\mathrm{d}\log\tilde{Z}_\triangle}{\mathrm{d}\boldsymbol{\Sigma}_t^{x\setminus\triangle}} = \frac{\partial\log\tilde{Z}_\triangle}{\partial\boldsymbol{\mu}_t^{y\setminus\triangle}} \textcolor{red}{\frac{\partial\boldsymbol{\mu}_t^{y\setminus\triangle}}{\partial\boldsymbol{\Sigma}_t^{x\setminus\triangle}}} + \frac{\partial\log\tilde{Z}_\triangle}{\partial\boldsymbol{\Sigma}_t^{y\setminus\triangle}} \textcolor{red}{\frac{\partial\boldsymbol{\Sigma}_t^{y\setminus\triangle}}{\partial\boldsymbol{\Sigma}_t^{x\setminus\triangle}}}, \tag{44}$$

which requires knowledge of the derivatives of moments in the observed space with respect to moments in the latent space (terms highlighted in red). Crucially, these do not necessarily vanish when $\boldsymbol{x}_t$ and $\boldsymbol{y}_t$ have non-zero correlation—we expect $(\boldsymbol{\mu}_t^{y\setminus\triangle}, \boldsymbol{\Sigma}_t^{y\setminus\triangle})$ to change with $(\boldsymbol{\mu}_t^{x\setminus\triangle}, \boldsymbol{\Sigma}_t^{x\setminus\triangle})$ if they are correlated.

Computing the terms in red can be achieved if we have an explicit relation between the random variables $\boldsymbol{x}_t$ and $\boldsymbol{y}_t$. Since $\boldsymbol{x}_t$ and $\boldsymbol{y}_t$ are jointly Gaussian in the approximation we made to our partition function (42), then by Lemma 1, they are related by an affine transformation up to a zero-mean Gaussian noise, that is, $\boldsymbol{y}_t = \boldsymbol{J}_t\boldsymbol{x}_t + \boldsymbol{u}_t + \boldsymbol{\eta}_t$ for some matrix $\boldsymbol{J}_t \in \mathbb{R}^{E\times D}$, vector $\boldsymbol{u}_t \in \mathbb{R}^E$, and $\boldsymbol{\eta}_t \sim \mathcal{N}(\boldsymbol{0}, \boldsymbol{R})$. Thus, we have

$$\boldsymbol{\mu}_t^{y\setminus\triangle} = \boldsymbol{J}_t\boldsymbol{\mu}_t^{x\setminus\triangle} + \boldsymbol{u}_t, \tag{45}$$

$$\boldsymbol{\Sigma}_t^{y\setminus\triangle} = \boldsymbol{J}_t\boldsymbol{\Sigma}_t^{x\setminus i}\boldsymbol{J}_t^\top + \boldsymbol{R}. \tag{46}$$

We now make an important assumption that the matrix/vectors $\boldsymbol{J}_t, \boldsymbol{u}_t, \boldsymbol{R}$ do not depend on the moments $\boldsymbol{\mu}_t^{x\setminus\triangle}, \boldsymbol{\Sigma}_t^{x\setminus\triangle}$, which is equivalent to saying that the approximate conditional distribution $\tilde{p}(\boldsymbol{y}_t | \boldsymbol{x}_t) = \mathcal{N}(\boldsymbol{y}_t | \boldsymbol{J}_t\boldsymbol{x}_t + \boldsymbol{u}_t, \boldsymbol{R})$ is independent of the moments $\boldsymbol{\mu}_t^{x\setminus\triangle}, \boldsymbol{\Sigma}_t^{x\setminus\triangle}$. This is a reasonable assumption since the likelihood $p(\boldsymbol{y}_t | \boldsymbol{x}_t)$ of our original, unapproximated system is independent of the prior $p(\boldsymbol{x}_t)$.

With this assumption, using (45)–(46), we get

$$\frac{\partial \boldsymbol{\mu}_t^{y\backslash\Delta}}{\partial \boldsymbol{\mu}_t^{x\backslash\Delta}} = \boldsymbol{J}_t \in \mathbb{R}^{E\times D}, \quad \frac{\partial \boldsymbol{\mu}_t^{y\backslash\Delta}}{\partial \boldsymbol{\Sigma}_t^{x\backslash\Delta}} = \boldsymbol{0} \in \mathbb{R}^{E\times D\times D}, \quad \frac{\partial \boldsymbol{\Sigma}_t^{y\backslash\Delta}}{\partial \boldsymbol{\mu}_t^{x\backslash\Delta}} = \boldsymbol{0} \in \mathbb{R}^{E\times E\times D}, \quad \frac{\partial [\boldsymbol{\Sigma}_t^{y\backslash\Delta}]_{ij}}{\partial [\boldsymbol{\Sigma}_t^{x\backslash\Delta}]_{kl}} = [\boldsymbol{J}_t]_{ik}[\boldsymbol{J}_t]_{jl}. \tag{47}$$

Thus, from (43)–(44), we obtain the following closed-form formulas for the derivatives in question

$$\boldsymbol{\nabla}_{\mu\backslash\Delta} = (\boldsymbol{y}_t - \boldsymbol{\mu}_t^{y\backslash\Delta})^\top (\boldsymbol{\Sigma}_t^{y\backslash\Delta})^{-1} \boldsymbol{J}_t \in \mathbb{R}^{1\times D}, \tag{48}$$

$$\boldsymbol{\nabla}_{\Sigma\backslash\Delta} = \frac{1}{2}\left(\boldsymbol{\nabla}_{\mu\backslash\Delta}^\top \boldsymbol{\nabla}_{\mu\backslash\Delta} - \boldsymbol{J}_t^\top (\boldsymbol{\Sigma}_t^{y\backslash\Delta})^{-1} \boldsymbol{J}_t\right) \in \mathbb{R}^{D\times D}, \tag{49}$$

where we used the identities (126)–(127) in Appendix A.1 to compute the derivatives of a log-Gaussian.

### 3.4 Derivative-based EP Updates and Kalman Smoothers

Section 2.1 provides generic equations for Bayesian filtering (7)–(8) and RTS smoothing (14)–(15) in non-linear systems with Gaussian state distributions; the only requirement is to be able to compute means and covariances of two joint distributions to apply these update equations. In Section 3.2, based on derivatives of the log-partition function, we derived generic update equations (38)–(39) for computing means and covariances of posterior distributions of the latent states of a non-linear dynamical system, but now formulated as a message-passing algorithm, which can potentially be iterated and may lead to more refined posterior distributions than RTS smoothing.

In the following, we will show that these generic update equations are identical for a single iteration of EP when updates are performed in the following specific order: As in Gaussian filtering, we start with the time update or "forward factor" ($\triangleright$) update, followed by a measurement update or "up factor" ($\triangle$) update and repeat until we reach $t = T$. Then we move backwards until we reach $t = 0$ by updating the "back factor" ($\triangleleft$) similar to the backward pass of an RTS smoother. The implication of this statement is that we can interpret the EP computations in the context of classical Gaussian filtering and RTS smoothing for non-linear dynamical systems. These equivalences include all classical filters and RTS smoothers for linear and non-linear systems, including the extended/unscented/cubature Kalman filters/smoothers.

**Forward message** To compute the mean and covariance for the forward message $f_\triangleright(\boldsymbol{x}_t)$, we obtain it directly instead of using the derivative-based update formulas (38)–(39), as this turns out to be much simpler. Note that the approximate forward message has the expression

$$f_\triangleright(\boldsymbol{x}_t) = \int p(\boldsymbol{x}_t|\boldsymbol{x}_{t-1})\, q_{\backslash\triangleleft}(\boldsymbol{x}_{t-1})\mathrm{d}\boldsymbol{x}_{t-1} = \int \mathcal{N}(\boldsymbol{x}_t \,|\, \boldsymbol{f}(\boldsymbol{x}_{t-1}), \boldsymbol{Q})\, q_{\backslash\triangleleft}(\boldsymbol{x}_{t-1})\mathrm{d}\boldsymbol{x}_{t-1}, \tag{50}$$

which we can see from the factor graphs in Figure 2. Due to the non-linearity of $\boldsymbol{f}$, the above integral is intractable. Hence, we approximate the integrand by a joint Gaussian, denoted $q(\boldsymbol{x}_{t-1}, \boldsymbol{x}_t)$, using techniques such as linearisation and unscented transform (see examples in Section 2.2 for details). By Lemma 1, we then obtain the Gaussian approximation

$$q_\triangleright(\boldsymbol{x}_t) = \int q(\boldsymbol{x}_{t-1}, \boldsymbol{x}_t)\mathrm{d}\boldsymbol{x}_{t-1} = \mathcal{N}(\boldsymbol{x}_t \,|\, \boldsymbol{M}_{t-1}\boldsymbol{\mu}_{t-1}^{\backslash\triangleleft} + \boldsymbol{v}_{t-1}, \boldsymbol{M}_{t-1}\boldsymbol{\Sigma}_{t-1}^{\backslash\triangleleft}\boldsymbol{M}_{t-1}^\top + \boldsymbol{P}_{t-1}), \tag{51}$$

to the true factor $f_\triangleright(\boldsymbol{x}_t)$ for some $\boldsymbol{M}_{t-1}, \boldsymbol{v}_{t-1}, \boldsymbol{P}_{t-1}$, where we used (18) and the Gaussian marginalisation rule in the last equality. Now, since $\boldsymbol{P}_{t-1}$ is the covariance of the conditional distribution $p(\boldsymbol{x}_t|\boldsymbol{x}_{t-1})$ (see proof of Lemma 1), we have $\boldsymbol{P}_{t-1} = \boldsymbol{Q}$. This results in the update formulas

$$\boldsymbol{\mu}_t^\triangleright = \boldsymbol{M}_{t-1}\boldsymbol{\mu}_{t-1}^{\backslash\triangleleft} + \boldsymbol{v}_{t-1}, \tag{52}$$

$$\boldsymbol{\Sigma}_t^\triangleright = \boldsymbol{M}_{t-1}\boldsymbol{\Sigma}_{t-1}^{\backslash\triangleleft}\boldsymbol{M}_{t-1}^\top + \boldsymbol{Q}, \tag{53}$$

for the forward message. In accordance with Remark 1, we can then also update the approximate marginal posterior $q(\boldsymbol{x}_t)$ by a simple Gaussian multiplication, i.e., $q(\boldsymbol{x}_t) \propto q_\triangleright(\boldsymbol{x}_t)q_{\backslash\triangleright}(\boldsymbol{x}_t)$.

Note that the approach that we use here to obtain the forward update formula (i.e., form a joint Gaussian approximation to the integrand of (50) and marginalise) is identical to how we obtain a general time update in standard Bayesian filtering (see Section 2.1). The only difference is that in Bayesian filtering, the term corresponding to the cavity distribution $q_{\backslash \lhd}(\boldsymbol{x}_{t-1})$ in (50) is given by the filtering distribution $p(\boldsymbol{x}_{t-1}|\boldsymbol{y}_{1:t-1})$ (see (3) in Section 2.1). However, if we refer to Table 1 for the correspondence between EP messages and filtering distributions, we see that in the very first iteration of EP, the cavity distribution $q_{\backslash \lhd}(\boldsymbol{x}_{t-1})$ is exactly equal to $p(\boldsymbol{x}_{t-1}|\boldsymbol{y}_{1:t-1})$. Hence, we have the following result.

**Theorem 1.** *In the very first iteration of EP, computing the approximate forward message $q_{\rhd}(\boldsymbol{x}_t)$ via the moments of $f_{\rhd}(\boldsymbol{x}_t)$ in (50) corresponds to the time update in a classical Bayesian filtering algorithm, where the moments are given by (9)–(10).*

**Measurement message** For the measurement message, we employ the derivative-based update formulas (38)–(39). We have already explained how to compute the (approximate) derivatives $\boldsymbol{\nabla}_{\mu\backslash\Delta}$ and $\boldsymbol{\nabla}_{\Sigma\backslash\Delta}$ for this case in Section 3.3, where the resulting expressions are given by (48)–(49). Using this, we can then obtain the updated posterior $q$ via (38)–(39) and an updated factor $q_\Delta$ by Gaussian division; see (30).

**Theorem 2.** *The moments*

$$\boldsymbol{\mu}_t = \boldsymbol{\mu}_t^{x\backslash\Delta} + \boldsymbol{\Sigma}_t^{x\backslash\Delta} \boldsymbol{\nabla}_{\mu\backslash\Delta}^\top, \tag{54}$$

$$\boldsymbol{\Sigma}_t = \boldsymbol{\Sigma}_t^{x\backslash\Delta} - \boldsymbol{\Sigma}_t^{x\backslash\Delta} \left[ \boldsymbol{\nabla}_{\mu\backslash\Delta}^\top \boldsymbol{\nabla}_{\mu\backslash\Delta} - 2\boldsymbol{\nabla}_{\Sigma\backslash\Delta} \right] \boldsymbol{\Sigma}_t^{x\backslash\Delta} \tag{55}$$

*of the update posterior $q(\boldsymbol{x}_t)$, which are computed via the derivatives of $\log \tilde{Z}_\Delta$, are identical to the moments (7)–(8) of the filtering distribution (measurement update)*

$$\boldsymbol{\mu}_{t|t}^x = \boldsymbol{\mu}_{t|t-1}^x + \boldsymbol{K}_t(\boldsymbol{y}_t - \boldsymbol{\mu}_{t|t-1}^y), \tag{56}$$

$$\boldsymbol{\Sigma}_{t|t}^x = \boldsymbol{\Sigma}_{t|t-1}^x - \boldsymbol{K}_t \boldsymbol{\Sigma}_{t|t-1}^{yx} \tag{57}$$

*obtained by any Gaussian filter, where*

$$\boldsymbol{K}_t := \boldsymbol{\Sigma}_t^{xy}(\boldsymbol{\Sigma}_t^y)^{-1} \tag{58}$$

*is the "Kalman gain".*

*Proof.* Since we are applying the EP forward and measurement updates in sequence and recalling the fact that we initialised the backward messages $q_{\lhd}(\boldsymbol{x}_t)$ with an uninformative prior, we see that

$$q_{\backslash\Delta}(\boldsymbol{x}_t) = q_{\rhd}(\boldsymbol{x}_t)q_{\lhd}(\boldsymbol{x}_t) = \mathcal{N}(\boldsymbol{x}_t \,|\, \boldsymbol{\mu}_{t|t-1}^x, \boldsymbol{\Sigma}_{t|t-1}^x)\mathcal{N}(\boldsymbol{x}_t \,|\, \boldsymbol{0}, \infty \boldsymbol{I}) = \mathcal{N}(\boldsymbol{x}_t \,|\, \boldsymbol{\mu}_{t|t-1}^x, \boldsymbol{\Sigma}_{t|t-1}^x), \tag{59}$$

where we used the relation in Table 1 to identify the forward message $q_{\rhd}(\boldsymbol{x}_t)$ with the predictive distribution $p(\boldsymbol{x}_t|\boldsymbol{y}_{1:t-1}) \approx \mathcal{N}(\boldsymbol{x}_t \,|\, \boldsymbol{\mu}_{t|t-1}^x, \boldsymbol{\Sigma}_{t|t-1}^x)$. This gives us $\boldsymbol{\mu}_t^{x\backslash\Delta} = \boldsymbol{\mu}_{t|t-1}^x$ and $\boldsymbol{\Sigma}_t^{x\backslash\Delta} = \boldsymbol{\Sigma}_{t|t-1}^x$, which allows us to re-express the standard Kalman update (56)–(57) in terms of EP messages, as

$$\boldsymbol{\mu}_{t|t}^x = \boldsymbol{\mu}_t^{x\backslash\Delta} + \boldsymbol{K}_t(\boldsymbol{y}_t - \boldsymbol{\mu}_t^{y\backslash\Delta}), \tag{60}$$

$$\boldsymbol{\Sigma}_{t|t}^x = \boldsymbol{\Sigma}_t^{x\backslash\Delta} - \boldsymbol{K}_t \boldsymbol{\Sigma}_t^{yx\backslash\Delta}, \tag{61}$$

where $\boldsymbol{K}_t = \boldsymbol{\Sigma}_t^{xy\backslash\Delta}(\boldsymbol{\Sigma}_t^{y\backslash\Delta})^{-1}$. We start by showing that the expressions for the means in (54) and (60) are identical, which is true if

$$\boldsymbol{\Sigma}_t^{x\backslash\Delta} \boldsymbol{\nabla}_{\mu\backslash\Delta}^\top = \boldsymbol{K}_t(\boldsymbol{y}_t - \boldsymbol{\mu}_t^{y\backslash\Delta}) \tag{62}$$

holds. To see this, we use (19) and (48) to re-express the left-hand side of (62) as

$$\boldsymbol{\Sigma}_t^{x\backslash\Delta} \boldsymbol{\nabla}_{\mu\backslash\Delta}^\top \overset{(48)}{=} \boldsymbol{\Sigma}_t^{x\backslash\Delta} \boldsymbol{J}_t^\top (\boldsymbol{\Sigma}_t^{y\backslash\Delta})^{-1}(\boldsymbol{y}_t - \boldsymbol{\mu}_t^{y\backslash\Delta}) \overset{(19)}{=} \boldsymbol{\Sigma}_t^{x\backslash\Delta}(\boldsymbol{\Sigma}_t^{x\backslash\Delta})^{-1}\boldsymbol{\Sigma}_t^{xy\backslash\Delta}(\boldsymbol{\Sigma}_t^{y\backslash\Delta})^{-1}(\boldsymbol{y}_t - \boldsymbol{\mu}_t^{y\backslash\Delta}) \tag{63}$$

$$= \underbrace{\boldsymbol{\Sigma}_t^{x\backslash\Delta}(\boldsymbol{\Sigma}_t^{x\backslash\Delta})^{-1}}_{=\boldsymbol{I}}\boldsymbol{\Sigma}_t^{xy\backslash\Delta}(\boldsymbol{\Sigma}_t^{y\backslash\Delta})^{-1}(\boldsymbol{y}_t - \boldsymbol{\mu}_t^{y\backslash\Delta}) = \underbrace{\boldsymbol{\Sigma}_t^{xy\backslash\Delta}(\boldsymbol{\Sigma}_t^{y\backslash\Delta})^{-1}}_{=\boldsymbol{K}_t}(\boldsymbol{y}_t - \boldsymbol{\mu}_t^{y\backslash\Delta}), \tag{64}$$

| Filtering/smoothing distributions | Corresponding EP messages |
|---|---|
| Predictive distribution (time update) $p(\boldsymbol{x}_t|\boldsymbol{y}_{1:t-1})$ | $q_\triangleright(\boldsymbol{x}_t)$ |
| Filtering distribution (measurement update) $p(\boldsymbol{x}_t|\boldsymbol{y}_{1:t})$ | $q_\triangleright(\boldsymbol{x}_t)q_\triangle(\boldsymbol{x}_t) = q_{\setminus\triangleleft}(\boldsymbol{x}_t)$ |
| Smoothing distribution $p(\boldsymbol{x}_t|\boldsymbol{y}_{1:T})$ | $q_\triangleright(\boldsymbol{x}_t)q_\triangle(\boldsymbol{x}_t)q_\triangleleft(\boldsymbol{x}_t) = q(\boldsymbol{x}_t)$ |

Table 1: Glossary of filtering/smoothing distributions and corresponding EP messages during the first iteration of EP. These equivalences can be verified inductively by comparing how they are both computed. Note that all posteriors are only approximate (Gaussian) in the non-linear case.

which verifies identity (62), thus concluding the proof for the mean.

To show the corresponding identity for the covariances in (55) and (61), we need to show that

$$\boldsymbol{\Sigma}_t^{x\setminus\triangle}\Big[\boldsymbol{\nabla}_{\mu\setminus\triangle}^\top\boldsymbol{\nabla}_{\mu\setminus\triangle} - 2\boldsymbol{\nabla}_{\Sigma\setminus\triangle}\Big]\boldsymbol{\Sigma}_t^{x\setminus\triangle} = \boldsymbol{K}_t\boldsymbol{\Sigma}_t^{yx\setminus\triangle} \tag{65}$$

holds. Using (19) and (49), the left-hand side of (65) becomes

$$\boldsymbol{\Sigma}_t^{x\setminus\triangle}\Big[\boldsymbol{\nabla}_{\mu\setminus\triangle}^\top\boldsymbol{\nabla}_{\mu\setminus\triangle} - 2\boldsymbol{\nabla}_{\Sigma\setminus\triangle}\Big]\boldsymbol{\Sigma}_t^{x\setminus\triangle} \overset{(49)}{=} \boldsymbol{\Sigma}_t^{x\setminus\triangle}\boldsymbol{J}_t^\top(\boldsymbol{\Sigma}_t^{y\setminus\triangle})^{-1}\boldsymbol{J}_t\boldsymbol{\Sigma}_t^{x\setminus\triangle} \tag{66}$$

$$\overset{(19)}{=} \boldsymbol{\Sigma}_t^{x\setminus\triangle}(\boldsymbol{\Sigma}_t^{x\setminus\triangle})^{-1}\boldsymbol{\Sigma}_t^{xy\setminus\triangle}(\boldsymbol{\Sigma}_t^{y\setminus\triangle})^{-1}\boldsymbol{\Sigma}_t^{yx\setminus\triangle}(\boldsymbol{\Sigma}_t^{x\setminus\triangle})^{-1}\boldsymbol{\Sigma}_t^{x\setminus\triangle} \tag{67}$$

$$= \underbrace{\boldsymbol{\Sigma}_t^{xy\setminus\triangle}(\boldsymbol{\Sigma}_t^{y\setminus\triangle})^{-1}}_{=\boldsymbol{K}_t}\boldsymbol{\Sigma}_t^{yx\setminus\triangle}, \tag{68}$$

concluding the proof of Theorem 2.

$\square$

Theorems 1–2 state that computing the Gaussian approximate posterior state distributions with derivative-based updates of the forward and measurement messages is equivalent to the time and measurement updates in standard Gaussian filtering for non-linear systems.

**Backward message**   For the backward message, we use the partition function

$$Z_\triangleleft(\boldsymbol{\mu}_t^{\setminus\triangleleft}, \boldsymbol{\Sigma}_t^{\setminus\triangleleft}) = \int f_\triangleleft(\boldsymbol{x}_t)q_{\setminus\triangleleft}(\boldsymbol{x}_t)\mathrm{d}\boldsymbol{x}_t \propto \int f_\triangleleft(\boldsymbol{x}_t)\mathcal{N}(\boldsymbol{x}_t\,|\,\boldsymbol{\mu}_t^{\setminus\triangleleft}, \boldsymbol{\Sigma}_t^{\setminus\triangleleft})\mathrm{d}\boldsymbol{x}_t, \tag{69}$$

where

$$f_\triangleleft(\boldsymbol{x}_t) = \int p(\boldsymbol{x}_{t+1}|\boldsymbol{x}_t)q_{\setminus\triangleright}(\boldsymbol{x}_{t+1})\mathrm{d}\boldsymbol{x}_{t+1}, \tag{70}$$

to update the backward message $q_\triangleleft(\boldsymbol{x}_t)$ and the posterior distribution $p(\boldsymbol{x}_t|\boldsymbol{y}_{1:T})$ using the derivative-based update formulas (38)–(39). The true factor $f_\triangleleft(\boldsymbol{x}_t)$ in (70) accounts for the coupling between $\boldsymbol{x}_t$ and $\boldsymbol{x}_{t+1}$. Substituting the expression for $f_\triangleleft(\boldsymbol{x}_t)$ in (70) into (69) and reordering the integration yields

$$Z_\triangleleft \propto \int q_{\setminus\triangleright}(\boldsymbol{x}_{t+1})\underbrace{\int p(\boldsymbol{x}_{t+1}|\boldsymbol{x}_t)q_{\setminus\triangleleft}(\boldsymbol{x}_t)\mathrm{d}\boldsymbol{x}_t}_{=f_\triangleright(\boldsymbol{x}_{t+1})}\mathrm{d}\boldsymbol{x}_{t+1}. \tag{71}$$

The inner integral in (71) corresponds to the forward message (50) at time step $t+1$, which can be approximated by $q_\triangleright(\boldsymbol{x}_{t+1}) \propto \mathcal{N}(\boldsymbol{x}_{t+1}\,|\,\boldsymbol{\mu}_{t+1}^\triangleright, \boldsymbol{\Sigma}_{t+1}^\triangleright)$, as discussed earlier. Then, using the identity (131) in Appendix A.2, the integral in (71) can be computed analytically, giving us the following Gaussian approximation

$$\tilde{Z}_\triangleleft = \mathcal{N}(\boldsymbol{\mu}_{t+1}^{\setminus\triangleright}\,|\,\boldsymbol{\mu}_{t+1}^\triangleright, \boldsymbol{\Sigma}_{t+1}^\triangleright + \boldsymbol{\Sigma}_{t+1}^{\setminus\triangleright}). \tag{72}$$

We can now use a similar trick as before to compute the derivatives $\boldsymbol{\nabla}_{\mu\backslash\vartriangleleft}, \boldsymbol{\nabla}_{\Sigma\backslash\vartriangleleft}$ in the backward message updates. Recall that when obtaining the approximate forward message $q_\vartriangleright(\boldsymbol{x}_{t+1}) \approx f_\vartriangleright(\boldsymbol{x}_{t+1})$, we have approximated the distribution $p(\boldsymbol{x}_{t+1}|\boldsymbol{x}_t)q_{\backslash\vartriangleleft}(\boldsymbol{x}_t)$ by a joint Gaussian $q(\boldsymbol{x}_t, \boldsymbol{x}_{t+1})$, giving us the relation $\boldsymbol{x}_{t+1} = \boldsymbol{M}_t\boldsymbol{x}_t + \boldsymbol{v}_t + \boldsymbol{\varepsilon}_t$ for some matrix $\boldsymbol{M}_t$, vector $\boldsymbol{v}_t$, and $\boldsymbol{\varepsilon}_t \sim \mathcal{N}(\boldsymbol{0}, \boldsymbol{Q})$ by Lemma 1. Then, assuming as before that $\boldsymbol{M}_t, \boldsymbol{v}_t, \boldsymbol{Q}$ are independent of $(\boldsymbol{\mu}_t^{\backslash\vartriangleleft}, \boldsymbol{\Sigma}_t^{\backslash\vartriangleleft})^3$, we deduce from (52)–(53) that

$$\frac{\partial \boldsymbol{\mu}_{t+1}^{\vartriangleright}}{\partial \boldsymbol{\mu}_t^{\backslash\vartriangleleft}} = \boldsymbol{M}_t, \quad \frac{\partial \boldsymbol{\mu}_{t+1}^{\vartriangleright}}{\partial \boldsymbol{\Sigma}_t^{\backslash\vartriangleleft}} = \boldsymbol{0}, \quad \frac{\partial \boldsymbol{\Sigma}_{t+1}^{\vartriangleright}}{\partial \boldsymbol{\mu}_t^{\backslash\vartriangleleft}} = \boldsymbol{0}, \quad \frac{\partial [\boldsymbol{\Sigma}_{t+1}^{\vartriangleright}]_{ij}}{\partial [\boldsymbol{\Sigma}_t^{\backslash\vartriangleleft}]_{kl}} = [\boldsymbol{M}_t]_{ik}[\boldsymbol{M}_t]_{jl}, \tag{73}$$

giving us

$$\boldsymbol{\nabla}_{\mu\backslash\vartriangleleft} := \frac{\mathrm{d}\log\tilde{Z}_\vartriangleleft}{\mathrm{d}\boldsymbol{\mu}_t^{\backslash\vartriangleleft}} = \frac{\partial\log\tilde{Z}_\vartriangleleft}{\partial\boldsymbol{\mu}_{t+1}^{\vartriangleright}}\frac{\partial\boldsymbol{\mu}_{t+1}^{\vartriangleright}}{\partial\boldsymbol{\mu}_t^{\backslash\vartriangleleft}} + \frac{\partial\log\tilde{Z}_\vartriangleleft}{\partial\boldsymbol{\Sigma}_{t+1}^{\vartriangleright}}\frac{\partial\boldsymbol{\Sigma}_{t+1}^{\vartriangleright}}{\partial\boldsymbol{\mu}_t^{\backslash\vartriangleleft}} = (\boldsymbol{\mu}_{t+1}^{\backslash\vartriangleright} - \boldsymbol{\mu}_{t+1}^{\vartriangleright})^\top(\boldsymbol{\Sigma}_{t+1}^{\vartriangleright} + \boldsymbol{\Sigma}_{t+1}^{\backslash\vartriangleright})^{-1}\boldsymbol{M}_t, \tag{74}$$

$$\boldsymbol{\nabla}_{\Sigma\backslash\vartriangleleft} := \frac{\mathrm{d}\log\tilde{Z}_\vartriangleleft}{\mathrm{d}\boldsymbol{\Sigma}_t^{\backslash\vartriangleleft}} = \frac{\partial\log\tilde{Z}_\vartriangleleft}{\partial\boldsymbol{\mu}_{t+1}^{\vartriangleright}}\frac{\partial\boldsymbol{\mu}_{t+1}^{\vartriangleright}}{\partial\boldsymbol{\Sigma}_t^{\backslash\vartriangleleft}} + \frac{\partial\log\tilde{Z}_\vartriangleleft}{\partial\boldsymbol{\Sigma}_{t+1}^{\vartriangleright}}\frac{\partial\boldsymbol{\Sigma}_{t+1}^{\vartriangleright}}{\partial\boldsymbol{\Sigma}_t^{\backslash\vartriangleleft}} = \frac{\boldsymbol{\nabla}_{\mu\backslash\vartriangleleft}^\top\boldsymbol{\nabla}_{\mu\backslash\vartriangleleft} - \boldsymbol{M}_t^\top(\boldsymbol{\Sigma}_{t+1}^{\vartriangleright} + \boldsymbol{\Sigma}_{t+1}^{\backslash\vartriangleright})^{-1}\boldsymbol{M}_t}{2}, \tag{75}$$

where again, we used the identities (126)–(127) in Appendix A.1 to compute the derivatives of $\log\tilde{Z}_\vartriangleleft$.

**Theorem 3.** *The backward message updates via the derivatives of* $\log\tilde{Z}_\vartriangleleft$

$$\boldsymbol{\mu}_t = \boldsymbol{\mu}_t^{\backslash\vartriangleleft} + \boldsymbol{\Sigma}_t^{\backslash\vartriangleleft}\boldsymbol{\nabla}_{\mu\backslash\vartriangleleft}^\top, \tag{76}$$

$$\boldsymbol{\Sigma}_t = \boldsymbol{\Sigma}_t^{\backslash\vartriangleleft} - \boldsymbol{\Sigma}_t^{\backslash\vartriangleleft}\Big[\boldsymbol{\nabla}_{\mu\backslash\vartriangleleft}^\top\boldsymbol{\nabla}_{\mu\backslash\vartriangleleft} - 2\boldsymbol{\nabla}_{\Sigma\backslash\vartriangleleft}\Big]\boldsymbol{\Sigma}_t^{\backslash\vartriangleleft}, \tag{77}$$

*are identical to those obtained by the RTS smoothing update*

$$\boldsymbol{\mu}_{t|T}^x = \boldsymbol{\mu}_{t|t}^x + \boldsymbol{L}_t(\boldsymbol{\mu}_{t+1|T}^x - \boldsymbol{\mu}_{t+1|t}^x), \tag{78}$$

$$\boldsymbol{\Sigma}_{t|T}^x = \boldsymbol{\Sigma}_{t|t}^x + \boldsymbol{L}_t\left(\boldsymbol{\Sigma}_{t+1|T}^x - \boldsymbol{\Sigma}_{t+1|t}^x\right)\boldsymbol{L}_t^\top, \tag{79}$$

*where*

$$\boldsymbol{L}_t := \boldsymbol{\Sigma}_{t,t+1|t}^x\left(\boldsymbol{\Sigma}_{t+1|t}^x\right)^{-1}. \tag{80}$$

To prove this, we use the following auxiliary result.

**Lemma 2.** For any $t$, the following identity holds

$$(\boldsymbol{\Sigma}_t^{\backslash\vartriangleright})^{-1}(\boldsymbol{\mu}_t^{\backslash\vartriangleright} - \boldsymbol{\mu}_t^{\vartriangleright}) = \boldsymbol{\Sigma}_t^{-1}(\boldsymbol{\mu}_t - \boldsymbol{\mu}_t^{\vartriangleright}), \tag{81}$$

where $\boldsymbol{\mu}_t, \boldsymbol{\Sigma}_t$ are the moments of the full approximate posterior $q(\boldsymbol{x}_t) = \mathcal{N}(\boldsymbol{x}_t \,|\, \boldsymbol{\mu}_t^{\vartriangleright}, \boldsymbol{\Sigma}_t^{\vartriangleright})\mathcal{N}(\boldsymbol{x}_t \,|\, \boldsymbol{\mu}_t^{\backslash\vartriangleright}, \boldsymbol{\Sigma}_t^{\backslash\vartriangleright})$.

*Proof.* Using identities (129)–(130) in Appendix A.2, we can expand the terms on the right-hand side of (81) as

$$\boldsymbol{\Sigma}_t^{-1}\boldsymbol{\mu}_t = (\boldsymbol{\Sigma}_t^{\backslash\vartriangleright})^{-1}\boldsymbol{\mu}_t^{\backslash\vartriangleright} + (\boldsymbol{\Sigma}_t^{\vartriangleright})^{-1}\boldsymbol{\mu}_t^{\vartriangleright} \tag{82}$$

$$\boldsymbol{\Sigma}_t^{-1}\boldsymbol{\mu}_t^{\vartriangleright} = \left((\boldsymbol{\Sigma}_t^{\backslash\vartriangleright})^{-1} + (\boldsymbol{\Sigma}_t^{\vartriangleright})^{-1}\right)\boldsymbol{\mu}_t^{\vartriangleright}. \tag{83}$$

Subtracting (83) from (82) then gives us

$$\boldsymbol{\Sigma}_t^{-1}(\boldsymbol{\mu}_t - \boldsymbol{\mu}_t^{\vartriangleright}) = (\boldsymbol{\Sigma}_t^{\backslash\vartriangleright})^{-1}(\boldsymbol{\mu}_t^{\backslash\vartriangleright} - \boldsymbol{\mu}_t^{\vartriangleright}) \tag{84}$$

as required. $\qquad\qquad\square$

---

[3]This is equivalent to saying that the approximate transition probability $\tilde{p}(\boldsymbol{x}_{t+1}|\boldsymbol{x}_t) = \mathcal{N}(\boldsymbol{x}_{t+1} \,|\, \boldsymbol{M}_t\boldsymbol{x}_t + \boldsymbol{v}_t, \boldsymbol{Q})$ is independent of $\boldsymbol{\mu}_t^{\backslash\vartriangleleft}, \boldsymbol{\Sigma}_t^{\backslash\vartriangleleft}$, which again is a reasonable assumption.

*Proof of Theorem 3.* Using the relations in Table 1, we can re-express the right-hand side of the RTS smoothing equations (78)–(79) in terms of EP messages as

$$\boldsymbol{\mu}_{t|T} = \boldsymbol{\mu}_t^{\backslash \lhd} + \boldsymbol{L}_t(\boldsymbol{\mu}_{t+1} - \boldsymbol{\mu}_{t+1}^{\rhd}), \tag{85}$$

$$\boldsymbol{\Sigma}_{t|T} = \boldsymbol{\Sigma}_t^{\backslash \lhd} + \boldsymbol{L}_t\left(\boldsymbol{\Sigma}_{t+1} - \boldsymbol{\Sigma}_{t+1}^{\rhd}\right)\boldsymbol{L}_t^{\top}. \tag{86}$$

Furthermore, noting the implicit linearisation $\boldsymbol{x}_{t+1} = \boldsymbol{M}_t\boldsymbol{x}_t + \boldsymbol{v}_t + \boldsymbol{\varepsilon}_t$ and using (52), we have

$$\boldsymbol{\Sigma}_{t,t+1|t}^x := \iint (\boldsymbol{x}_t - \boldsymbol{\mu}_{t|t})(\boldsymbol{x}_{t+1} - \boldsymbol{\mu}_{t+1|t})^{\top} \, p(\boldsymbol{x}_t|\boldsymbol{y}_{1:t})p(\boldsymbol{x}_{t+1}|\boldsymbol{y}_{1:t}) \, \mathrm{d}\boldsymbol{x}_t\mathrm{d}\boldsymbol{x}_{t+1} \tag{87}$$

$$\overset{\mathrm{Tab.1}}{=} \iint (\boldsymbol{x}_t - \boldsymbol{\mu}_t^{\backslash \lhd})(\boldsymbol{x}_{t+1} - \boldsymbol{\mu}_{t+1}^{\rhd})^{\top} \, q^{\backslash \lhd}(\boldsymbol{x}_t)q^{\rhd}(\boldsymbol{x}_{t+1}) \, \mathrm{d}\boldsymbol{x}_t\mathrm{d}\boldsymbol{x}_{t+1} \tag{88}$$

$$\overset{(52)}{=} \int (\boldsymbol{x}_t - \boldsymbol{\mu}_t^{\backslash \lhd})(\boldsymbol{M}_t\boldsymbol{x}_t - \boldsymbol{M}_t\boldsymbol{\mu}_t^{\backslash \lhd})^{\top} \, q^{\backslash \lhd}(\boldsymbol{x}_t)\mathrm{d}\boldsymbol{x}_t + \underbrace{\mathbb{E}_{q^{\backslash \lhd}}[(\boldsymbol{x}_t - \boldsymbol{\mu}_t^{\backslash \lhd})\boldsymbol{\varepsilon}_t^{\top}]}_{=\mathbf{0}} \tag{89}$$

$$= \boldsymbol{\Sigma}_t^{\backslash \lhd}\boldsymbol{M}_t^{\top}, \tag{90}$$

which gives us

$$\boldsymbol{L}_t := \boldsymbol{\Sigma}_{t,t+1|t}^x\left(\boldsymbol{\Sigma}_{t+1|t}^x\right)^{-1} = \boldsymbol{\Sigma}_t^{\backslash \lhd}\boldsymbol{M}_t^{\top}(\boldsymbol{\Sigma}_{t+1}^{\rhd})^{-1}. \tag{91}$$

We now show the equivalence in the expressions for the means (76) and (85). To prove this, it suffices to show that

$$\boldsymbol{\Sigma}_t^{x\backslash \lhd}\boldsymbol{\nabla}_{\mu\backslash \lhd}^{\top} = \boldsymbol{L}_t(\boldsymbol{\mu}_{t+1} - \boldsymbol{\mu}_{t+1}^{\rhd}). \tag{92}$$

By identity (136) in Appendix A.2, we have that

$$(\boldsymbol{\Sigma}_{t+1}^{\rhd} + \boldsymbol{\Sigma}_{t+1}^{\backslash \rhd})^{-1} = (\boldsymbol{\Sigma}_{t+1}^{\rhd})^{-1}\underbrace{\left((\boldsymbol{\Sigma}_{t+1}^{\rhd})^{-1} + (\boldsymbol{\Sigma}_{t+1}^{\backslash \rhd})^{-1}\right)^{-1}}_{=\boldsymbol{\Sigma}_{t+1}}(\boldsymbol{\Sigma}_{t+1}^{\backslash \rhd})^{-1} = (\boldsymbol{\Sigma}_{t+1}^{\rhd})^{-1}\boldsymbol{\Sigma}_{t+1}(\boldsymbol{\Sigma}_{t+1}^{\backslash \rhd})^{-1}. \tag{93}$$

Inserting the expression (74) for $\boldsymbol{\nabla}_{\mu\backslash \lhd}$ and invoking Lemma 2, the left-hand side of (92) becomes

$$\boldsymbol{\Sigma}_t^{x\backslash \lhd}\boldsymbol{\nabla}_{\mu\backslash \lhd}^{\top} \overset{(74)}{=} \boldsymbol{\Sigma}_t^{\backslash \lhd}\boldsymbol{M}_t^{\top}(\boldsymbol{\Sigma}_{t+1}^{\rhd} + \boldsymbol{\Sigma}_{t+1}^{\backslash \rhd})^{-1}(\boldsymbol{\mu}_{t+1}^{\backslash \rhd} - \boldsymbol{\mu}_{t+1}^{\rhd}) \tag{94}$$

$$\overset{(93)}{=} \boldsymbol{\Sigma}_t^{\backslash \lhd}\boldsymbol{M}_t^{\top}(\boldsymbol{\Sigma}_{t+1}^{\rhd})^{-1}\boldsymbol{\Sigma}_{t+1}(\boldsymbol{\Sigma}_{t+1}^{\backslash \rhd})^{-1}(\boldsymbol{\mu}_{t+1}^{\backslash \rhd} - \boldsymbol{\mu}_{t+1}^{\rhd}) \tag{95}$$

$$\overset{\mathrm{Lem.2}}{=} \boldsymbol{\Sigma}_t^{\backslash \lhd}\boldsymbol{M}_t^{\top}(\boldsymbol{\Sigma}_{t+1}^{\rhd})^{-1}\underbrace{\boldsymbol{\Sigma}_{t+1}\boldsymbol{\Sigma}_{t+1}^{-1}}_{=\boldsymbol{I}}(\boldsymbol{\mu}_{t+1} - \boldsymbol{\mu}_{t+1}^{\rhd}) \tag{96}$$

$$= \boldsymbol{\Sigma}_t^{\backslash \lhd}\boldsymbol{M}_t^{\top}(\boldsymbol{\Sigma}_{t+1}^{\rhd})^{-1}(\boldsymbol{\mu}_{t+1} - \boldsymbol{\mu}_{t+1}^{\rhd}) \overset{(91)}{=} \boldsymbol{L}_t(\boldsymbol{\mu}_{t+1} - \boldsymbol{\mu}_{t+1}^{\rhd}). \tag{97}$$

This proves the equivalence in the expressions for the mean.

Next, we show the equivalence in the expressions for the covariances (77) and (86). To do this, it suffices to show that

$$-\boldsymbol{\Sigma}_t^{\backslash \lhd}\left[\boldsymbol{\nabla}_{\mu\backslash \lhd}^{\top}\boldsymbol{\nabla}_{\mu\backslash \lhd} - 2\boldsymbol{\nabla}_{\Sigma\backslash \lhd}\right]\boldsymbol{\Sigma}_t^{\backslash \lhd} = \boldsymbol{L}_t\left(\boldsymbol{\Sigma}_{t+1} - \boldsymbol{\Sigma}_{t+1}^{\rhd}\right)\boldsymbol{L}_t^{\top}. \tag{98}$$

First, we use the matrix identity (137) in Appendix A.2, to get

$$\left(\boldsymbol{\Sigma}_{t+1}^{\rhd} + \boldsymbol{\Sigma}_{t+1}^{\backslash \rhd}\right)^{-1} = (\boldsymbol{\Sigma}_{t+1}^{\rhd})^{-1} - (\boldsymbol{\Sigma}_{t+1}^{\rhd})^{-1}\underbrace{\left((\boldsymbol{\Sigma}_{t+1}^{\rhd})^{-1} + (\boldsymbol{\Sigma}_{t+1}^{\backslash \rhd})^{-1}\right)^{-1}}_{=\boldsymbol{\Sigma}_{t+1}}(\boldsymbol{\Sigma}_{t+1}^{\rhd})^{-1} \tag{99}$$

$$= (\boldsymbol{\Sigma}_{t+1}^{\rhd})^{-1}\left(\boldsymbol{\Sigma}_{t+1}^{\rhd} - \boldsymbol{\Sigma}_{t+1}\right)(\boldsymbol{\Sigma}_{t+1}^{\rhd})^{-1}. \tag{100}$$

In (99), we used the fact that the sum of the precision matrices of all messages gives the precision matrix of the marginal. Note that the precision of the cavity distribution is the sum of the precisions of all messages except the forward message. Combining (100) with (75), the left-hand side of (98) yields

$$-\boldsymbol{\Sigma}_t^{\backslash\triangleleft}\boldsymbol{M}_t^\top\left(\boldsymbol{\Sigma}_{t+1}^{\triangleright}+\boldsymbol{\Sigma}_{t+1}^{\backslash\triangleright}\right)^{-1}\boldsymbol{M}_t\boldsymbol{\Sigma}_t^{\backslash\triangleleft}\overset{(100)}{=}\underbrace{\boldsymbol{\Sigma}_t^{\backslash\triangleleft}\boldsymbol{M}_t^\top(\boldsymbol{\Sigma}_{t+1}^{\triangleright})^{-1}}_{=\boldsymbol{L}_t}\left(\boldsymbol{\Sigma}_{t+1}-\boldsymbol{\Sigma}_{t+1}^{\triangleright}\right)\underbrace{(\boldsymbol{\Sigma}_{t+1}^{\triangleright})^{-1}\boldsymbol{M}_t\boldsymbol{\Sigma}_t^{\backslash\triangleleft}}_{=\boldsymbol{L}_t^\top}, \quad (101)$$

which concludes the proof. □

Theorems 1–3 state that for a single iteration of EP, the Gaussian filtering and smoothing updates and the derivative-based EP updates with implicit linearisation are identical. This means that the first iteration of EP can be implemented using derivative-free updates, i.e., the classical update equations for Gaussian filters and smoothers. For subsequent EP iterations, we can incorporate updated messages by using the following equations, which very much resemble the general Gaussian filtering/smoothing equations:

- Measurement message

$$\boldsymbol{\mu}_t = \boldsymbol{\mu}_t^{\backslash\triangle} + \boldsymbol{K}_t(\boldsymbol{y}_t - \boldsymbol{\mu}_t^{y\backslash\triangle}), \tag{102}$$
$$\boldsymbol{\Sigma}_t = \boldsymbol{\Sigma}_t^{\backslash\triangle} - \boldsymbol{K}_t\boldsymbol{\Sigma}_t^{yx\backslash\triangle}, \tag{103}$$

where for $\boldsymbol{x}_t \sim q^{\backslash\triangle}(\boldsymbol{x}_t)$, we define[4]

$$\boldsymbol{\mu}_t^{y\backslash\triangle} = \mathbb{E}_{q^{\backslash\triangle}}[\boldsymbol{h}(\boldsymbol{x}_t)], \quad \boldsymbol{\Sigma}_t^{y\backslash\triangle} = \mathrm{cov}_{q^{\backslash\triangle}}[\boldsymbol{h}(\boldsymbol{x}_t)] + \boldsymbol{R}, \tag{104}$$
$$\boldsymbol{\Sigma}_t^{xy\backslash\triangle} = \mathrm{cov}_{q^{\backslash\triangle}}[\boldsymbol{x}_t, \boldsymbol{h}(\boldsymbol{y}_t)], \quad \boldsymbol{K}_t = \boldsymbol{\Sigma}_t^{xy\backslash\triangle}(\boldsymbol{\Sigma}_t^{y\backslash\triangle})^{-1}. \tag{105}$$

- Backward message

$$\boldsymbol{\mu}_t = \boldsymbol{\mu}_t^{\backslash\triangleleft} + \boldsymbol{L}_t(\boldsymbol{\mu}_{t+1} - \boldsymbol{\mu}_{t+1}^{\backslash\triangleleft}) \tag{106}$$
$$\boldsymbol{\Sigma}_t = \boldsymbol{\Sigma}_t^{\backslash\triangleleft} + \boldsymbol{L}_t(\boldsymbol{\Sigma}_{t+1} - \boldsymbol{\Sigma}_{t+1}^{\backslash\triangleleft})\boldsymbol{L}_t^\top \tag{107}$$
$$\boldsymbol{L}_t = \mathrm{cov}_{q^{\backslash\triangleleft}}[\boldsymbol{x}_t, \boldsymbol{f}(\boldsymbol{x}_t)](\boldsymbol{\Sigma}_{t+1}^{\backslash\triangleleft})^{-1}. \tag{108}$$

Here, we identify $\boldsymbol{\mu}_t^{\backslash\triangleleft} = \mathbb{E}_{q^{\backslash\triangleleft}}[f(\boldsymbol{x}_t)] = \boldsymbol{\mu}^{\triangleright}(\boldsymbol{x}_{t+1})$ and $\boldsymbol{\Sigma}_t^{\backslash\triangleleft} = \mathrm{cov}_{q^{\backslash\triangleleft}}[f(\boldsymbol{x}_t)] + \boldsymbol{Q} = \boldsymbol{\Sigma}_{\boldsymbol{x}_{t+1}}^{\triangleright}$.

*Remark* 3. As shown in Algorithm 1, we initialise all approximate factors $q_i(\boldsymbol{x}_t) = \mathcal{N}(\boldsymbol{0}, \infty\boldsymbol{I})$, $i \in \{\triangleright, \triangle, \triangleleft\}$ with zero mean and infinite variance (zero precision). This implies that the division and multiplication by these densities corresponds to subtracting/adding zeros to precision matrices, respectively. Therefore, before updating $q_i$ for the very first time, it holds that $q_{\backslash i}(\boldsymbol{x}_t) = q(\boldsymbol{x}_t)$, i.e., the cavity computation step leaves the posterior unchanged as division by a zero-precision Gaussian (subtracting zero) has no effect. Therefore, in the first EP iteration, the generic derivative-free update equations in (102)–(103) and (106)–(107) are identical to generic Gaussian filter/smoother updates in (56)–(57) and (78)–(79), respectively.

To summarize, we showed that a single iteration of EP recovers classical Gaussian filters/RTS smoothers as special cases. We can use this close relationship to implement EP without the explicit use of derivatives, which is often more numerically stable for non-linear approximations (Seeger, 2005).

*Remark* 4 (Computational Complexity). The derivative-free updates, which are nearly identical to the generic updates for Gaussian filters/smoothers, tells us that the computational complexity of EP for state estimation is $\mathcal{O}(KF)$, where $K$ is the number of EP iterations, and $F$ is the computational complexity of the corresponding Gaussian filter/smoother. This means that EP has the same computational complexity as a standard Gaussian filter/smoother for each EP iteration.

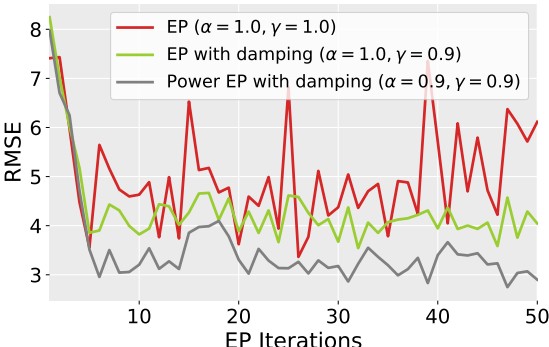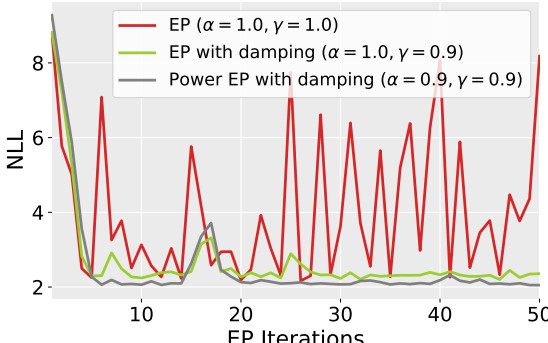

Figure 3: RMSE and NLL for EP (red), damped EP (green), and damped power EP (grey). We extend on the running example from Figure 1, but omit the filtering/smoothing baseline graphs to focus on the effects of damping and power. Both damped EP and damped power EP reduce the oscillations seen in EP substantially. Using power EP, we further improve the performance of EP and damped EP. The power of EP and damping factor are identical, which improves numerical stability as the power and damping cancel out in the computations of the updated marginals, see (116) in Algorithm 2.

## 4 Improving EP by Damping and $\alpha$-Divergence

Revisiting Figure 1, we see that although EP improves the results of the RTS smoother, EP does not converge to a fixed point but oscillates. In general, it is possible that in the case of non-linear systems, approximate EP may deteriorate and produce results inferior to ones of the RTS smoother. It may even diverge and fail to perform the estimation task. One can employ heuristics to determine termination conditions, but there are no established stopping or convergence criteria. In this section, we introduce damping and minimisation of the $\alpha$-divergence as possible remedies to this well-known shortcoming of EP.

### 4.1 Damped EP

Oscillatory behaviour of EP, as observed in Figure 1, is well-known (Heskes & Zoeter, 2003). Standard EP has an iteration scheme without termination criteria, and it is not guaranteed to converge (Minka, 2001a). EP operates on one factor at a time. However, one can establish a convergence criterion based on the covariance of all the factors in the algorithm. This is called *double-loop EP* (Heskes & Zoeter, 2003). In a double-loop EP, the inner loop is a standard EP with damped updates for each factor. The outer loop is used to pick the optimal damping rate for the inner EP update based on all the factors. However, this outer loop is prohibitively expensive for most practical applications. Moreover, non-linear systems, as discussed in this article, use *approximate* EP, and the approximation would lose any convergence guarantees associated with the double-loop version.

Instead of computing an optimal damping for each iteration, we can use a fixed damping rate for the EP updates, which results in the so-called *damped EP* (Minka, 2004). This damping helps the algorithm to avoid limit cycles (Heskes & Zoeter, 2003). This changes the EP updates of the marginal and the factor $q_i$ in (29)–(30) to

$$q'(\boldsymbol{x}_t) = \text{proj}\left[f_i(\boldsymbol{x}_t)q_{\backslash i}(\boldsymbol{x}_t)\right] \tag{109}$$

$$q(\boldsymbol{x}_t)^{\text{new}} \propto q'(\boldsymbol{x}_t)^\gamma q(\boldsymbol{x}_t)^{1-\gamma} \tag{110}$$

$$q_i(\boldsymbol{x}_t)^{\text{new}} \propto \frac{q(\boldsymbol{x}_t)^{\text{new}}}{q_{\backslash i}(\boldsymbol{x}_t)}, \tag{111}$$

where $q'$ corresponds to the undamped update in (29). Damping can be interpreted as a log-opinion pool between the current estimate and an updated estimate, where the damping factor $\gamma \in (0, 1]$ indicates how

---

[4]As usual, the integrals below are only computed approximately, e.g., by using linearisation or the unscented transform.

much weight is given to the current estimate. For $\gamma = 1$, we obtain the classical EP updates from (29)–(30), whereas for $\gamma = 0$, we would not update at all.

The damping factor $\gamma$ is a hyper-parameter, which can be tuned depending on the filtering task. In Figure 3, we plot the effect that damping and $\alpha$-divergence (this will be explained in the next section) has on EP performance, where we use the Unscented Transform to implicitly linearise the system (see also Figures 10 and 11 in Appendix C for the equivalent plots using the Monte-Carlo Transform and Taylor Transform to linearise the system, respectively). We see that a damping factor of 0.9 reduces oscillations and stabilises EP in this particular example. We chose 0.9 to illustrate the effect of damping on oscillations; in Section 5, we provide a more thorough sensitivity analysis for $\gamma$.

### 4.2  $\alpha$-**Divergence**

In some cases, the damping factor $\gamma$ alone may not improve the results of EP. This is particularly true for non-linear systems, where the true posteriors are often multi-modal. EP, as discussed so far in this article, minimizes the KL divergence $\mathrm{KL}\,(p||q)$ between the true posterior and its approximation, so that the approximate posterior $q$ has significant probability mass where $p$ has significant probability mass, i.e., it averages between modes; this direction of the KL divergence is not mode-seeking (Bishop, 2006). Depending on the application, one may want to encourage the algorithm to cover most of the probability mass (i.e., a mode averaging behaviour), in which case standard EP may be desirable. On the other hand, it may be desirable in some applications that the algorithm picks one of the modes rather than averaging them, e.g., in the case of tracking a car moving past a junction. To achieve this flexibility, we can modify the optimisation objective to encourage the desired behaviour by using power EP instead (Minka, 2004). Power EP minimises the $\alpha$-divergence (Amari, 1985) instead of the KL-divergence used by standard EP (Minka, 2004). The $\alpha$-divergence is a generalisation of the KL divergence (Zhu & Rohwer, 1995). We follow (Minka, 2005; Zhu & Rohwer, 1995) and define the $\alpha$-divergence as

$$D_\alpha(p||q) := \frac{\int \alpha p(\boldsymbol{x}) + (1-\alpha)q(\boldsymbol{x}) - p(\boldsymbol{x})^\alpha q(\boldsymbol{x})^{1-\alpha}\,\mathrm{d}\boldsymbol{x}}{\alpha(1-\alpha)}, \tag{112}$$

where $D_\alpha(p||q) = 0$ if and only if $p = q$ and positive otherwise. For $\alpha = 1$, we obtain the KL-divergence $\mathrm{KL}\,(p||q)$ (Zhu & Rohwer, 1995), for $\alpha = 0.5$, the objective corresponds to the Hellinger distance and for $\alpha = 0$, we obtain the KL-divergence $\mathrm{KL}\,(q||p)$ again, except with the arguments reversed. Clearly, the case $\alpha = 1$ recovers standard EP and the case $\alpha = 0$ recovers variational Bayesian inference. Thus, power EP can be seen as a bridge between these two commonly used algorithms (Minka, 2005).

To increase the flexibility of our distributed state-estimation algorithm, we propose to use the $\alpha$-divergence as our cost function to which we find the approximate Gaussian posterior. That is, we wish to find a Gaussian $q'$ such that $q' = \arg\min_q D_\alpha(p||q)$ in our projection step, instead of $q' = \arg\min_q \mathrm{KL}\,(p||q)$ as in EP. Here, the 'true' distribution $p$ is given by the tilted distribution $p(\boldsymbol{x}) = f_i(\boldsymbol{x})q_{\backslash i}(\boldsymbol{x})$ as usual. Typically it would be difficult to compute or even approximate (112). However, we can work around this challenge by using the following property of $\alpha$-divergences (Minka, 2005):

**Proposition 1.** Any stationary point $q$ of $D_\alpha(p||q)$ is also a stationary point of $\mathrm{proj}\left[p(\boldsymbol{x})^\alpha q(\boldsymbol{x})^{1-\alpha}\right]$.

Algorithm 2 describes EP (with damping) using the $\alpha$-divergence to find the approximate posterior, instead of the usual KL divergence minimisation. Setting $p(\boldsymbol{x}) \propto f_i(\boldsymbol{x})q_{\backslash i}(\boldsymbol{x})$, one can check by a straightforward calculation that $\mathrm{proj}\left[p(\boldsymbol{x})^\alpha q(\boldsymbol{x})^{1-\alpha}\right] = \mathrm{proj}\left[f_i(\boldsymbol{x})^\alpha q_{\alpha\backslash i}(\boldsymbol{x})\right]$, where we defined $q_{\alpha\backslash i}(\boldsymbol{x}) \propto q(\boldsymbol{x})/q_i^\alpha(\boldsymbol{x})$. Thus the projection step in this modified EP reads

$$q' = \arg\min_q D_\alpha(p||q) \quad \Leftrightarrow \quad q'(\boldsymbol{x}) = \mathrm{proj}\left[f_i^\alpha(\boldsymbol{x})\,q_{\alpha\backslash i}(\boldsymbol{x})\right], \tag{117}$$

giving us (114) in step 6 of the algorithm.

We see from (117) that an interpretation of power EP is that it allows for the *fractional* inclusion of a true factor $f_i$, which then yields a partial update to the corresponding approximate *fractional* factor $q_i^\alpha$ in step 7 of the algorithm. In Appendix B, we detail the explicit computations of the projection step (117) for state-estimation problems, analogous to the computations we did in Section 3.4 for standard EP.

---

**Algorithm 2** Damped Power EP for Dynamical Systems

---

1: **Init:** Set all factors $q_i^\alpha$ to $\mathcal{N}(\mathbf{0}, \infty \boldsymbol{I})$; Set $q(\boldsymbol{x}_1) = p(\boldsymbol{x}_1)$ and marginals $q(\boldsymbol{x}_{t>1}) = \mathcal{N}(\mathbf{0}, 10^{10} \boldsymbol{I})$
2: **repeat**
3:     **for** $t = 1$ to $T$ **do**
4:         **for** all factors $q_i^\alpha$, where $i = \triangleright, \triangle, \triangleleft$ **do**
5:             **Compute cavity distribution:**

$$q_{\alpha \backslash i}(\boldsymbol{x}_t) \propto q(\boldsymbol{x}_t)/q_i^\alpha(\boldsymbol{x}_t) \tag{113}$$

6:             **Project:**

$$q'(\boldsymbol{x}_t) = \mathrm{proj}\left[f_i^\alpha(\boldsymbol{x}_t)\, q_{\alpha \backslash i}(\boldsymbol{x}_t)\right] \tag{114}$$

7:             **Update fractional factor (with damping):**

$$q_i^\alpha(\boldsymbol{x}_t)^{\mathrm{new}} = q_i^\alpha(\boldsymbol{x}_t)^{1-\gamma}\left(\frac{q'(\boldsymbol{x}_t)}{q_{\alpha \backslash i}(\boldsymbol{x}_t)}\right)^\gamma \propto q_i^\alpha(\boldsymbol{x}_t)\left(\frac{q'(\boldsymbol{x}_t)}{q(\boldsymbol{x}_t)}\right)^\gamma \tag{115}$$

8:             **Update posterior (with damping):**

$$q(\boldsymbol{x}_t)^{\mathrm{new}} \propto q(\boldsymbol{x}_t)\left(\frac{q_i^\alpha(\boldsymbol{x}_t)^{\mathrm{new}}}{q_i^\alpha(\boldsymbol{x}_t)}\right)^{1/\alpha} = q(\boldsymbol{x}_t)\left(\frac{q'(\boldsymbol{x}_t)}{q(\boldsymbol{x}_t)}\right)^{\gamma/\alpha} \tag{116}$$

9:         **end for**
10:     **end for**
11: **until** Convergence or maximum number of iterations exceeded

---

The advantage of applying damped power EP to our running example is demonstrated in Figure 3. We see that the RMSE/NLL are both lower than that of the damped EP. The same choice of damping $\gamma$ and $\alpha$ is convenient mathematically, since the power and damping cancel each other out, which simplifies the posterior update step 8 in Algorithm 2.

## 5 Results

In the following, we evaluate the proposed iterative state estimator on three benchmark problems. We use the root mean square error (RMSE) and the negative log-likelihood (NLL) to evaluate its performance, whose precise expressions can be found in Appendix A.3.

### 5.1 Non-linear Growth Model

First we consider the non-linear benchmark from Section 2.3. The example shown in Figure 3 uses the unscented transform for a single seed. We now repeat the same experiment with 10 different random seeds to evaluate the robustness of different algorithms. Our first goal is to evaluate power EP for different values of power $\alpha$ on the non-linear benchmark problem using different transforms.

#### 5.1.1 Effects of Damping, Power and Linearisation Methods

We test $\alpha$-divergence EP with three different linearisation techniques, the Taylor transform (TT), the (scaled) unscented transform (UT) and the Monte Carlo transform (MCT). For the scaled UT we use parameters $(\alpha_{\mathrm{UT}}, \beta_{\mathrm{UT}}, \kappa_{\mathrm{UT}}) = (1, 2, 3)$ for the transition function and $(\alpha_{\mathrm{UT}}, \beta_{\mathrm{UT}}, \kappa_{\mathrm{UT}}) = (1, 2, 2)$ for the measurement function throughout the paper. Similar to the TT (employed by the EKS) and the UT (employed by the UKS), the MCT computes Gaussian approximations; however, the means and covariances of these Gaussian approximations are computed by sampling from the (Gaussian) cavity distribution and then using a Monte Carlo estimate to obtain the desired means and covariances. We refer to this as the Monte-Carlo Kalman

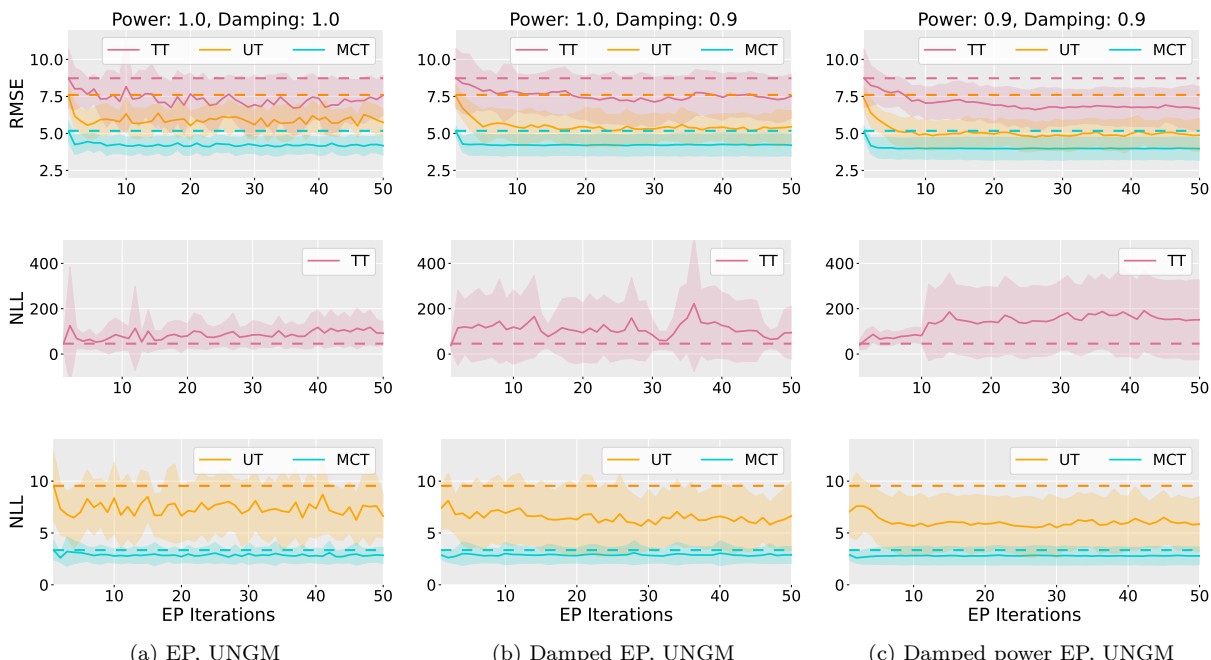

Figure 4: RMSE and NLL performance over multiple runs of the experimental set-up corresponding to Figure 3. Other linearisation methods, namely the Monte Carlo transform (MCT) and the first-order Taylor transform (TT) are shown alongside the unscented transform (UT). Solid lines show average performance across 10 independent evaluations and shaded regions show one standard deviation away from the mean. Dashed lines show average performance of the corresponding Gaussian smoother across the same 10 seeds. (a) $(\gamma, \alpha) = (1, 1)$; (b) $(\gamma, \alpha) = (1, 0.9)$; (c) $(\gamma, \alpha) = (0.9, 0.9)$. MCT is the most stable and performs well for a wide range of damping factors and powers. In contrast, the TT performs poorly on the NLL and is plotted separately from the UT and the MCT due to the difference in scales.

smoother (abbreviated as MCKS). We used $10^4$ samples throughout this paper. The posteriors are updated according to (102)–(103) and (106)–(107). The MCT determines the exact moments in the limit of infinitely many samples; that means, in the context of this article, the MCT serves as a 'best-approximation' baseline, which, however, is computationally more demanding than the TT or the UT.

In Figure 4a, we run standard approximate EP ($\alpha = 1.0$, $\gamma = 1.0$) using all three transforms for approximate inference (TT, UT, MCT). In this setting, the first iteration is identical to the respective Gaussian smoother. We see that both the TT and UT based iterations, in terms of RMSE and NLL, oscillate significantly; for the MCT, the RMSE improves rapidly in the beginning, however, subsequent updates do not improve the results much further; only slight oscillations occur. Note that the overall higher accuracy and stability of the MCT is due to the large number of samples ($10^4$) used for statistical linearisation in this one-dimensional system.

Changing the damping rate to $\gamma = 0.9$ (Figure 4b), as used in the running example, we see that there are fewer oscillations in the RMSE across all transformations (TT, UT, MCT). Further reductions in oscillations can be observed by setting $\gamma$ to smaller values, but at the cost of requiring more EP iterations for convergence. Note that damping does not necessarily improve the RMSE (or the variability of the results) significantly, compared with Figure 4a. An exception is the UT, where damping does improve the results slightly. However, the main impact of the damping factor on the RMSE is therefore to reduce oscillations. The same could be said for the NLL in the UT and MCT examples. However, for the TT, damping appears to impact NLL negatively, increasing the variability of the results even further across seeds.

In Figure 4c, we use $\alpha = 0.9$ and damping $\gamma = 0.9$ as in Figure 3. All transforms now show more stable RMSE curves with visibly more consistent behaviour for TT. The performance, in terms of RMSE, is slightly

improved across all transformations compared to Figure 4b. For the NLL, the same observations hold with the exception of the TT, where the performance is generally worse compared to the previous two cases, with high spread across seeds after around iteration 10. Thus, with the exception of the TT, slightly decreasing the power and damping rate improves the overall results on average, which, for the UT, we found to be consistent across all random seeds used in this experiment. For the MCT, this is true for nine out of ten seeds that we tested, and when improvement is observed, it is more pronounced compared to the UT (see, e.g., Figure 10 in Appendix C).

In Table 2, we display the RMSE and the NLL results of damped power EP after 50 iterations for different combinations of $\alpha$ and $\gamma$. This is compared against the results of the corresponding Kalman-type smoother for the given type of linearisation (e.g. the EKS when using the Taylor transform for linearisation) and the IEKS (Bell, 1994), which we take as baseline methods. Note that the IEKS is a MAP estimator (Bell, 1994) and therefore only produces pointwise predictions $(s_t)_{t=1}^T$. To evaluate the NLL, we supplement these predictions with covariance matrices $(P_t)_{t=1}^T$, obtained by the RTS smoother at every iteration. We see in general that EP iterations improve predictions in terms of the RMSE and NLL. However, setting the $\alpha$ parameter too low could result in bad NLL values, which we discuss in more details in the next section. The IEKS baseline does not produce good predictions in this example; damped power EP overall does better, although there are some exceptions, e.g. the NLL values when $\alpha$ is small.

| | Taylor transform | | Unscented transform | | Monte-Carlo transform | |
|---|---|---|---|---|---|---|
| Baselines | RMSE | NLL | RMSE | NLL | RMSE | NLL |
| EKS/UKS/MCKS | $8.73 \pm 1.99$ | $45.8 \pm 22.9$ | $7.60 \pm 0.73$ | $9.54 \pm 3.28$ | $5.17 \pm 0.69$ | $3.35 \pm 0.94$ |
| IEKS | $15.7 \pm 4.0$ | $98.8 \pm 22.8$ | $-$ | $-$ | $-$ | $-$ |
| $\gamma = 1.0$ | RMSE | NLL | RMSE | NLL | RMSE | NLL |
| Power EP ($\alpha = 1.0$) | $7.58 \pm 1.37$ | $91.8 \pm 52.5$ | $5.74 \pm 0.90$ | $6.63 \pm 2.13$ | $4.17 \pm 0.65$ | $2.84 \pm 0.80$ |
| Power EP ($\alpha = 0.8$) | $6.68 \pm 1.47$ | $166 \pm 190$ | $5.07 \pm 0.99$ | $7.58 \pm 3.45$ | $3.63 \pm 0.80$ | $2.81 \pm 0.97$ |
| Power EP ($\alpha = 0.6$) | $6.87 \pm 1.49$ | $97.8 \pm 87.3$ | $5.69 \pm 1.11$ | $9.39 \pm 4.77$ | $3.17 \pm 0.84$ | $3.52 \pm 1.44$ |
| Power EP ($\alpha = 0.4$) | $7.26 \pm 1.62$ | $200 \pm 223$ | $5.92 \pm 1.38$ | $43.0 \pm 32.8$ | $3.04 \pm 0.84$ | $27.5 \pm 37.7$ |
| Power EP ($\alpha = 0.2$) | $7.28 \pm 1.27$ | $136 \pm 114$ | $6.01 \pm 1.40$ | $197 \pm 161$ | $3.81 \pm 1.06$ | $189 \pm 127$ |
| $\gamma = 0.8$ | RMSE | NLL | RMSE | NLL | RMSE | NLL |
| Power EP ($\alpha = 1.0$) | $7.70 \pm 1.75$ | $91.8 \pm 85.9$ | $4.90 \pm 0.95$ | $5.47 \pm 2.55$ | $4.14 \pm 0.74$ | $2.72 \pm 0.66$ |
| Power EP ($\alpha = 0.8$) | $7.31 \pm 1.08$ | $86.6 \pm 38.0$ | $5.10 \pm 1.17$ | $6.59 \pm 3.00$ | $3.33 \pm 0.60$ | $\mathbf{2.41 \pm 0.75}$ |
| Power EP ($\alpha = 0.6$) | $7.65 \pm 1.06$ | $93.9 \pm 63.9$ | $5.26 \pm 0.98$ | $9.14 \pm 4.19$ | $2.94 \pm 0.70$ | $4.00 \pm 4.66$ |
| Power EP ($\alpha = 0.4$) | $7.71 \pm 1.23$ | $180 \pm 117$ | $4.97 \pm 1.03$ | $28.2 \pm 20.8$ | $2.86 \pm 0.73$ | $31.0 \pm 34.5$ |
| Power EP ($\alpha = 0.2$) | $8.08 \pm 1.29$ | $130 \pm 94$ | $6.14 \pm 1.80$ | $463 \pm 474$ | $3.30 \pm 0.47$ | $258 \pm 176$ |
| $\gamma = 0.6$ | RMSE | NLL | RMSE | NLL | RMSE | NLL |
| Power EP ($\alpha = 1.0$) | $8.29 \pm 1.68$ | $104 \pm 85.2$ | $5.05 \pm 0.83$ | $5.20 \pm 2.25$ | $4.11 \pm 0.71$ | $2.68 \pm 0.65$ |
| Power EP ($\alpha = 0.8$) | $7.53 \pm 1.03$ | $109 \pm 79.1$ | $4.70 \pm 0.78$ | $5.00 \pm 2.11$ | $3.41 \pm 0.77$ | $2.58 \pm 1.04$ |
| Power EP ($\alpha = 0.6$) | $7.80 \pm 1.13$ | $125 \pm 91.2$ | $5.25 \pm 1.13$ | $7.70 \pm 3.73$ | $2.96 \pm 0.70$ | $3.67 \pm 4.06$ |
| Power EP ($\alpha = 0.4$) | $7.89 \pm 1.06$ | $140 \pm 57.2$ | $5.14 \pm 0.99$ | $21.6 \pm 16.0$ | $\mathbf{2.74 \pm 0.55}$ | $22.7 \pm 35.8$ |
| Power EP ($\alpha = 0.2$) | $8.15 \pm 1.25$ | $157 \pm 66$ | $5.56 \pm 0.81$ | $336 \pm 272$ | $2.80 \pm 0.62$ | $247 \pm 183$ |

Table 2: Comparison of RMSE and NLL performance of iterated damped power EP for the UNGM experiment. We compare the results of damped power EP with $\alpha \in \{0.2, 0.4, 0.6, 0.8, 1.0\}$ and $\gamma \in \{0.6, 0.8, 1.0\}$ against the corresponding Gaussian smoother. We also consider the IEKS as a baseline. The mean and standard deviation is displayed across 10 different seeds after 50 EP iterations (we also take 50 iterations for the IEKS), for all transformation types (TT, UT and MCT). We highlight in **bold** the best results for the RMSE and the NLL. In general, iterating EP improves the RMSE and the NLL, provided $\alpha$ is not too small. An exception is the NLL when using the TT for linearisation, in which case iterating EP gives worse results. The IEKS performed poorly in this example and power EP yields better results in general.

### 5.1.2 Sensitivity Analysis

For a more detailed sensitivity analysis, we perform a full sweep across both power $\alpha \in [0.1, 1.0]$ and damping $\gamma \in [0.1, 1.0]$, with a fixed number of 50 EP iterations per experiment. We plot the resulting heat maps of the mean of both RMSE and NLL across ten different seeds and all three transforms in Figures 6 and 7.

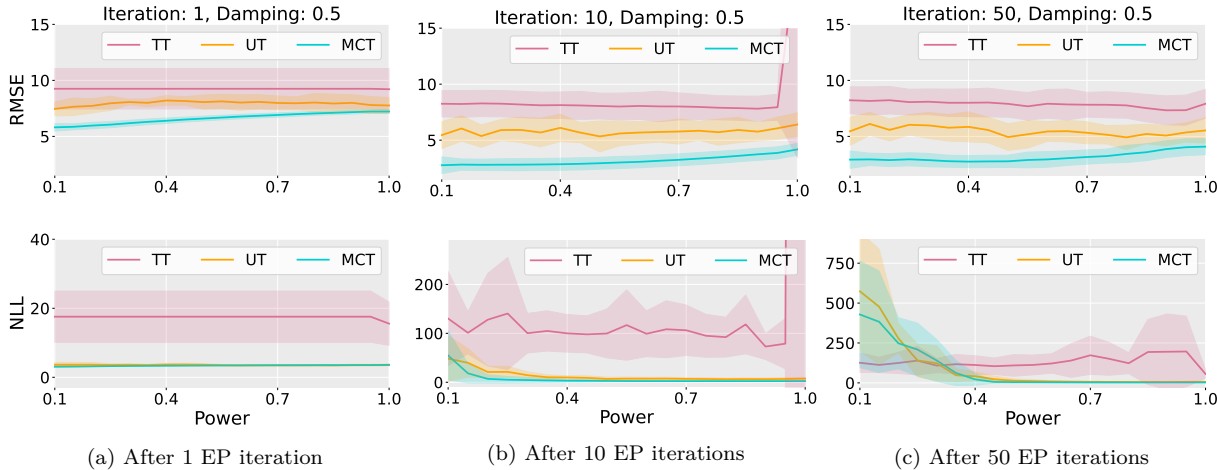

(a) After 1 EP iteration      (b) After 10 EP iterations      (c) After 50 EP iterations

Figure 5: Sensitivity analysis of the proposed algorithm across different choices of power $\alpha$. The damping is constant for all the experiments and set to $\gamma = 0.5$. We average across 10 different seeds with shaded areas indicating the standard deviation. (a) For the first iteration, we see that the results are largely insensitive to power, with RMSE slightly preferring lower powers. (b) and (c) show the corresponding 10th and 50th iteration, respectively. For the UT and the MCT, the RMSE is slightly better at lower powers, whereas the NLL is much worse (and increasing with iterations). The TT is less sensitive to power overall, except in the 10th iteration, there is a sudden jump in the RMSE and NLL at $\alpha = 1.0$.

The RMSE in Figure 6a for the Taylor transform shows optimal performance in the top-right corner, for moderately high values of power and damping $\alpha, \gamma \in [0.8, 1.0)$. Decreasing both parameters leads to increasingly worse performance, especially at very low damping rates of around $\gamma \in [0.1, 0.3]$. The NLL (Figure 7a) is less sensitive to the power, but is similarly affected negatively by decreasing damping rates.

The worst performance, in terms of both RMSE and NLL, mutually occurs under the combination of low damping rates $\gamma \in [0.1, 0.3]$ and high power $\gamma \in [0.9, 1.0]$. This is due to slow convergence at low damping rates and can be checked to converge to better values by iterating sufficiently many times. This is shown in Figure 12 (Appendix C), where we plot the RMSE and NLL values of damped EP after 500 iterations. The damping factor is set to $\gamma = 0.1$ and the TT was used for linearisation. It is somewhat surprising that the RMSE and NLL eventually converge to reasonable values, despite reaching values several orders of magnitude higher ($\mathcal{O}(10^3)$ for the RMSE and $\mathcal{O}(10^{11})$ for the NLL!) during the iterations.

The RMSE plot (Figure 6a) also shows that, while it generally prefers high power and damping rates, fixing $\alpha = 1.0$ can affect the performance negatively. This does not occur for the NLL (Figure 7a), which, on the other hand, sees sub-optimal performance when the damping rate is set to 1.0. This implies that when using the TT for this example, the optimal performance can be achieved by setting both $\alpha$ and $\gamma$ close to, but not equal to 1.0.

For the UT (Figures 6b and 7b), the optimal performance in terms of RMSE occurs for a wider range of parameters compared to the TT, at around $\alpha, \gamma \in (0.4, 1.0)$, and the worst performance occurs at the corner case $\alpha = 0.1, \gamma = 0.1$. Note that the NLL value is still fairly modest in this corner case despite it performing badly at other values of damping, with fixed $\alpha = 0.1$. This can be explained by the slow convergence when the damping factor is small. We see that the NLL is overall insensitive to damping. However, it is sensitive to power. A low power (around $\alpha < 0.3$) harms the performance substantially. This is due to the fact that in the low-power region, power EP is mode-seeking and gives over-confident predictions at incorrect modes. This can be seen for example in Figure 8a, where we plot the outputs of power EP with $\alpha = 0.1$ and $\gamma = 0.4$. In the figure, we see that in the 10th and 50th iterations, the posterior of the latent states does not cover the ground truth well at several places due to overconfident predictions. Contrast this with Figure 8b, which is the same plot except with $\alpha = 0.9$, where much of the ground truth is covered and therefore gives a much better NLL score.

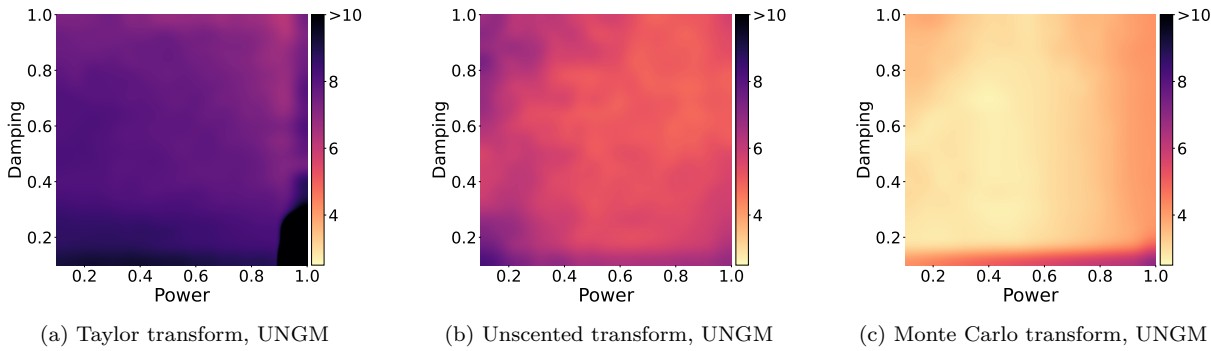

(a) Taylor transform, UNGM      (b) Unscented transform, UNGM      (c) Monte Carlo transform, UNGM

Figure 6: RMSE heatmap for the Uniform Non-Linear Growth model experiment. We fixed the number of iterations to 50 and plot the average RMSE over 10 independent runs of the proposed method for varying powers and damping factors. We vary both power and damping between $[0.1, 1.0]$ for different transformation types. Optimal performance is achieved for mid-to-high damping and power factors. Very low damping rates yield poor performance due to slow convergence.

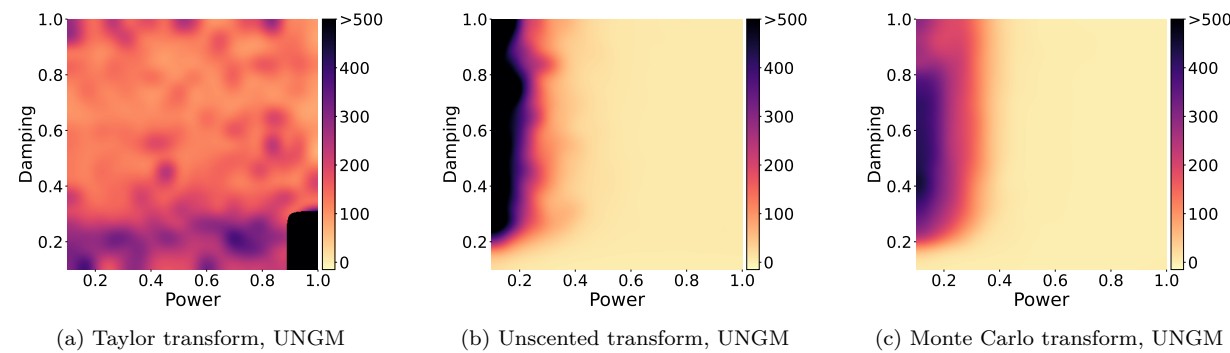

(a) Taylor transform, UNGM      (b) Unscented transform, UNGM      (c) Monte Carlo transform, UNGM

Figure 7: Negative Log-Likelihood (NLL) heatmap for the UNGM experiment. Again, we fixed the number of iterations to 50 and plot the average NLL over 10 independent runs. Similar to the RMSE, the best performance occurs for mid-to-high power and damping factors. However in contrast to the RMSE, worst performance in the UT and the MCT occur at very low powers. The poor performance in the bottom-right corner in the MCT is due to slow convergence.

For the MCT, Figure 7c shows that the NLL results are similar to the ones obtained with the UT, with low power of around $\alpha < 0.3$ being harmful to the performance. In contrast, the result in Figure 6c, while subtle, indicates that the RMSE slightly prefers mid-to-low power at around $\alpha < 0.6$. This is indicated more clearly in Figure 5, where we plotted the RMSE and NLL against the power at different EP iterations. The significant degradation in NLL due to low power however is clearly not worth the slight gain in RMSE. Hence, it is optimal to use mid-to-high power of around $\alpha > 0.6$ in this example. We see that both the RMSE and NLL results in Figures 6c and 7c are less sensitive to damping rates; only the RMSE is affected negatively by low damping rates of around $\alpha < 0.2$, due to the slower convergence rate in this region.

## 5.2 Bearings-only Tracking of a Turning Target

We now test our proposed method for a classic bearings-only tracking problem that is commonly encountered in passive sensor environments. In this problem, we assume that a single target is executing a maneuvering turn, with four sensors measuring the angles between the target and the sensors, similar to the example

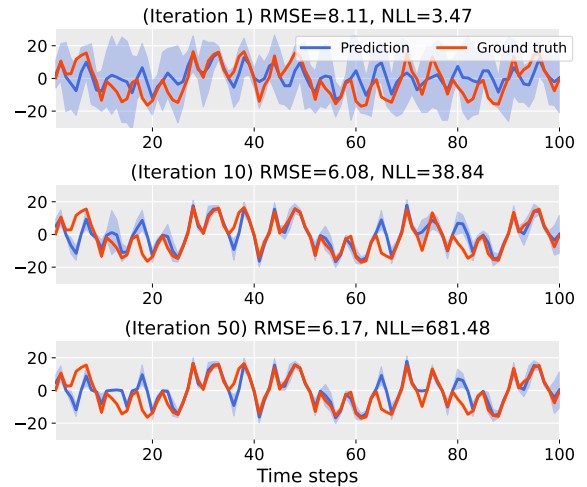 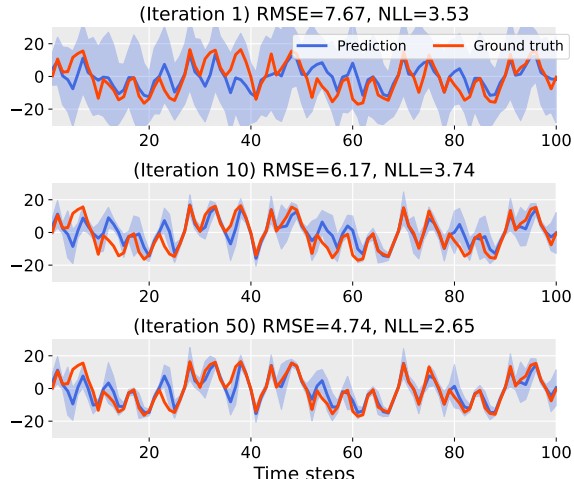

(a) $\alpha = 0.1$. While the RMSE values generally improve, the NLL values get progressively worse with iterations as a result of the mode-seeking behaviour of power EP with $\alpha = 0.1$.

(b) $\alpha = 0.9$. The RMSE decreases with iterations. However, in contrast to the results in (a), we see that the NLL decreases too due to better uncertainty coverage.

Figure 8: Outputs of damped power EP for a single run of the UNGM experiment. The ground-truth trajectory is shown in red; the shaded blue areas indicate the standard deviation around the mean (solid blue) of the posterior state estimate using EP with damping factor $\gamma = 0.4$. The unscented transform is used for linearisation. We show the results at EP iterations 1, 10 and 50 for (a) $\alpha = 0.1$; (b) $\alpha = 0.9$.

considered in Särkkä & Hartikainen (2010). The dynamical system is given by

$$
\begin{aligned}
\boldsymbol{x}_t &= \boldsymbol{A}(\omega_{t-1})\boldsymbol{x}_{t-1} + \boldsymbol{w}_{t-1}, & \boldsymbol{w}_{t-1} &\sim \mathcal{N}(\boldsymbol{0}, \boldsymbol{Q}), \\
\boldsymbol{y}_t &= \boldsymbol{h}(\boldsymbol{x}_t) + \boldsymbol{v}_t, & \boldsymbol{v}_t &\sim \mathcal{N}(\boldsymbol{0}, \boldsymbol{R}),
\end{aligned}
\tag{118}
$$

where the state is given by $\boldsymbol{x} = (x_1, \dot{x}_1, x_2, \dot{x}_2, \omega)^\top \in \mathbb{R}^5$, such that the position of the target is $\boldsymbol{z} = (x_1, x_2)^\top$, the corresponding velocities are $\dot{\boldsymbol{z}} = (\dot{x}_1, \dot{x}_2)^\top$, respectively, and $\omega$ is the time-varying turning rate of the maneuver. The transition matrix $\boldsymbol{A}(\omega)$ takes the form

$$
\boldsymbol{A}(\omega) = \begin{pmatrix}
1 & \frac{\sin(\omega \Delta t)}{\omega} & 0 & -\frac{1 - \cos(\omega \Delta t)}{\omega} & 0 \\
0 & \cos(\omega \Delta t) & 0 & -\sin(\omega \Delta t) & 0 \\
0 & \frac{1 - \cos(\omega \Delta t)}{\omega} & 1 & \frac{\sin(\omega \Delta t)}{\omega} & 0 \\
0 & \sin(\omega \Delta t) & 0 & \cos(\omega \Delta t) & 0 \\
0 & 0 & 0 & 0 & 1
\end{pmatrix},
\tag{119}
$$

where $\Delta t$ is the timestep between two consecutive measurements. The measurement model $\boldsymbol{h}(\boldsymbol{x}) = (h^1(\boldsymbol{x}), h^2(\boldsymbol{x}), h^3(\boldsymbol{x}), h^4(\boldsymbol{x}))^\top \in \mathbb{R}^4$ is given by

$$
h^i(\boldsymbol{x}) = \tan^{-1}\left(\frac{x_2 - s_2^i}{x_1 - s_1^i}\right), \quad i \in \{1, 2, 3, 4\},
\tag{120}
$$

which measures the angle between the target and the $i$-th sensor $\boldsymbol{s}^i = (s_1^i, s_2^i)^\top$. Following the same setup as in the target tracking example considered by Arasaratnam & Haykin (2009), we choose the initial state distribution $p(\boldsymbol{x}_0) = \mathcal{N}(\boldsymbol{\mu}_0, \boldsymbol{\Sigma}_0)$, where

$$
\boldsymbol{\mu}_0 = (1000\text{m}, 300\text{ms}^{-1}, 1000\text{m}, 0\text{ms}^{-1}, -3°\text{s}^{-1})^\top,
\tag{121}
$$

$$
\boldsymbol{\Sigma}_0 = \text{diag}(100\text{m}^2, 10\text{m}^2\text{s}^{-2}, 100\text{m}^2, 10\text{m}^2\text{s}^{-2}, 10^{-4}\text{rad}^2\text{s}^{-2}),
\tag{122}
$$

and choose our model error covariance to be of the form $\boldsymbol{Q} = \mathrm{diag}(q_1\boldsymbol{M}, q_1\boldsymbol{M}, q_2\Delta t)$, where $q_1 = 0.1\mathrm{m}^2\mathrm{s}^{-3}$, $q_2 = 1.75 \times 10^{-1}\mathrm{s}^{-3}$, and

$$\boldsymbol{M} = \begin{pmatrix} \frac{\Delta t^3}{3} & \frac{\Delta t^2}{2} \\ \frac{\Delta t^2}{2} & \Delta t \end{pmatrix}. \tag{123}$$

Since the measurements $\boldsymbol{y}_t$ only depend on the position variables, the cross-correlation in (123) is important for reconstructing the full latent state. The observation errors for the four sensors are assumed to be independent and given by $R_i = \mathcal{N}(0, r_i^2)$, where $r_i = \sqrt{10} \times 10^{-3}\mathrm{rad}$ for all $i \in \{1, 2, 3, 4\}$. We consider trajectories of length 50s and set the sampling frequency to $\Delta t = 1$s. The sensors are placed on the four corners of the square $\boldsymbol{s}^1 = (-20\mathrm{km}, 20\mathrm{km}), \boldsymbol{s}^2 = (20\mathrm{km}, 20\mathrm{km}), \boldsymbol{s}^3 = (-20\mathrm{km}, -20\mathrm{km}), \boldsymbol{s}^4 = (20\mathrm{km}, -20\mathrm{km})$ and assumed to produce measurements at every time step.

| | Position ($\boldsymbol{z}$) | | Velocity ($\dot{\boldsymbol{z}}$) | | Angular velocity ($\omega$) | |
|---|---|---|---|---|---|---|
| TT | RMSE | NLL | RMSE | NLL | RMSE ($\times 10^{-3}$) | NLL |
| EKS | $43.1 \pm 5.5$ | $11.0 \pm 1.2$ | $11.1 \pm 2.4$ | $11.9 \pm 2.3$ | $1.44 \pm 0.35$ | $-2.80 \pm 0.27$ |
| IEKS | $\mathbf{24.8 \pm 1.8}$ | $\mathbf{8.33 \pm 0.26}$ | $\mathbf{7.10 \pm 2.33}$ | $\mathbf{5.01 \pm 0.81}$ | $\mathbf{1.26 \pm 0.33}$ | $-3.03 \pm 0.10$ |
| PEP ($\alpha = 1.0$) | $43.1 \pm 5.5$ | $11.0 \pm 1.2$ | $11.1 \pm 2.4$ | $11.9 \pm 2.3$ | $1.44 \pm 0.35$ | $-2.80 \pm 0.27$ |
| PEP ($\alpha = 0.8$) | $36.8 \pm 5.8$ | $9.65 \pm 0.84$ | $9.82 \pm 2.14$ | $11.9 \pm 3.1$ | $\mathbf{1.26 \pm 0.31}$ | $-3.00 \pm 0.14$ |
| PEP ($\alpha = 0.6$) | $36.8 \pm 5.9$ | $9.62 \pm 0.83$ | $9.80 \pm 2.13$ | $12.1 \pm 3.3$ | $\mathbf{1.26 \pm 0.33}$ | $-2.99 \pm 0.16$ |
| PEP ($\alpha = 0.4$) | $36.7 \pm 5.9$ | $9.61 \pm 0.83$ | $9.79 \pm 2.12$ | $12.1 \pm 3.3$ | $\mathbf{1.26 \pm 0.33}$ | $-2.99 \pm 0.16$ |
| PEP ($\alpha = 0.2$) | $36.7 \pm 5.9$ | $9.61 \pm 0.82$ | $9.78 \pm 2.11$ | $12.2 \pm 3.3$ | $\mathbf{1.26 \pm 0.33}$ | $-2.99 \pm 0.16$ |
| UT | RMSE | NLL | RMSE | NLL | RMSE ($\times 10^{-3}$) | NLL |
| UKS | $26.5 \pm 1.9$ | $8.58 \pm 0.49$ | $7.64 \pm 1.80$ | $5.66 \pm 0.42$ | $\mathbf{1.26 \pm 0.28}$ | $\mathbf{-3.04 \pm 0.07}$ |
| PEP ($\alpha = 1.0$) | $26.5 \pm 1.9$ | $8.58 \pm 0.49$ | $7.64 \pm 1.80$ | $5.66 \pm 0.42$ | $\mathbf{1.26 \pm 0.28}$ | $\mathbf{-3.04 \pm 0.07}$ |
| PEP ($\alpha = 0.8$) | $26.2 \pm 2.1$ | $8.51 \pm 0.37$ | $7.50 \pm 1.87$ | $5.37 \pm 0.48$ | $\mathbf{1.26 \pm 0.29}$ | $\mathbf{-3.04 \pm 0.08}$ |
| PEP ($\alpha = 0.6$) | $26.2 \pm 2.1$ | $8.50 \pm 0.36$ | $7.49 \pm 1.86$ | $5.33 \pm 0.47$ | $\mathbf{1.26 \pm 0.29}$ | $\mathbf{-3.04 \pm 0.08}$ |
| PEP ($\alpha = 0.4$) | $26.2 \pm 2.1$ | $8.50 \pm 0.36$ | $7.48 \pm 1.85$ | $5.33 \pm 0.46$ | $\mathbf{1.26 \pm 0.29}$ | $\mathbf{-3.04 \pm 0.08}$ |
| PEP ($\alpha = 0.2$) | $26.2 \pm 2.1$ | $8.50 \pm 0.36$ | $7.48 \pm 1.85$ | $5.32 \pm 0.45$ | $\mathbf{1.26 \pm 0.28}$ | $\mathbf{-3.04 \pm 0.08}$ |
| MCT | RMSE | NLL | RMSE | NLL | RMSE ($\times 10^{-3}$) | NLL |
| MCKS | $28.6 \pm 3.1$ | $8.70 \pm 0.45$ | $9.85 \pm 2.17$ | $6.03 \pm 0.48$ | $1.66 \pm 0.35$ | $-2.85 \pm 0.12$ |
| PEP ($\alpha = 1.0$) | $26.4 \pm 2.2$ | $8.57 \pm 0.58$ | $7.68 \pm 1.84$ | $5.69 \pm 0.55$ | $1.27 \pm 0.29$ | $-3.03 \pm 0.08$ |
| PEP ($\alpha = 0.8$) | $26.1 \pm 2.0$ | $8.51 \pm 0.40$ | $7.58 \pm 1.89$ | $5.40 \pm 0.57$ | $1.27 \pm 0.29$ | $-3.03 \pm 0.08$ |
| PEP ($\alpha = 0.6$) | $25.9 \pm 2.2$ | $8.50 \pm 0.42$ | $7.49 \pm 1.89$ | $5.35 \pm 0.58$ | $1.27 \pm 0.29$ | $-3.03 \pm 0.08$ |
| PEP ($\alpha = 0.4$) | $26.0 \pm 2.2$ | $8.50 \pm 0.41$ | $7.44 \pm 1.93$ | $5.34 \pm 0.53$ | $\mathbf{1.26 \pm 0.30}$ | $\mathbf{-3.04 \pm 0.08}$ |
| PEP ($\alpha = 0.2$) | $26.3 \pm 2.1$ | $8.52 \pm 0.43$ | $7.55 \pm 1.86$ | $5.38 \pm 0.55$ | $1.27 \pm 0.29$ | $-3.03 \pm 0.08$ |

Table 3: Comparison of RMSE and NLL performance of iterated power EP (abbreviated as PEP) for the bearings-only turning target tracking problem. We compare the results for $\alpha \in \{0.2, 0.4, 0.6, 0.8, 1.0\}$. The damping factor is fixed at $\gamma = 1.0$ for all experiments; changing the damping factor did not affect the results much. We display the mean and standard deviation across 10 different seeds after 50 EP iterations, for all transformation types (TT, UT and MCT). As baselines, we also display the results of the corresponding Gaussian smoother and the IEKS after 50 iterations. For each variable, the best results in terms of the RMSE and the NLL are highlighted in **bold**. We observe that when $\alpha = 1.0$, i.e., standard approximate EP, iteration does not improve upon the results of the corresponding Kalman-type smoother for the TT and UT (improvement is seen however for the MCT). Choosing smaller values of $\alpha$ can improve the result, which is most signficant in the position variable ($\boldsymbol{z}$) when using the TT. However overall, the best performance is obtained by the IEKS, with power EP performing competitively only in the $\omega$ variable.

We compare the performance of iterated power EP for various values of $\alpha$, including $\alpha = 1.0$, which is equivalent to standard approximate EP. In all the experiments, we take 50 EP iterations, which we verified is enough to ensure convergence, and set the damping factor to $\gamma = 1.0$ (i.e., no damping). No oscillatory behaviour was observed during the EP iterations (except for the MCT, where oscillations occurred primarily due to its stochastic nature; taking larger number of samples helped reduce the oscillations) and therefore no damping was necessary to aid convergence. For reference, we also compared the results against the

corresponding Kalman-type smoother for the given type of linearisation and the IEKS (Bell, 1994), which form our baseline methods.

Below are further details on the configurations used in the experiments

- The parameters of the scaled UT were set to $(\alpha_{\mathrm{UT}}, \beta_{\mathrm{UT}}, \kappa_{\mathrm{UT}}) = (1, 0, -1)$ for both the transition and measurement functions, in accordance with Särkkä & Hartikainen (2010).

- $10^4$ particles were used in all the MCT experiments.

- We used the values $\alpha \in \{0.2, 0.4, 0.6, 0.8, 1.0\}$ for the power.

The results of the experiments are displayed in Table 3. For this system, we observe that simply iterating the approximate EP (i.e., power EP with $\alpha = 1.0$) does not improve the results of the corresponding Kalman-type smoother when we use the TT and UT for linearisation. This is true for both the RMSE and NLL performance across all variables (position, velocity, angular velocity). However, by taking smaller values of $\alpha$, we see an overall gain in performance. For the TT, significant improvement is obtained in the RMSE for the position variable when taking $\alpha \le 0.8$, with the best performance occurring at $\alpha = \{0.2, 0.4\}$ although the results are very close when $\alpha \in \{0.6, 0.8\}$. We also see improvements in the RMSE for the velocity and angular velocity variables by taking $\alpha \le 0.8$, although they are less pronounced than the former. The NLL also sees some improvement by taking smaller values of $\alpha$, except for the velocity variable, where taking $\alpha \le 0.6$ appears to slightly deteriorate the NLL performance. This is likely due to overconfident predictions owing to the mode-seeking nature of power EP for small $\alpha$. While similar improvements in the RMSE and NLL are observed for the UT, the difference is not as significant (in fact, no difference at all in the angular velocity variable). This is likely due to the UKS already performing very well on this task and therefore has smaller margins for improvements.

On the other hand, when using the MCT for linearisation, we see that most of the improvements in the RMSE and the NLL comes from iterating as opposed to taking smaller $\alpha$, although taking smaller $\alpha$ can help squeeze out better performance. This is evident from the fact that the difference in values between the result from the Kalman smoother and EP after 50 iterations are on average greater than the difference between the results of power EP with different $\alpha$. Interestingly, we also observed that smaller values for $\alpha$ can help to reduce the oscillations during the EP iterations even with moderate number of particles (in this case, $10^4$). We show this in Figure 13 (Appendix C), where we plotted the average RMSE and NLL across all variables (position, velocity and angular velocity) against the number of EP iterations for $(\alpha, \gamma) \in \{(1.0, 1.0), (1.0, 0.8), (0.8, 1.0)\}$. When $\alpha = 1.0$, we see that damping reduces the oscillations slightly in both the RMSE and NLL, however the oscillations in the NLL is still very large. Taking $\alpha = 0.8$ reduces this oscillation in the NLL dramatically, even when no damping is applied (although the RMSE becomes slightly oscillatory again).

While power EP does give better results than the corresponding Gaussian smoother, it does not outperform the IEKS, which overall gives the best performance in this example. Only in the $\omega$ variable does power EP yield competitive results.

Overall, we conclude that for this system, the IEKS performs the best, although power EP can give good results. We see that simply iterating the EP does not alter the smoother results when the TT and UT are used for linearisation. However, notable improvements can be observed by considering smaller values of $\alpha$ in power EP. When using the MCT, we see benefits from iterating the approximate EP, even when $\alpha = 1.0$. Moreover, further gains in performance can be obtained by taking smaller values of $\alpha$. The results for the IEKS however are hard to beat even with these improvements, with power EP only slightly outperforming the IEKS in the $\omega$ variable.

### 5.3 Lorenz 96 Model

In our final example, we test our method on the Lorenz 96 model (Lorenz, 1996), another commonly used benchmark for nonlinear filters/smoothers (Brajard et al., 2020; Fablet et al., 2021; Ott et al., 2004). The

model is defined by a system of ODEs

$$\frac{\mathrm{d}x_i}{\mathrm{d}t} = (x_{i+1} - x_{i-2})x_{i-1} - x_i + F, \quad i = 1, \ldots, d,$$
$$\text{where} \quad x_{-1} = x_{d-1}, \quad x_0 = x_d, \quad x_{d+1} = x_1. \tag{124}$$

Here, $d > 4$ is the dimension of the system and $F$ is a forcing constant, which we set to $F = 8$. For our transition model $\boldsymbol{f}$, we consider a fourth-order Runge–Kutta discretisation of system (124) with step size $\Delta t = 0.05$. The model error covariance is $\boldsymbol{Q} = 0.1\boldsymbol{I}$. For the measurement model, we consider the quadratic operator $\boldsymbol{h}(\boldsymbol{x}) = (x_1^2, \ldots, x_d^2)^\top \in \mathbb{R}^d$, with error covariance $\boldsymbol{R} = \boldsymbol{I}$.

Since we are free to choose the state dimension $d$ of the system, we investigate how our method performs with increasing dimensions. We evaluate this for systems with dimensions $d \in \{5, 10, 20, 40, 100, 200\}$. Our initial state $\boldsymbol{\mu}_0$ is obtained by spinning-up the model (124) using the fourth-order Runge–Kutta (RK4) discretisation, starting from the state $(0.01, 0, 0, \ldots, 0)^\top \in \mathbb{R}^d$ and taking 99 RK4 timesteps with step size $\Delta t = 0.05$. We set the initial state distribution to $p(\boldsymbol{x}_0) = \mathcal{N}(\boldsymbol{x}_0 \mid \boldsymbol{\mu}_0, \boldsymbol{I})$.

The RMSE and NLL results of power EP applied to the Lorenz 96 model across various values of $d$ is shown in Table 4. We used the values $\alpha \in \{0.1, 0.8, 1.0\}$ for the power factor and took 10 EP iterations with no damping for all the experiments. Oscillatory behaviour was not observed during the EP iterations and adding damping did not change the results by much. For the linearisation methods, we considered the TT and the UT, with the same UT parameters $(\alpha_{\mathrm{UT}}, \beta_{\mathrm{UT}}, \kappa_{\mathrm{UT}})$ that we used for the UNGM experiment (Section 5.1). We found the MCT with $10^4$ particles to be numerically unstable at $d \geq 40$ for some seed settings and therefore do not report results for MCT here.[5]

We see that overall, iterating EP helps to improve both the NLL and RMSE performance over the corresponding Kalman-type smoothers. In contrast, the values of $\alpha$ do not affect the performance by much. For the TT, improvements by EP iteration is observed for the RMSE across all values of $d$, even outperforming the IEKS. At low dimensions ($d \leq 10$) we see a similar improvement in the NLL. However, for $d \geq 20$, EP iterations appear to affect the NLL results negatively for some values of $\alpha$. Moreover, the obtained NLL values are worse by several orders of magnitude compared to those obtained for $d \leq 10$. To investigate this further, we plotted the results for individual seeds in Figure 9 with $d = 20$ and $(\alpha, \gamma) = (1.0, 1.0)$. We see that in nine out of ten cases, EP iterations actually improve the NLL performance. However, there is a single outlier case that starts out significantly worse than the rest in the first iteration and becomes worse with subsequent iterations for the NLL (for the RMSE, iteration appears to always improve the results, which means that the uncertainty estimates become increasingly overconfident). These outlier scenarios that start out badly are therefore responsible for the significantly worse NLL values and large spread that we see for $d \geq 20$, even though in most cases, EP iterations actually improve the NLL.

For the UT, we see improvements in the RMSE and NLL with EP iterations for all experiments that we conducted. Furthermore, we see that the gain in performance due to iterations becomes increasingly pronounced with increasing $d$, especially for the NLL. For example, at $d = 5$, we only see an improvement in the NLL by a few decimal points upon iterating. However at $d = 200$, we see a performance gain of $\mathcal{O}(10^2)$—much more than what we had anticipated. While less obvious, this trend of increasing performance gain with increasing dimensions also holds for the RMSE. In Figure 14 (Appendix C), we show Hovmöller diagrams illustrating the absolute errors for the predictions made by the Unscented Kalman smoother and the errors made by approximate EP with $(\alpha, \gamma) = (1.0, 1.0)$ after 10 iterations. The state dimension is set to $d = 200$. The improvements in predictions due to EP iteration is clear—the errors made by EP after 10 iterations is visibly smaller compared to the errors made after one iteration (i.e., the UKS).

Overall, we conclude that when using the UT for linearisation, EP iterations have the effect of improving predictions in this system, which becomes more significant at higher dimensions. Thus, our method may prove to be more beneficial when applied to higher-dimensional systems, but with the downside of increased computational cost. While the cost of our method only scales linearly in the number of EP iterations, the cost of running a single iteration is already expensive for higher-dimensional systems. Therefore, it may not be feasible in practice to run many iterations.

---

[5]Experiments with $> 10^4$ many particles would have required unjustified computational resources and time.

|  | Taylor transform | | Unscented transform | |
| --- | --- | --- | --- | --- |
| $d = 5$ | RMSE | NLL | RMSE | NLL |
| EKS/UKS | $0.43 \pm 0.04$ | $-2.47 \pm 0.49$ | $0.42 \pm 0.05$ | $-2.61 \pm 0.38$ |
| IEKS | $0.41 \pm 0.04$ | $-2.69 \pm 0.40$ | $-$ | $-$ |
| Power EP ($\alpha = 1.0$) | $\mathbf{0.40 \pm 0.03}$ | $-2.77 \pm 0.38$ | $\mathbf{0.40 \pm 0.03}$ | $-2.78 \pm 0.36$ |
| Power EP ($\alpha = 0.8$) | $\mathbf{0.40 \pm 0.03}$ | $-2.78 \pm 0.37$ | $\mathbf{0.40 \pm 0.03}$ | $\mathbf{-2.80 \pm 0.36}$ |
| Power EP ($\alpha = 0.1$) | $0.41 \pm 0.03$ | $-2.77 \pm 0.37$ | $\mathbf{0.40 \pm 0.03}$ | $-2.78 \pm 0.36$ |
| $d = 10$ | RMSE | NLL | RMSE | NLL |
| EKS/UKS | $0.60 \pm 0.04$ | $-5.19 \pm 0.45$ | $0.60 \pm 0.02$ | $-5.38 \pm 0.29$ |
| IEKS | $0.58 \pm 0.03$ | $-5.71 \pm 0.30$ | $-$ | $-$ |
| Power EP ($\alpha = 1.0$) | $\mathbf{0.56 \pm 0.02}$ | $-5.86 \pm 0.31$ | $\mathbf{0.56 \pm 0.02}$ | $-5.85 \pm 0.29$ |
| Power EP ($\alpha = 0.8$) | $\mathbf{0.56 \pm 0.02}$ | $\mathbf{-5.88 \pm 0.31}$ | $\mathbf{0.56 \pm 0.02}$ | $\mathbf{-5.88 \pm 0.30}$ |
| Power EP ($\alpha = 0.1$) | $0.57 \pm 0.03$ | $-5.84 \pm 0.35$ | $\mathbf{0.56 \pm 0.02}$ | $-5.86 \pm 0.33$ |
| $d = 20$ | RMSE | NLL | RMSE | NLL |
| EKS/UKS | $2.49 \pm 4.29$ | $1543 \pm 4626$ | $0.95 \pm 0.09$ | $-9.33 \pm 1.19$ |
| IEKS | $2.37 \pm 4.22$ | $1789 \pm 5390$ | $-$ | $-$ |
| Power EP ($\alpha = 1.0$) | $2.25 \pm 4.26$ | $1637 \pm 4943$ | $0.84 \pm 0.05$ | $-11.0 \pm 0.74$ |
| Power EP ($\alpha = 0.8$) | $2.34 \pm 4.23$ | $1741 \pm 5252$ | $\mathbf{0.82 \pm 0.04}$ | $\mathbf{-11.3 \pm 0.5}$ |
| Power EP ($\alpha = 0.1$) | $2.36 \pm 4.23$ | $1792 \pm 5400$ | $0.87 \pm 0.12$ | $-10.5 \pm 2.2$ |
| $d = 40$ | RMSE | NLL | RMSE | NLL |
| EKS/UKS | $3.27 \pm 3.92$ | $1462 \pm 3184$ | $1.41 \pm 0.09$ | $-16.5 \pm 1.19$ |
| IEKS | $3.15 \pm 3.85$ | $1540 \pm 3388$ | $-$ | $-$ |
| Power EP ($\alpha = 1.0$) | $3.11 \pm 3.86$ | $1424 \pm 3146$ | $1.21 \pm 0.05$ | $-21.6 \pm 1.22$ |
| Power EP ($\alpha = 0.8$) | $3.11 \pm 3.85$ | $1495 \pm 3298$ | $\mathbf{1.20 \pm 0.05}$ | $\mathbf{-21.9 \pm 1.3}$ |
| Power EP ($\alpha = 0.1$) | $3.13 \pm 3.86$ | $1544 \pm 3398$ | $1.28 \pm 0.20$ | $-18.3 \pm 9.5$ |
| $d = 100$ | RMSE | NLL | RMSE | NLL |
| EKS/UKS | $3.04 \pm 2.01$ | $741 \pm 2064$ | $2.54 \pm 0.08$ | $-22.1 \pm 2.31$ |
| IEKS | $2.90 \pm 1.92$ | $850 \pm 2410$ | $-$ | $-$ |
| Power EP ($\alpha = 1.0$) | $2.85 \pm 1.94$ | $741 \pm 2125$ | $1.98 \pm 0.08$ | $-49.2 \pm 2.35$ |
| Power EP ($\alpha = 0.8$) | $2.85 \pm 1.93$ | $821 \pm 2342$ | $\mathbf{1.97 \pm 0.05}$ | $\mathbf{-51.1 \pm 0.9}$ |
| Power EP ($\alpha = 0.1$) | $2.88 \pm 1.93$ | $849 \pm 2410$ | $\mathbf{1.97 \pm 0.12}$ | $-50.6 \pm 4.2$ |
| $d = 200$ | RMSE | NLL | RMSE | NLL |
| EKS/UKS | $5.09 \pm 2.93$ | $1399 \pm 2374$ | $4.47 \pm 0.17$ | $8.72 \pm 4.13$ |
| IEKS | $4.92 \pm 2.91$ | $1753 \pm 3024$ | $-$ | $-$ |
| Power EP ($\alpha = 1.0$) | $4.84 \pm 2.96$ | $1604 \pm 2833$ | $2.95 \pm 0.05$ | $-89.0 \pm 1.67$ |
| Power EP ($\alpha = 0.8$) | $4.83 \pm 2.96$ | $1698 \pm 2968$ | $2.80 \pm 0.04$ | $-103 \pm 1$ |
| Power EP ($\alpha = 0.1$) | $4.87 \pm 2.93$ | $1752 \pm 3027$ | $\mathbf{2.61 \pm 0.03}$ | $\mathbf{-111 \pm 1}$ |

Table 4: Comparison of RMSE and NLL performance of iterated power EP for the Lorenz 96 experiment across different dimensions $d \in \{5, 10, 20, 40, 100, 200\}$. We compare the results for $\alpha \in \{0.1, 0.8, 1.0\}$ and consider only the Taylor transform (TT) and the Unscented transform (TT). The Monte Carlo transform is numerically unstable at high dimensions. We display the mean and standard deviation across 10 different seeds after 10 EP iterations with $\gamma = 1.0$. As baselines, we also display the results of the corresponding Gaussian smoother and the IEKS after 10 iterations. For each dimension, we highlight in **bold**, the best results in the RMSE and the NLL. We see that in general, iterating EP improves the performance of the smoother, which can even be quite significant at high dimensions for the UT. An exception is the NLL performance at moderate-to-high dimensions when using the TT. The poor performance seen here is due to outlier results. If we exclude these, the result remains overall positive (see Figure 9).

# 6    Conclusion

We proposed an iterative state estimator for non-linear dynamical systems based on Expectation Propagation (EP). We show that any Gaussian filter or smoother can be shown as special case of the proposed method.

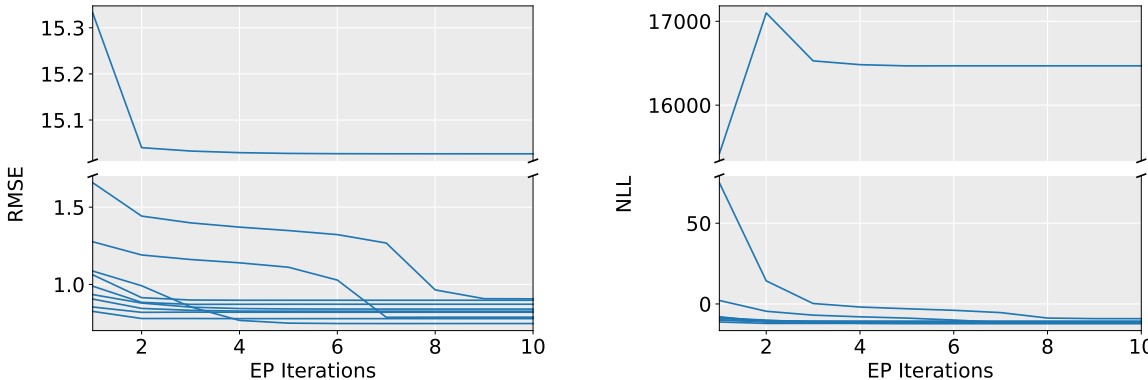

Figure 9: RMSE and NLL performance of the Lorenz 96 experiment where we used the TT for linearisation and set $(\alpha, \gamma) = (1.0, 1.0)$. The state dimension is set to $d = 20$. We display the results for 10 independent seeds. For most seeds, we observe that EP iterations improve predictions. However, there is one outlier that starts off badly after the first iteration. In this case, while subsequent iterations improve the RMSE performance, it deteriorates the NLL performance, leading to the results seen in Table 4, where the NLL performance becomes worse on average with iterations.

The standard derivative-based EP method is numerically unstable for non-linear systems (Seeger, 2005). We show that by explicitly accounting for the implicit linearisation typically used in Gaussian smoothers, we can use derivative-free EP updates, which closely resemble the generic Gaussian smoothing update equations, to build an iterative state estimator. The proposed method gets the best of both worlds, numerical stability of Gaussian state estimators and iterative improvements of the EP algorithm. We further improve the algorithm by using damping and power EP to improve convergence and performance.

Our experiments show the efficacy of EP for state estimation to improve standard (non-iterative) Gaussian filtering and smoothing methods on three benchmark problems. The iterative EP method still needs the user to choose the linearisation algorithm, which is identical to choosing the type of Gaussian filter in the standard setting. In the three examples, we showcase that iterating our approximate EP state-estimation algorithms (with the help of damping and power EP) generally lead to improved results in terms of the RMSE and NLL performance compared with their Kalman filter/smoother counterpart. Further improvements are also observed by (1) applying damping, which aids in convergence when EP is highly oscillatory, as was the case in the UNGM experiment, and (2) replacing standard EP with power EP, which makes the predictions more mode seeking. The latter was particularly beneficial in the bearings-only tracking experiment, where simply taking EP iterations did not result in improved performance (when taking the TT and UT for linearisation). However, setting $\alpha$ to smaller values lead to improved results. We also saw in the Lorenz 96 experiment that iterating EP may be particularly effective for higher-dimensional systems as we saw much more significant improvements at higher dimensions when considering the UT for linearisation. This may be potentially useful for data assimilation problems in weather forecasting (Carrassi et al., 2018; Evensen, 2009), which deals with filtering/smoothing in very high dimensions.

We have also compared our algorithm with another iterative smoother, namely the IEKS (Bell, 1994), which is an extension of the EKS that iteratively refines the linearisation point after every RTS smoother pass. While this can produce good results, sometimes even better than our EP algorithm as we saw in the bearings-only tracking example, these are merely MAP estimators and strictly speaking, do not predict uncertainties (although in our experiments, we have supplemented their outputs with covariance matrices generated by the RTS smoother at every iteration). EP on the other hand, outputs uncertainty naturally and moreover refines them iteratively without double counting observations. We have seen that on complex tasks such as the UNGM and the L96 experiments, power EP generally outperforms the IEKS.

Overall, EP-based state estimation generalises classical Gaussian smoothers (we proved that they are a special case), but it allows for iterative refinement of the state estimates. Furthermore, the message-passing formulation lends itself to asynchronous and distributed state estimation; we leave this for future work.

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

# A  Appendix: Some Useful Mathematical Identities

## A.1  Identities for Gaussians

For a random vector $\boldsymbol{x} \in \mathbb{R}^D$, we define the Gaussian probability density function as

$$\mathcal{N}(\boldsymbol{x} \mid \boldsymbol{\mu}, \boldsymbol{\Sigma}) := (2\pi)^{-\frac{D}{2}} |\boldsymbol{\Sigma}|^{-\frac{1}{2}} \exp\left(-\tfrac{1}{2}(\boldsymbol{x} - \boldsymbol{\mu})^\top \boldsymbol{\Sigma}^{-1}(\boldsymbol{x} - \boldsymbol{\mu})\right) \tag{125}$$

where $\boldsymbol{\mu} \in \mathbb{R}^D$ and $\boldsymbol{\Sigma} \in \mathbb{R}^{D \times D}$ are the mean and covariance function, respectively.

Partial derivatives of a log-Gaussian with respect to its mean and covariance are (Petersen & Pedersen, 2012):

$$\frac{\partial \log \mathcal{N}(\boldsymbol{x} \mid \boldsymbol{\mu}, \boldsymbol{\Sigma})}{\partial \boldsymbol{\mu}} = (\boldsymbol{x} - \boldsymbol{\mu})^\top \boldsymbol{\Sigma}^{-1} \tag{126}$$

$$\frac{\partial \log \mathcal{N}(\boldsymbol{x} \mid \boldsymbol{\mu}, \boldsymbol{\Sigma})}{\partial \boldsymbol{\Sigma}} = \frac{\boldsymbol{\Sigma}^{-1}(\boldsymbol{x} - \boldsymbol{\mu})(\boldsymbol{x} - \boldsymbol{\mu})^\top \boldsymbol{\Sigma}^{-1} - \boldsymbol{\Sigma}^{-1}}{2}. \tag{127}$$

For Gaussian distributions $\mathcal{N}(\boldsymbol{x} \mid \boldsymbol{a}, \boldsymbol{A})$, $\mathcal{N}(\boldsymbol{x} \mid \boldsymbol{b}, \boldsymbol{B})$, their multiplication and division results in another (unnormalised) Gaussian. For multiplication, we obtain

$$\mathcal{N}(\boldsymbol{x} \mid \boldsymbol{a}, \boldsymbol{A})\mathcal{N}(\boldsymbol{x} \mid \boldsymbol{b}, \boldsymbol{B}) = c\,\mathcal{N}(\boldsymbol{x} \mid \boldsymbol{c}, \boldsymbol{C}) \tag{128}$$

$$\boldsymbol{C} = (\boldsymbol{A}^{-1} + \boldsymbol{B}^{-1})^{-1} \tag{129}$$

$$\boldsymbol{c} = \boldsymbol{C}(\boldsymbol{A}^{-1}\boldsymbol{a} + \boldsymbol{B}^{-1}\boldsymbol{b}) \tag{130}$$

$$c = \mathcal{N}(\boldsymbol{a} \mid \boldsymbol{b}, \boldsymbol{A} + \boldsymbol{B}), \tag{131}$$

where in (131), $c$ is not a random variable (but a constant); however, we use the $\mathcal{N}$ notation for the functional form of $c$. The division of two Gaussians is given by

$$\frac{\mathcal{N}(\boldsymbol{x} \mid \boldsymbol{a}, \boldsymbol{A})}{\mathcal{N}(\boldsymbol{x} \mid \boldsymbol{b}, \boldsymbol{B})} \propto \mathcal{N}(\boldsymbol{x} \mid \boldsymbol{c}, \boldsymbol{C}) \tag{132}$$

$$\boldsymbol{C} = (\boldsymbol{A}^{-1} - \boldsymbol{B}^{-1})^{-1} \tag{133}$$

$$\boldsymbol{c} = \boldsymbol{C}(\boldsymbol{A}^{-1}\boldsymbol{a} - \boldsymbol{B}^{-1}\boldsymbol{b}). \tag{134}$$

Finally, raising a Gaussian to the power of $\alpha > 0$ gives

$$[\mathcal{N}(\boldsymbol{x} \mid \boldsymbol{a}, \boldsymbol{A})]^\alpha = \mathcal{N}(\boldsymbol{x} \mid \boldsymbol{a}, \alpha^{-1}\boldsymbol{A}). \tag{135}$$

## A.2  Identities for Matrix Inversion

A useful identity for matrix inversion (as a special case of the Woodbury identity) is

$$(\boldsymbol{A}^{-1} + \boldsymbol{B}^{-1})^{-1} = \boldsymbol{A}(\boldsymbol{A} + \boldsymbol{B})^{-1}\boldsymbol{B} \tag{136}$$

$$= \boldsymbol{A} - \boldsymbol{A}(\boldsymbol{A} + \boldsymbol{B})^{-1}\boldsymbol{A}. \tag{137}$$

## A.3  Metrics

Given the ground truth trajectory $(\boldsymbol{x}_t)$ and outputs $(\boldsymbol{\mu}_t, \boldsymbol{\Sigma}_t)$ of a given state-estimation algorithm for time steps $t = 1, \ldots, T$, we evaluate its performance using the root mean square error (RMSE) and the negative

log-likelihood (NLL). For $\boldsymbol{x}_t \in \mathbb{R}^D$, they are given by

$$RMSE = \sqrt{\frac{1}{T}\sum_{t=1}^{T}\|\boldsymbol{x}_t - \boldsymbol{\mu}_t\|^2}, \tag{138}$$

$$NLL = -\frac{1}{T}\sum_{t=1}^{T}\log\mathcal{N}(\boldsymbol{x}_t|\boldsymbol{\mu}_t, \boldsymbol{\Sigma}_t), \tag{139}$$

$$\log\mathcal{N}(\boldsymbol{x}_t|\boldsymbol{\mu}_t, \boldsymbol{\Sigma}_t) = -\frac{D}{2}\log(2\pi) - \frac{1}{2}\log|\boldsymbol{\Sigma}_t| - \frac{1}{2}(\boldsymbol{x}_t - \boldsymbol{\mu}_t)^\top\boldsymbol{\Sigma}_t^{-1}(\boldsymbol{x}_t - \boldsymbol{\mu}_t), \tag{140}$$

respectively. While the RMSE only tells us how close our predictive mean is to the ground truth, the NLL also evaluates the quality of our predictive uncertainties—it penalises heavily when the ground truth does not lie within some credible region around the predictive mean, or when the uncertainty is much larger than necessary even if it captures the ground truth.

## B  Appendix: Projections for Power EP

Here, we derive the expressions for the projections $\text{proj}\left[f_i^\alpha(\boldsymbol{x}_t)q_{\alpha\backslash i}(\boldsymbol{x}_t)\right]$ that are necessary in the implementation of power EP. Recall that we denoted by $q_{\alpha\backslash i}(\boldsymbol{x}_t) := q(\boldsymbol{x}_t)/q_i^\alpha(\boldsymbol{x}_t)$ the cavity distribution with respect to the fractional factor $q_i^\alpha(\boldsymbol{x}_t)$. In our implementation of power EP, we keep track of the fractional factors $q_i^\alpha(\boldsymbol{x}_t)$ instead of the full factors $q_i(\boldsymbol{x}_t)$. To make this explicit, we adopt the notation $r_i(\boldsymbol{x}_t) := q_i^\alpha(\boldsymbol{x}_t)$ below.

**Forward Projection**  Recall that the forward factor at time step $t$ is given by

$$f_\triangleright(\boldsymbol{x}_t) = \int p(\boldsymbol{x}_t|\boldsymbol{x}_{t-1})q_{\backslash\triangleleft}(\boldsymbol{x}_{t-1})\mathrm{d}\boldsymbol{x}_{t-1}. \tag{141}$$

In practice, we compute the cavity distribution $q_{\backslash\triangleleft}(\boldsymbol{x}_{t-1})$ from the fractional factor $r_\triangleleft(\boldsymbol{x}_{t-1})$ by taking

$$q_{\backslash\triangleleft}(\boldsymbol{x}_{t-1}) = q(\boldsymbol{x}_{t-1})/r_\triangleleft^{1/\alpha}(\boldsymbol{x}_{t-1}). \tag{142}$$

Following the arguments in Section 3.4, the message (141) can be approximated by

$$q_\triangleright(\boldsymbol{x}_t) = \mathcal{N}(\boldsymbol{x}_t \,|\, \boldsymbol{M}_{t-1}\boldsymbol{\mu}_{t-1}^{\backslash\triangleleft} + \boldsymbol{v}_{t-1}, \boldsymbol{M}_{t-1}\boldsymbol{\Sigma}_{t-1}^{\backslash\triangleleft}\boldsymbol{M}_{t-1}^\top + \boldsymbol{Q}) = \mathcal{N}(\boldsymbol{x}_t \,|\, \boldsymbol{\mu}_t^\triangleright, \boldsymbol{\Sigma}_t^\triangleright), \tag{143}$$

where $\boldsymbol{M}_{t-1}$ and $\boldsymbol{v}_{t-1}$ are a matrix and a vector respectively, obtained by implicit linearisation (Lemma 1). Thus, we find the following approximation to the fractional forward factor $f_\triangleright^\alpha(\boldsymbol{x}_t)$:

$$r_\triangleright(\boldsymbol{x}_t) = q_\triangleright^\alpha(\boldsymbol{x}_t) \overset{(135)}{=} \mathcal{N}(\boldsymbol{x}_t \,|\, \boldsymbol{\mu}_t^\triangleright, \alpha^{-1}\boldsymbol{\Sigma}_t^\triangleright). \tag{144}$$

**Measurement Projection**  As in standard EP, we derive the measurement projection for power EP via the partition function, which in this case, reads

$$Z_\triangle = \int f_\triangle^\alpha(\boldsymbol{x}_t)q_{\alpha\backslash\triangle}(\boldsymbol{x}_t)\mathrm{d}\boldsymbol{x}_t = \int \mathcal{N}(\boldsymbol{y}_t \,|\, \boldsymbol{h}(\boldsymbol{x}_t), \alpha^{-1}\boldsymbol{R})\, q_{\alpha\backslash\triangle}(\boldsymbol{x}_t)\mathrm{d}\boldsymbol{x}_t. \tag{145}$$

Let $q_{\alpha\backslash\triangle}(\boldsymbol{x}_t) = \mathcal{N}(\boldsymbol{x}_t \,|\, \boldsymbol{\mu}_t^{\alpha,x\backslash\triangle}, \boldsymbol{\Sigma}_t^{\alpha,x\backslash\triangle})$. By following the same procedure as in Section 3.4, we obtain the following approximation to $Z_\triangle$:

$$\tilde{Z}_\triangle = \mathcal{N}(\boldsymbol{y}_t \,|\, \boldsymbol{J}_t\boldsymbol{\mu}_t^{\alpha,x\backslash\triangle} + \boldsymbol{u}_t, \boldsymbol{J}_t\boldsymbol{\Sigma}_t^{\alpha,x\backslash\triangle}\boldsymbol{J}_t^\top + \alpha^{-1}\boldsymbol{R}) = \mathcal{N}(\boldsymbol{y}_t \,|\, \boldsymbol{\mu}_t^{\alpha,y\backslash\triangle}, \boldsymbol{\Sigma}_t^{\alpha,y\backslash\triangle}), \tag{146}$$

where as before, we used the implicit linearisation $\boldsymbol{y}_t = \boldsymbol{J}_t\boldsymbol{x}_t + \boldsymbol{u}_t + \boldsymbol{\varepsilon}$. The gradients of this approximate log partition function can thus be computed explicitly, which can be checked to be

$$\nabla_{\boldsymbol{\mu}\backslash\triangle} := \frac{\mathrm{d}\log\tilde{Z}_\triangle}{\mathrm{d}\boldsymbol{\mu}_t^{\alpha,x\backslash\triangle}} = (\boldsymbol{y}_t - \boldsymbol{\mu}_t^{\alpha,y\backslash\triangle})^\top(\boldsymbol{\Sigma}_t^{\alpha,y\backslash\triangle})^{-1}\boldsymbol{J}_t, \tag{147}$$

$$\nabla_{\boldsymbol{\Sigma}\backslash\triangle} := \frac{\mathrm{d}\log\tilde{Z}_\triangle}{\mathrm{d}\boldsymbol{\Sigma}_t^{\alpha,x\backslash\triangle}} = \frac{1}{2}\left(\nabla_{\boldsymbol{\mu}\backslash\triangle}\nabla_{\boldsymbol{\mu}\backslash\triangle}^\top - \boldsymbol{J}_t^\top(\boldsymbol{\Sigma}_t^{\alpha,y\backslash\triangle})^{-1}\boldsymbol{J}_t\right). \tag{148}$$

This gives us the update formulas (i.e., the moments of proj $\left[f_\Delta^\alpha(\boldsymbol{x}_t)q_{\alpha\setminus\Delta}(\boldsymbol{x}_t)\right] = \mathcal{N}(\boldsymbol{x}_t \,|\, \boldsymbol{\mu}_t', \boldsymbol{\Sigma}_t')$)

$$\boldsymbol{\mu}_t' = \boldsymbol{\mu}_t^{\alpha,x\setminus\Delta} + \boldsymbol{\Sigma}_t^{\alpha,x\setminus\Delta}\nabla_{\boldsymbol{\mu}\setminus\Delta}^\top \tag{149}$$

$$= \boldsymbol{\mu}_t^{\alpha,x\setminus\Delta} + \boldsymbol{\Sigma}_t^{\alpha,x\setminus\Delta}\boldsymbol{J}_t(\boldsymbol{\Sigma}_t^{\alpha,y\setminus\Delta})^{-1}(\boldsymbol{y}_t - \boldsymbol{\mu}_t^{\alpha,y\setminus\Delta}) \tag{150}$$

$$= \boldsymbol{\mu}_t^{\alpha,x\setminus\Delta} + \boldsymbol{K}_t^\alpha(\boldsymbol{y}_t - \boldsymbol{\mu}_t^{\alpha,y\setminus\Delta}) \tag{151}$$

$$\boldsymbol{\Sigma}_t' = \boldsymbol{\Sigma}_t^{\alpha,x\setminus\Delta} - \boldsymbol{\Sigma}_t^{\alpha,x\setminus\Delta}(\nabla_{\boldsymbol{\mu}\setminus\Delta}\nabla_{\boldsymbol{\mu}\setminus\Delta}^\top - 2\nabla_{\boldsymbol{\Sigma}\setminus\Delta})\boldsymbol{\Sigma}_t^{\alpha,x\setminus\Delta} \tag{152}$$

$$= \boldsymbol{\Sigma}_t^{\alpha,x\setminus\Delta} - \boldsymbol{\Sigma}_t^{\alpha,x\setminus\Delta}\boldsymbol{J}_t^\top(\boldsymbol{\Sigma}_t^{\alpha,y\setminus\Delta})^{-1}\boldsymbol{J}_t\boldsymbol{\Sigma}_t^{\alpha,x\setminus\Delta} \tag{153}$$

$$= \boldsymbol{\Sigma}_t^{\alpha,x\setminus\Delta} - \boldsymbol{K}_t^\alpha\boldsymbol{\Sigma}_t^{\alpha,yx\setminus\Delta}, \tag{154}$$

where $\boldsymbol{K}_t^\alpha := \boldsymbol{\Sigma}_t^{\alpha,xy\setminus\Delta}(\boldsymbol{\Sigma}_t^{\alpha,y\setminus\Delta})^{-1}$ is the Kalman gain matrix and $\boldsymbol{\Sigma}_t^{\alpha,xy\setminus\Delta} := \boldsymbol{\Sigma}_t^{\alpha,x\setminus\Delta}\boldsymbol{J}_t^\top$.

**Backward Projection**   Again, we use the derivatives of the log partition function to compute the backward projection proj $\left[f_\triangleleft^\alpha(\boldsymbol{x}_t)q_{\alpha\setminus\triangleleft}(\boldsymbol{x}_t)\right]$. The partition function corresponding to this case reads

$$Z_\triangleleft = \int f_\triangleleft^\alpha(\boldsymbol{x}_t)q_{\alpha\setminus\triangleleft}(\boldsymbol{x}_t)\mathrm{d}\boldsymbol{x}_t = \int \left(\int p(\boldsymbol{x}_{t+1}|\boldsymbol{x}_t)q_{\setminus\triangleright}(\boldsymbol{x}_{t+1})\mathrm{d}\boldsymbol{x}_{t+1}\right)^\alpha q_{\alpha\setminus\triangleleft}(\boldsymbol{x}_t)\mathrm{d}\boldsymbol{x}_t. \tag{155}$$

Using (131), we obtain the following useful identity, which allows us to move the power inside the integral

$$\left(\int p(\boldsymbol{x}_{t+1}|\boldsymbol{x}_t)q_{\setminus\triangleright}(\boldsymbol{x}_{t+1})\mathrm{d}\boldsymbol{x}_{t+1}\right)^\alpha = \left(\int \mathcal{N}(\boldsymbol{x}_{t+1}\,|\,\boldsymbol{f}(\boldsymbol{x}_t),\boldsymbol{Q})\,q_{\setminus\triangleright}(\boldsymbol{x}_{t+1})\mathrm{d}\boldsymbol{x}_{t+1}\right)^\alpha \tag{156}$$

$$\overset{(131)}{=} \left[\mathcal{N}(\boldsymbol{f}(\boldsymbol{x}_t)\,|\,\boldsymbol{\mu}_{t+1}^{\setminus\triangleright},\boldsymbol{Q}+\boldsymbol{\Sigma}_{t+1}^{\setminus\triangleright})\right]^\alpha \tag{157}$$

$$\overset{(135)}{=} \mathcal{N}(\boldsymbol{f}(\boldsymbol{x}_t)\,|\,\boldsymbol{\mu}_{t+1}^{\setminus\triangleright},\alpha^{-1}\boldsymbol{Q}+\alpha^{-1}\boldsymbol{\Sigma}_{t+1}^{\setminus\triangleright}) \tag{158}$$

$$\overset{(131)}{=} \int p(\boldsymbol{x}_{t+1}|\boldsymbol{x}_t)^\alpha q_{\setminus\triangleright}^\alpha(\boldsymbol{x}_{t+1})\mathrm{d}\boldsymbol{x}_{t+1}. \tag{159}$$

Thus, using Fubini's theorem to switch the order of integrals, we get

$$Z_\triangleleft = \int \left(\int p(\boldsymbol{x}_{t+1}|\boldsymbol{x}_t)q_{\setminus\triangleright}(\boldsymbol{x}_{t+1})\mathrm{d}\boldsymbol{x}_{t+1}\right)^\alpha q_{\alpha\setminus\triangleleft}(\boldsymbol{x}_t)\mathrm{d}\boldsymbol{x}_t \tag{160}$$

$$\overset{(159)}{=} \int \left(\int p(\boldsymbol{x}_{t+1}|\boldsymbol{x}_t)^\alpha q_{\setminus\triangleright}^\alpha(\boldsymbol{x}_{t+1})\mathrm{d}\boldsymbol{x}_{t+1}\right) q_{\alpha\setminus\triangleleft}(\boldsymbol{x}_t)\mathrm{d}\boldsymbol{x}_t \tag{161}$$

$$\overset{\text{Fubini}}{=} \int \left(\int p(\boldsymbol{x}_{t+1}|\boldsymbol{x}_t)^\alpha q_{\alpha\setminus\triangleleft}(\boldsymbol{x}_t)\mathrm{d}\boldsymbol{x}_t\right) q_{\setminus\triangleright}^\alpha(\boldsymbol{x}_{t+1})\mathrm{d}\boldsymbol{x}_{t+1}. \tag{162}$$

Now, using the implicit linearisation $\boldsymbol{x}_{t+1} = \boldsymbol{M}_t\boldsymbol{x}_t + \boldsymbol{v}_t + \boldsymbol{\varepsilon}$, we get the following approximation to $Z_\triangleleft$:

$$\tilde{Z}_\triangleleft = \int \mathcal{N}(\boldsymbol{x}_{t+1}\,|\,\underbrace{\boldsymbol{M}_t\boldsymbol{\mu}_t^{\alpha\setminus\triangleleft}+\boldsymbol{v}_t}_{=:\boldsymbol{\mu}_{t+1}^{\alpha,\triangleright}},\underbrace{\boldsymbol{M}_t\boldsymbol{\Sigma}_t^{\alpha\setminus\triangleleft}\boldsymbol{M}_t^\top+\alpha^{-1}\boldsymbol{Q}}_{=:\boldsymbol{\Sigma}_{t+1}^{\alpha,\triangleright}})q_{\setminus\triangleright}^\alpha(\boldsymbol{x}_{t+1})\mathrm{d}\boldsymbol{x}_{t+1} \tag{163}$$

$$\overset{(131)}{=} \mathcal{N}(\boldsymbol{\mu}_{t+1}^{\setminus\triangleright}\,|\,\boldsymbol{\mu}_{t+1}^{\alpha,\triangleright},\boldsymbol{\Sigma}_{t+1}^{\alpha,\triangleright}+\alpha^{-1}\boldsymbol{\Sigma}_{t+1}^{\setminus\triangleright}). \tag{164}$$

This gives us the following expressions for the gradients:

$$\nabla_{\boldsymbol{\mu}\setminus\triangleleft} := \frac{\mathrm{d}\log\tilde{Z}_\triangleleft}{\mathrm{d}\boldsymbol{\mu}_t^{\alpha,x\setminus\Delta}} = (\boldsymbol{\mu}_{t+1}^{\setminus\triangleright}-\boldsymbol{\mu}_{t+1}^{\alpha,\triangleright})^\top(\boldsymbol{\Sigma}_{t+1}^{\alpha,\triangleright}+\alpha^{-1}\boldsymbol{\Sigma}_{t+1}^{\setminus\triangleright})^{-1}\boldsymbol{M}_t, \tag{165}$$

$$\nabla_{\boldsymbol{\Sigma}\setminus\triangleleft} := \frac{\mathrm{d}\log\tilde{Z}_\triangleleft}{\mathrm{d}\boldsymbol{\Sigma}_t^{\alpha,x\setminus\Delta}} = \frac{1}{2}\left(\nabla_{\boldsymbol{\mu}\setminus\triangleleft}\nabla_{\boldsymbol{\mu}\setminus\triangleleft}^\top - \boldsymbol{M}_t^\top(\boldsymbol{\Sigma}_{t+1}^{\alpha,\triangleright}+\alpha^{-1}\boldsymbol{\Sigma}_{t+1}^{\setminus\triangleright})^{-1}\boldsymbol{M}_t\right). \tag{166}$$

Now set $q_{\alpha,\triangleright}(\boldsymbol{x}_{t+1}) := \mathcal{N}(\boldsymbol{x}_{t+1} \mid \boldsymbol{\mu}_{t+1}^{\alpha,\triangleright}, \boldsymbol{\Sigma}_{t+1}^{\alpha,\triangleright})$ and define $\tilde{q}(\boldsymbol{x}_{t+1}) := q_{\alpha,\triangleright}(\boldsymbol{x}_{t+1})q_{\backslash\triangleright}^{\alpha}(\boldsymbol{x}_{t+1})$, whose mean and covariance we denote by $\tilde{\boldsymbol{\mu}}_{t+1}$ and $\tilde{\boldsymbol{\Sigma}}_{t+1}$ respectively. In the case $\alpha = 1$, the distribution $\tilde{q}$ can be checked to be equivalent to the marginal $q(\boldsymbol{x}_{t+1})$. Since we have $\boldsymbol{M}_t = \boldsymbol{\Sigma}_{t,t+1}^{\alpha,\triangleright}(\boldsymbol{\Sigma}_t^{\alpha\backslash\triangleleft})^{-1}$ (follows from (19)), the update formulas for backward messages (i.e., moments of $\mathrm{proj}\left[f_\triangleleft^\alpha(\boldsymbol{x}_t)q_{\alpha\backslash\triangleleft}(\boldsymbol{x}_t)\right] = \mathcal{N}(\boldsymbol{x}_t \mid \boldsymbol{\mu}_t', \boldsymbol{\Sigma}_t'))$ can be checked to be

$$\boldsymbol{\mu}_t' = \boldsymbol{\mu}_t^{\alpha\backslash\triangleleft} + \boldsymbol{\Sigma}_t^{\alpha\backslash\triangleleft}\boldsymbol{M}_t^\top(\boldsymbol{\Sigma}_{t+1}^{\alpha,\triangleright} + \alpha^{-1}\boldsymbol{\Sigma}_{t+1}^{\backslash\triangleright})^{-1}(\boldsymbol{\mu}_{t+1}^{\backslash\triangleright} - \boldsymbol{\mu}_{t+1}^{\alpha,\triangleright}) \tag{167}$$

$$= \boldsymbol{\mu}_t^{\alpha\backslash\triangleleft} + \boldsymbol{L}_t^\alpha(\tilde{\boldsymbol{\mu}}_{t+1} - \boldsymbol{\mu}_{t+1}^{\alpha,\triangleright}) \tag{168}$$

$$\boldsymbol{\Sigma}_t' = \boldsymbol{\Sigma}_t^{\alpha\backslash\triangleleft} - \boldsymbol{\Sigma}_t^{\alpha\backslash\triangleleft}\boldsymbol{M}_t^\top(\boldsymbol{\Sigma}_{t+1}^{\alpha,\triangleright} + \alpha^{-1}\boldsymbol{\Sigma}_{t+1}^{\backslash\triangleright})^{-1}\boldsymbol{M}_t\boldsymbol{\Sigma}_t^{\alpha\backslash\triangleleft} \tag{169}$$

$$= \boldsymbol{\Sigma}_t^{\alpha\backslash\triangleleft} + (\boldsymbol{L}_t^\alpha)^\top(\tilde{\boldsymbol{\Sigma}}_{t+1} - \boldsymbol{\Sigma}_{t+1}^{\alpha,\triangleright})^{-1}\boldsymbol{L}_t^\alpha, \tag{170}$$

where $\boldsymbol{L}_t^\alpha = \boldsymbol{\Sigma}_{t,t+1}^{\alpha,\triangleright}(\boldsymbol{\Sigma}_{t+1}^{\alpha,\triangleright})^{-1}$ is the smoother gain. This can be checked in a similar way that we derived the standard RTS smoother update equations from the derivative-based backward EP updates in Section 3.4.

## C  Appendix: Additional Figures

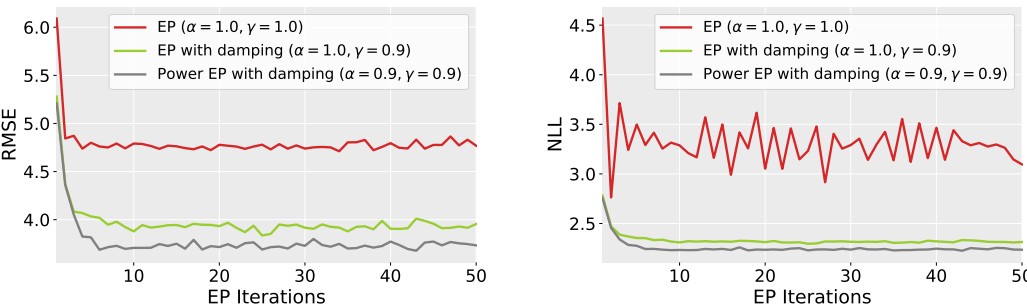

Figure 10: RMSE and NLL for EP (red), damped EP (green), and damped power EP (grey) for the UNGM experiment. The Monte Carlo transform (MCT) is used as the linearisation method here. Damping and power improves the performance significantly and reduces the oscillation seen in the NLL for standard EP.

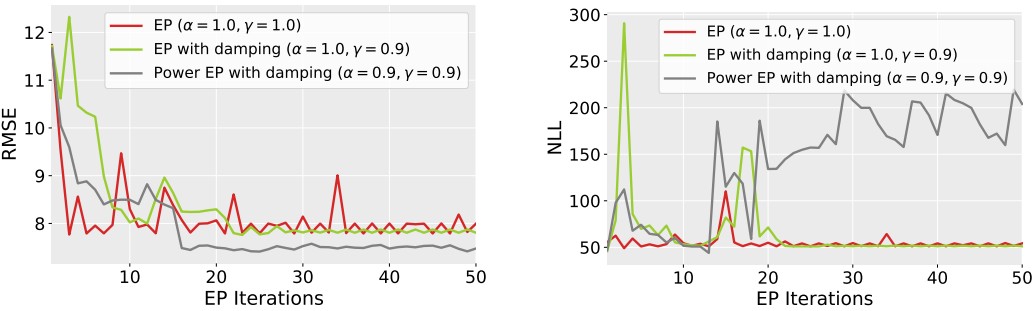

Figure 11: RMSE and NLL for EP (red), damped EP (green), and damped power EP (grey) for the UNGM experiment. The Taylor transform (TT) is used as the linearisation method here. Both damped EP and damped power EP reduce the oscillations seen in EP for the RMSE, with damped power EP even improving the performance. However, this is not reflected in the NLL, where power appears to degrade performance.

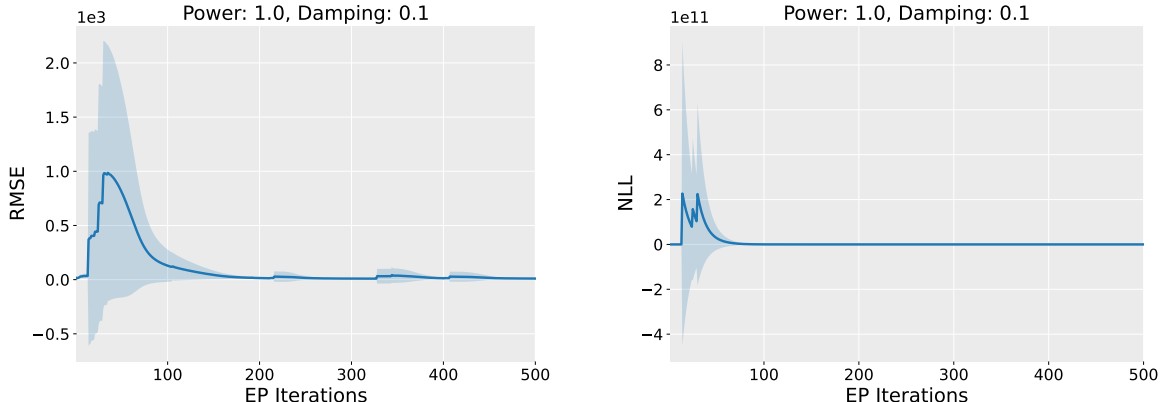

Figure 12: RMSE and NLL performance over multiple runs of the UNGM experiment, recorded for 500 iterations. We only consider the corner case ($\alpha = 1.0, \gamma = 0.1$) and used the Taylor transform (TT) for linearisation. The solid lines show the mean across ten different seeds and the shaded areas indicate the standard deviation. Initially, both the RMSE and NLL increase rapidly for a couple of iterations. However, they peak at around 25–30 iterations before starting to decrease. Eventually, they converge to good values after around 200 iterations for the RMSE and 100 iterations for the NLL.

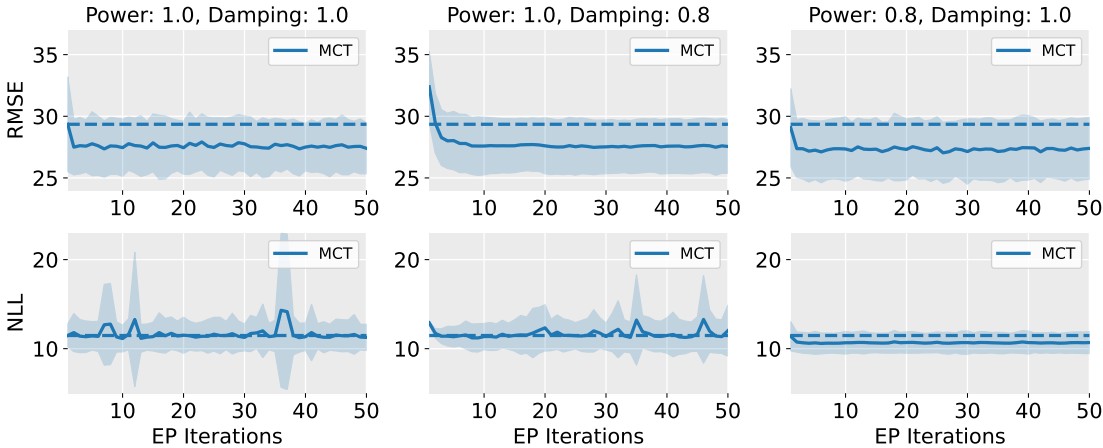

Figure 13: RMSE and NLL performance of the bearings-only turning target tracking problem, averaged over all variables (position, velocity, angular velocity). The MCT is used for linearisation. Solid line shows average performance across 10 random seeds and shaded regions indicate standard deviation. Dashed line shows the result of the Monte Carlo Kalman smoother. The NLL is highly oscillatory when $\alpha = 1.0$. Damping does little to reduce this oscillation, however it is easily controlled by setting $\alpha = 0.8$.

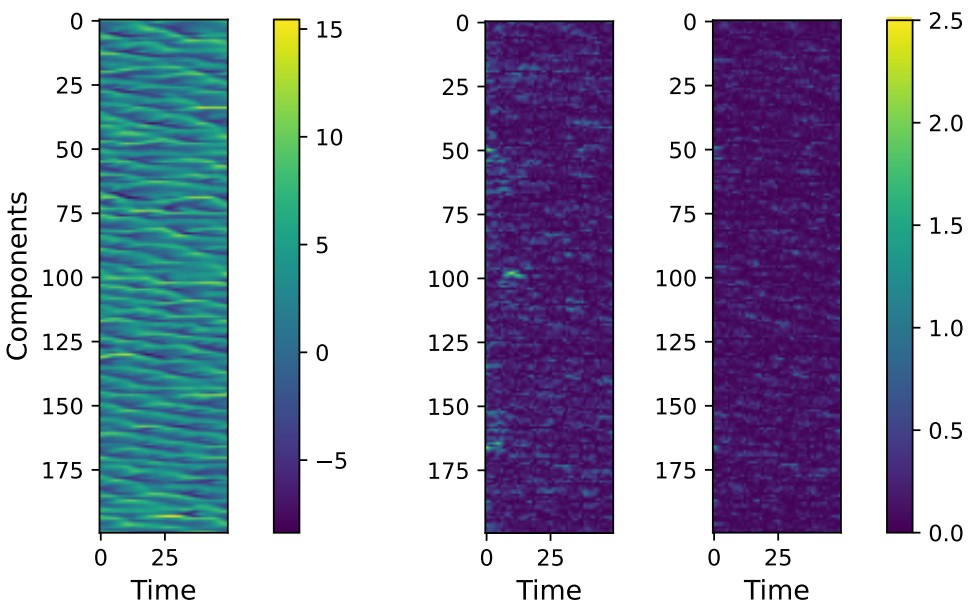

(a) Ground truth (a simulation of the Lorenz 96 model with $d = 200$).

(b) Absolute errors for the predictions ($|\boldsymbol{x}_{\mathrm{GT}} - \boldsymbol{x}_{\mathrm{pred}}|$). The left panel displays the result from the UKS. The right panel displays the result of EP after 10 iterations.

Figure 14: Hovmöller diagram depicting the time-evolution of each component in the Lorenz 96 system with 200 dimensions. In (a), we display the ground truth trajectory of a single simulation of the Lorenz 96 model. In (b), we display the absolute errors for the predictions made by the Unscented Kalman smoother (left) vs approximate EP after 10 iterations (right). We used the UT for the linearisation method in approximate EP. We see overall improvements in prediction quality by iterating EP.

