# OpenReview forum: "Iterative State Estimation in Non-linear Dynamical Systems Using Approximate Expectation Propagation"
_TMLR — Accepted by TMLR_

### Review · Reviewer_NPfV · 2022-04-13

**Summary Of Contributions:**

This paper describes the use of approximate expectation propagation (EP) to carry out approximate inference in non-linear dynamical systems. The method consists in approximating non-Gaussian distributions using a Gaussian distribution. This is done in an iterative process by using expectation propagation approximate factors. This factors are found by minimizing the more general alpha-divergence, instead of the typical KL divergence. The algorithm, after one iteration, is similar to already existing techniques. Performing more iterations is expected to lead to improvements, similar to those obtained of EP over ADF. This is verified in two experiments involving synthetic systems where the ground truth is known.  The results obtained indicate that damping the EP updates is needed for stability and that minimizing alpha-divergences can sometimes lead to better results.


**Requested Changes:**

Eq. (28) should be propto.

In Fig. 3 damping reduces oscillations, but does not provide convergence. Why is this? Is it due to the approximate EP updates? Does increasing the amount of damping improve more convergence? From my experience, non-convergent EP is very rare and by using enough damping EP always seems to converge in practice.

It is not clear what is the computational cost of the proposed method.

Approximate EP updates where the exact normalization constant of the factor times the cavity is also proposed and use in this work which could be relevant:

Bui, T. D., Hernández-Lobato, D., Hernández-Lobato, J. M., Li, Y., and Turner, R. E. (2016). Deep Gaussian
processes for regression using approximate expectation propagation. In International Conference on
Machine Learning, pages 1472–1481.

I am curious to know if a more principled approach in which the cavity distributions are all the same, as considered in that work, could be also applied to this setting. That could have the advantage of leaving aside the pesky EP update equations and could allow to use standard optimization methods. May be the authors, can comment on that.

Using only 50 iterations for EP at most seems like a small number. The authors also indicate that and in the appendix they use up to 500. Is there a reason for not using more than 50 iterations? I wonder if the authors can use something to check convergence. May be looking at changes in the approximate factors from one iteration to another?

I think that overall this is a nice paper that will be interesting for the community. However, the weakest part is the experimental section. I would suggest that the authors try to strengthen it by including at least another problem.


**Strengths And Weaknesses:**

Strengths:

 * The paper is clearly written.
 * The method is simple to use and can use many of the already existing techniques.
 * More iterations of EP often lead to better results.

Weaknesses:

 * There is not that much novelty in the proposed approach, which is based on already known techniques. However, this is not a critical point for this journal.
 * The experimental section is weak since it is based on only two problems that are synthetic.

---

> ### Author Response · Authors · 2022-04-27
> **propto**
>
> >Eq. (28) should be propto
>
> Agreed. Will be fixed with the next version we upload.

---

> ### Author Response · Authors · 2022-04-27
> **Damping and convergence**
>
> > In Fig. 3 damping reduces oscillations, but does not provide convergence. Why is this? Is it due to the approximate EP updates? Does increasing the amount of damping improve more convergence? From my experience, non-convergent EP is very rare and by using enough damping EP always seems to converge in practice.
>
> As the reviewer is likely aware, there is no guarantee that EP converges even with damping; only double loop EP can converge for unimodal or fairly linear systems. For non-linear systems, all bets are off from a theoretical perspective. However, we do observe better convergent behaviour by taking smaller damping factors. We will add a remark about this in the next iteration of the paper.

---

> ### Author Response · Authors · 2022-04-27
> **Computational cost**
>
> >It is not clear what is the computational cost of the proposed method.
>
> In Remark 3 (p. 16), we have stated the computational cost of iterative approximate EP: "The derivative-free updates, which are nearly identical to the generic updates for Gaussian filters/smoothers, tells us that the computational complexity of EP for state estimation is $O(KF)$, where $K$ is the number of EP iterations, and $F$ is the computational complexity of the corresponding Gaussian filter/smoother. This means that EP has the same computational complexity as a standard Gaussian filter/smoother for each EP iteration."
>
> Note that in the above remark, we have deliberately kept the cost $F$ of the Gaussian smoother abstract as this depends on the type of approximation used. The main point is that EP has the same computational cost as a Gaussian smoother (per iteration).

---

> ### Author Response · Authors · 2022-04-27
> **Relation to work by Bui et al.**
>
> >Approximate EP updates where the exact normalization constant of the factor times the cavity is also proposed and use in this work which could be relevant:
>
> > Bui, T. D., Hernández-Lobato, D., Hernández-Lobato, J. M., Li, Y., and Turner, R. E. (2016). Deep Gaussian processes for regression using approximate expectation propagation. In International Conference on Machine Learning, pages 1472–1481.
>
> > I am curious to know if a more principled approach in which the cavity distributions are all the same, as considered in that work, could be also applied to this setting. That could have the advantage of leaving aside the pesky EP update equations and could allow to use standard optimization methods. May be the authors, can comment on that.
>
>
> We would like to point out some reasons why we have not considered stochastic EP-style updates in this work:
>
> - This approach would not generalise nonlinear Gaussian filters/smoothers (since the correspondence shown in Table 1 would no longer hold true), which was one of the aims of this paper.
>
> - It is not immediately clear to us how to extend SEP (or the variant introduced in the work above) to our setting, especially since we consider marginals of the latent states, whose full distribution is non-separable (the state at time $t$ depends on the state at the previous timestep via a nonlinear transition function). The structure of our factor graph therefore calls naturally for sequential updates of the marginals as we do here.
>
> However, we agree that this is an interesting idea. In particular, replacing Kalman smoother-style updates by a single optimisation loop (if it can be done) may be an interesting alternative approach to iterative state-estimation that might yield some benefits, such as convergence guarantees.

---

> ### Author Response · Authors · 2022-04-27
> **Number of EP iterations**
>
> > Using only 50 iterations for EP at most seems like a small number. The authors also indicate that and in the appendix they use up to 500. Is there a reason for not using more than 50 iterations? I wonder if the authors can use something to check convergence. May be looking at changes in the approximate factors from one iteration to another?
>
> We observed that in most cases, a few (around 10) EP iterations were sufficient to reach converging behaviour when the damping factor was not tiny. Also from a practical viewpoint, using a very high number of iterations per experiment was simply too time consuming for our sensitivity analysis in section 5.1.2, where we ran over 2,000 experiments per CPU core.
>
> The experiments in the appendix only consider the cases when the damping factor is set very low (such as $\gamma=0.1$), where indeed, 50 iterations is not enough for convergence. We included these additional experiments with 500 iterations in the appendix to demonstrate that the negative results we see in the heat maps (Figures 6-7 and 9-10) for low damping factors are merely due to insufficient number of iterations.
>
> Regarding the second point, there exist heuristic convergence checks including changes in the approximate factors, but we chose to simply go for the number of iterations to avoid introducing more hyper-parameters, such as the thresholds.

---

> ### Author Response · Authors · 2022-04-27
> **Experiments**
>
> >I think that overall this is a nice paper that will be interesting for the community. However, the weakest part is the experimental section. I would suggest that the authors try to strengthen it by including at least another problem.
>
> Thank you for the encouraging comment. The experiments that we have on the synthetic problems serve two purposes. First, to demonstrate the equivalence of Gaussian filters and approximate EP in the first iteration (as shown in Figure 1) and second, to do an extensive set of ablation studies to investigate the effects of iteration, damping, power and approximation methods on the overall performance. For this purpose, we felt that the two commonly used example, which we have already in the paper, were sufficient. However, if the reviewer has any suggestion for an additional experiment that could shed further light on the inner workings and properties of approximate EP, we would be more than happy to include this in our paper.

---

> > ### Comment · Reviewer_NPfV · 2022-04-27
> > **Response to Authors**
> >
> > I am not familiar with the literature on non-linear dynamical systems. But I think that evaluating the method on only two problems is too little. I would simply suggest including another benchmark that is often used in the literature on methods tackling non-linear dynamical systems.

---

> > > ### Author Response · Authors · 2022-05-13
> > > **Added experiments**
> > >
> > > We have now added one more experiment (see section 5.3). We used the Lorenz 96 model to test how our method performs with respect to dimensions. We see positive results throughout, with increasingly significant performance with increased dimensions.
> > >
> > > We have also replaced our initial experiments for the bearings-only tracking with a more complicated example of bearings-only tracking of a turning target (see section 5.2), which is nonlinear with respect to both measurement and dynamics. This is because our initial example was too simple to give interesting results. Our renewed experiments give more interesting results where we observe that decreasing $\alpha$ yields better results in general.

---

### Review · Reviewer_JrZp · 2022-04-25

**Summary Of Contributions:**

This paper proposes an expectation propagation method that generalizes standard nonlinear filtering techniques. Interestingly, one iteration of the proposed method recovers some of the well-known nonlinear filters, while running many iterations has the promise of improving the properties of the estimators. It could be an interesting contribution to the filtering techniques.

**Broader Impact Concerns:**

There is no immediate ethical concerns regarding this work.

**Requested Changes:**

(i) While the authors seem to use the standard EP notation, I found it slightly confusing. It could be introduced better if the authors somehow find a way to introduce the proper conditioning to the variational distributions (e.g. what's called "marginal" q(x_t) is often conditioned on y_{1:t} or y_{1:T} and this is not clear from the context easily). Also while for the linear case, the correspondences in Table 1 are true, for the nonlinear case, these are approximations and maybe could be made clearer.

(ii) I urge authors to study a nonlinear example where things are more complex and conduct a sensitivity analysis for increasing dimension. This does not necessarily need to be a positive story but it could be important how much improvement is made, if one runs the EKF (i.e. a one iteration of this method) or iterate according to EP updates in a high-dimensional scenario. A Lorenz 96 example often used in the literature could be a good example and can be used to test unscented Kalman filter with a linear observation model. Note that this system allows one to test things with increasing dimension. Any other linear (but not independent) high dimensional system can also be tried like in the bearings-only scenario. Results with varying (increasing) d would be crucial for us to understand the usability of this methodology in a realistic setup.

**Strengths And Weaknesses:**

Strengths: The paper uses implicit linearization ideas to propose a generalization of nonlinear filters, which enables an improvement over standard "one-pass" methods. This is a welcome contribution. The derivation could be clarified (see below) but I found the idea important and useful.

Weaknesses: The notation seems slightly confusing and the numerical simulations can be improved.

---

> ### Author Response · Authors · 2022-04-27
> **Notation and correspondences in Table 1**
>
> >(i) While the authors seem to use the standard EP notation, I found it slightly confusing. It could be introduced better if the authors somehow find a way to introduce the proper conditioning to the variational distributions (e.g. what's called "marginal" $q(x_t)$ is often conditioned on $y_{1:t}$ or $y_{1:T}$ and this is not clear from the context easily). Also while for the linear case, the correspondences in Table 1 are true, for the nonlinear case, these are approximations and maybe could be made clearer.
>
> Regarding the notation: The marginal $q(x_t)$ is the marginal posterior conditioned on some observations as indicated by the reviewer. We chose not to explicit condition on the observations since this depends on the particular schedule of EP. While in this paper we choose to go with the sequential forward-backward schedule, this isn't really necessary, which would then result in marginal posteriors conditioned on a (non-consecutive) subset of observations. In the sequential case, after the first EP iteration, the posterior will always be conditioned on all observations $y_{1:T}$. To reduce notational burden, we left the conditioning out (this is generally done in variational inference as well, although we agree that it's not 100\% precise).
>
> Regarding the correspondences in Table 1: We propose to add a clarifying statement in the caption of the table. The reviewer is correct that the posteriors $p$ will only be approximate, not exact, when we have nonlinear transition/measurement functions.

---

> ### Author Response · Authors · 2022-05-13
> **Added Lorenz 96 experiment**
>
> > (ii) I urge authors to study a nonlinear example where things are more complex and conduct a sensitivity analysis for increasing dimension. This does not necessarily need to be a positive story but it could be important how much improvement is made, if one runs the EKF (i.e. a one iteration of this method) or iterate according to EP updates in a high-dimensional scenario. A Lorenz 96 example often used in the literature could be a good example and can be used to test unscented Kalman filter with a linear observation model. Note that this system allows one to test things with increasing dimension. Any other linear (but not independent) high dimensional system can also be tried like in the bearings-only scenario. Results with varying (increasing) d would be crucial for us to understand the usability of this methodology in a realistic setup.
>
>
> We have now added the Lorenz 96 experiment as requested (see section 5.3). We tested how our method performs with respect to increasing dimensions. We see positive results throughout, with increasingly significant performance with increased dimensions. We thank the reviewer for suggesting this.

---

### Review · Reviewer_NB9x · 2022-05-05

**Summary Of Contributions:**

The paper presents an algorithm to approximate the smoothing distribution for a nonlinear stochastic dynamical system. The approximation is in terms of a Gaussian distribution. The algorithm approximates the smoothing distribution p(x_t|y_1,...y_T) with product of certain Gaussian factors, and these factors are updated through an iterative message passing procedure going from t=0 to t=T. It is shown that a single iteration of the algorithm is equivalent to the Kalman smoother, and the algorithm can incorporate any linearization procedure. An extension of the algorithm with damping effects is also presented to remove the oscillatory behavior. The algorithm is illustrated with several numerical experiments in comparison with Kalman smoother.

**Broader Impact Concerns:**

I did not observe any broader impact concerns.

**Requested Changes:**

My suggestions are reflected in the weakness points written above.

**Strengths And Weaknesses:**

Strength:
- The paper is well-written overall (subject to minor improvements) and the mathematical results are correct to the best of my knowledge.

Weakness:
- I found the paper bit vague at the beginning on its objective: if the objective is filtering p(x_t| y_1,..,y_t) or smoothing p(x_t| y_1,..,y_T). There is a significant difference between the two. filtering is recursive, does not need it store the past observations, and give estimate of the state of the system at current time. Hence, it is useful and important for control. Smoothing requires storing all the observation from time zero to the current time, not scalable with the time horizon T, and give a better estimate of the state in the past which is not necessarily useful for control. So the paper needs to be more clear on its objective, and the presented motivation based on control does not seem reasonable.

- I am not sure why it was stated that implementing the Kalman smother iteratively is difficult. I found this article on iterative Kalman smoother [1]. This is important as it forms the correct baseline for comparison.

- Figure 1 should be explained better: how is RMSE calculated? (how many runs?) To which time step x_t does the reported rmse correspond to?

- The paper should present approaches on how to make the algorithm scalable with the time horizon to be implemented in real-time manner. How to extend the approximate Gaussians from T to T+1. Can a time-window procedure be implemented?

- I did not understand how the formulas for the factors in (50) and (70) are obtained. Is there an expression for the factors in (26) in terms of distributions? I would appreciate if the message passing is explained better for readers like me who are unfamiliar with the topic.

- Minor: it will be better to only number equations that are referenced.

[1] Bell, Bradley M. "The iterated Kalman smoother as a Gauss–Newton method." SIAM Journal on Optimization 4.3 (1994): 626-636.

---

> ### Author Response · Authors · 2022-05-13
> **Calculation of RMSE and NLL**
>
> > Figure 1 should be explained better: how is RMSE calculated? (how many runs?) To which time step x_t does the reported rmse correspond to?$(\mu_n, \Sigma_n)_{n=1}^N$
>
> Given the ground truth trajectory $(x_n)$ and outputs $(\mu_n, \Sigma_n)$ of the smoother for time steps $n=1, \ldots, N$, the formulae for the RMSE and the NLL that we use for evaluation are given as follows:
>
> $$RMSE = \sqrt{\frac{1}{N} \sum_{n=1}^N \|x_{n} - \mu_{n}\|^2}$$
>
> $$NLL = \frac{1}{N} \sum_{n=1}^N \log \mathcal{N}(x_{n} | \mu_{n}, \Sigma_{n}),$$
>
> where $\mathcal{N}(x | \mu, \Sigma)$ is the standard Gaussian pdf with mean $\mu$ and covariance $\Sigma$. This outputs a single number for each seed . It does not correspond to any time step in particular as it is averaged over time. We hope that this clarifies our figure better.

---

> > ### Comment · Reviewer_NB9x · 2022-05-19
> > **Respond to comment**
> >
> > Thank you! Adding this to the paper would clarify the figure.

---

> ### Author Response · Authors · 2022-05-13
> **Computations of factors (50) and (70)**
>
> > I did not understand how the formulas for the factors in (50) and (70) are obtained. Is there an expression for the factors in (26) in terms of distributions? I would appreciate if the message passing is explained better for readers like me who are unfamiliar with the topic.
>
> The computations of the factors (50) and (70) follow via a straightforward application of the sum-product algorithm (also known as belief propagation) applied to the factor graph in Figure 2a. We refer to Chapter 8.4 in the book:
>
> Bishop, Christopher. Pattern recognition and machine learning. New York: Springer, 2006.
>
> for a nice overview of this algorithm.
> We would also like to point out this excellent video lecture given by Tom Minka, where he explains how to derive equations (50) and (70) via message passing (note that the setting he considers in the 1:13:42 mark is exactly the same as ours):
>
> http://videolectures.net/mlss09uk_minka_ai/
>
> We have explained this procedure in passing in the caption of Figure 2.
>
> > Is there an expression for the factors in (26) in terms of distributions?
>
> The factors in (26) typically do not have a clean closed-form expression (except for the measurement message, which is just the observation likelihood). This is the reason why we consider approximations of the form (50) and (70). In the linear setting, equations (50) and (70) do indeed give exact expressions for the forward and backward factors, provided the messages are passed in the correct order (apply a forward pass first to update the filtering distributions, then a backward pass to get the smoothing distributions). In the nonlinear case, getting exact expressions for these messages are hopeless.
>
> >  I would appreciate if the message passing is explained better for readers like me who are unfamiliar with the topic
>
> We opt to simply refer to existing references for how message passing works as this will take up space and is in any case a commonly used framework in machine learning. We hope the reviewer understands.

---

> > ### Comment · Reviewer_NB9x · 2022-05-19
> > **Respond to comment**
> >
> > Thank you for the response and providing the link!

---

> ### Author Response · Authors · 2022-05-13
> **Clarifying the objective**
>
> > I found the paper bit vague at the beginning on its objective: if the objective is filtering p(x_t| y_1,..,y_t) or smoothing p(x_t| y_1,..,y_T). There is a significant difference between the two. filtering is recursive, does not need it store the past observations, and give estimate of the state of the system at current time. Hence, it is useful and important for control. Smoothing requires storing all the observation from time zero to the current time, not scalable with the time horizon T, and give a better estimate of the state in the past which is not necessarily useful for control. So the paper needs to be more clear on its objective, and the presented motivation based on control does not seem reasonable.
>
> Indeed, we mostly focus on smoothing problems in this paper. However, it's entirely possible to run EP also for filtering in an online inference setting. Here, we would run EP at every time step to refine the posterior (filtering distribution) by iterating the first $t$ time steps. This does make a difference, and it is comparable with the smoothing setting, where the posterior distribution at the final time step $T$ changes when iterating.
>
> > The paper should present approaches on how to make the algorithm scalable with the time horizon to be implemented in real-time manner. How to extend the approximate Gaussians from T to T+1.
>
> We believe it is not necessary to restrict ourselves to online settings. For instance in weather forecasting, commonly used data assimilation methods are based on smoothing (e.g. 4DVar) and is still used in real-time.
> The main idea that we want to convey is that, if we can get the result of a smoother at a reasonable computational cost, then we can obtain further improvements at a cost that scales only linearly in the number of EP iterations. This should therefore also be reasonable in most applications.

---

> > ### Comment · Reviewer_NB9x · 2022-05-19
> > **Respond to comment**
> >
> > Thank you for the response. Adding these explanations to the paper would help the reader significantly.

---

> ### Author Response · Authors · 2022-05-19
> **Online inference and time-window procedure**
>
> >The paper should present approaches on how to make the algorithm scalable with the time horizon to be implemented in real-time manner. How to extend the approximate Gaussians from T to T+1. Can a time-window procedure be implemented?
>
> The focus of the paper is on the smoothing part, i.e., the offline inference setting. However, a time-window procedure with message passing in the online setting could be implemented, e.g., by just updating the last $k$ messages. Iterations within the considered window would still give improved filtering distributions (as we can see that the smoothing distribution at time $T$ in our paper is also updated as we iterate).
>
> We hope that this answers the question.

---

> > ### Comment · Reviewer_NB9x · 2022-05-19
> > **Response to the comment.**
> >
> > Thanks. I think a discussion, as a remark, on the real-time implementation, that contains these points, and the ones from the comments below, would improve the paper

---

> > > ### Author Response · Authors · 2022-05-24
> > > **remark added**
> > >
> > > As suggested, we added a remark.

---

> ### Author Response · Authors · 2022-05-19
> **Added iterative Kalman smoother baseline**
>
> > I am not sure why it was stated that implementing the Kalman smother iteratively is difficult. I found this article on iterative Kalman smoother [1]. This is important as it forms the correct baseline for comparison.
>
> Thank you for pointing this out. We have now included the results from the iterative Kalman smoother [1] (we refer to this as IEKS) on all three experiments that we considered. We found that this baseline performs extremely well on our bearings-only tracking experiment, however struggled on the other experiments. A potential downside of IEKS is that it is a MAP estimator and therefore does not produce uncertainties that are theoretically consistent. Our EP algorithm, which produces Gaussian approximations to the true posterior, generally performs better than IEKS on the UNGM and L96 experiments and this might explain why. We have included this discussion in the text.

---

> > ### Comment · Reviewer_NB9x · 2022-05-19
> > **Response to comment**
> >
> > That is great. I think adding this baseline improves the contribution of the paper.

---

### Decision · Action_Editors · 2022-06-13

**Recommendation:** Accept as is

**Comment:**

All reviewers agreed that the contribution is interesting and relevant to the TMLR community. A common concern/suggestion about the experiments was addressed. After the rebuttal period, all reviewers were unanimous in their decision to accept this submission.